# A self-organized synthetic morphogenic liposome responds with shape changes to local light cues

Konstantin Gavriljuk[1,4], Bruno Scocozza[1,4], Farid Ghasemalizadeh [1,4], Hans Seidel [1,4], Akhilesh P. Nandan[1,3], Manuel Campos-Medina[1], Malte Schmick[1], Aneta Koseska[1,3] & Philippe I. H. Bastiaens [1,2✉]

Reconstituting artificial proto-cells capable of transducing extracellular signals into cytoskeletal changes can reveal fundamental principles of how non-equilibrium phenomena in cellular signal transduction affect morphogenesis. Here, we generated a Synthetic Morphogenic Membrane System (SynMMS) by encapsulating a dynamic microtubule (MT) aster and a light-inducible signaling system driven by GTP/ATP chemical potential into cell-sized liposomes. Responding to light cues in analogy to morphogens, this biomimetic design embodies basic principles of localized Rho-GTPase signal transduction that generate an intracellular MT-regulator signaling gradient. Light-induced signaling promotes membrane-deforming growth of MT-filaments by dynamically elevating the membrane-proximal tubulin concentration. The resulting membrane deformations enable recursive coupling of the MT-aster with the signaling system, which generates global self-organized morphologies that reorganize towards local external cues in dependence on prior shape. SynMMS thereby signifies a step towards bio-inspired engineering of self-organized cellular morphogenesis.

[1] Department of Systemic Cell Biology, Max Planck Institute for Molecular Physiology, Dortmund, Germany. [2] Faculty of Chemistry and Chemical Biology, TU Dortmund, Dortmund, Germany. [3] Present address: Cellular Computations and Learning, Center of Advanced European Studies and Research (caesar), Bonn, Germany. [4] These authors contributed equally: Konstantin Gavriljuk, Bruno Scocozza, Farid Ghasemalizadeh, Hans Seidel. ✉email: philippe.bastiaens@mpi-dortmund.mpg.de

Cells acquire their shape, which is tightly linked to their biological function, by dynamic cytoskeletal systems that deform the plasma membrane. Cell shape changes and motility are driven by the microtubule (MT) and actin filament systems, which operate at different length- and timescales[1]. Actin filaments dictate the rapid, more local morphological dynamics at the cell periphery, whereas MTs are far more long lived, persist over longer distances and are globally organized through MT-organizing centers. The dynamical organization of the MT-network therefore accounts for both, plasticity in shape formation in undifferentiated cells depending on environmental cues, but also shape stabilization after differentiation[2–6].

Cytoskeletal reorganization is directed by extracellular morphogens by inducing localized signaling reactions, generating a polarized cytoplasmic activity of MT-associated proteins. Prenylated Rho GTPases such as Rac thereby serve as switchable recruitment factors that concentrate cytoplasmic kinases e.g., PAK1 at morphogen exposed areas of the plasma membrane through the principle of dimensionality reduction[7–9]. The kinase then phosphorylates the negative MT-regulator stathmin[10] and alleviates its effect on suppressing MT growth resulting in MT growth towards the signal. However, cells do not reorganize their shape solely in response to extracellular signals, but integrate in their response previous sensory experiences in order to commit to distinct shapes during differentiation[11–13]. How memory of previous morphogen patterns is maintained and how this affects the signal-induced morphogenesis remains elusive.

In order to study principles of morphogen-induced morphogenesis, we reconstituted a minimal out-of-equilibrium system based on the physicochemical processes of the canonical Rac1-Pak1-stathmin pathway[14–17] in cell-sized liposomes with encapsulated dynamic MT-asters. The system was designed to respond to light cues, while maintaining the basic principles of localized signal transduction from morphogens that generate intracellular MT-regulator signaling gradients[18]. Our results show that the signaling gradient affects astral-MT growth leading to membrane deformations, which not only define cell shape, but also constitute a means by which the MT-cytoskeleton and signaling can recursively interact. This interaction causes these proto-cells to self-organize into shapes that can transform their morphology in response to localized light cues in dependence on their initial shape.

## Results

### The encapsulated MT-asters system.
To investigate the conditions under which centrally organized dynamic MTs generate vesicle morphologies by deforming the membrane of a giant unilamellar vesicle (GUV), we encapsulated purified centrosomes together with tubulin and GTP in GUVs using cDICE[19]. This generated dynamic[20] MT-asters inside GUVs of ~25 ± 5 μm diameter, which could be monitored by confocal laser scanning microscopy (CLSM) using trace amounts of fluorescently labeled Alexa568-tubulin (~10%, henceforth referred to as tubulin[568], Methods). Asters were formed around a single centrosome or a centrosome cluster acting as a single MT-organizing center. GUVs containing trace amounts of DOPE-biotin (0.05%) were immobilized to a biotinylated cover slide via biotin-cross-linking streptavidin (Methods). Aster size relative to the GUVs was controlled by encapsulating different concentrations of tubulin (Supplementary Fig. 1a–c) and the membrane tension was controlled by the outside osmolarity to generate GUVs with rigid or deformable membranes (Methods).

Encapsulating tubulin[568] concentrations between 15 and 25 μM in GUVs with rigid membranes (iso-osmotic conditions) resulted in a spherical morphology and asters smaller than the size of the GUV, with a centrally positioned centrosome.

Encapsulating higher tubulin[568] concentrations (35–40 μM) in GUVs with rigid membranes led to an average astral-MT length that was longer than the GUV radius, resulting in cortical MTs with peripheral centrosome positioning (Fig. 1a, b)[21]. When GUVs with high tubulin concentration were, however, formed under hyperosmotic conditions, the decreased luminal GUV volume enabled the MTs to deform the membrane with spiking protrusions (SPs) in which several MTs converged. This led to an asymmetric morphology with one or more polar SPs and a decentered centrosome, reflected in an increased GUV eccentricity and average MT length (Fig. 1a, b). The bundling of bending MTs into the protrusions indicated that MT-induced membrane deformations can further serve as capture sites for neighboring MTs, which constitutes an amplifying process of self-induced capture (SIC).

The net MT growth of small encapsulated asters was next induced by raising the temperature from 20 to 34 °C (Supplementary Fig. 1d). This is analogous to a uniform cytoplasmic signal that regulates MT-aster size by globally affecting growth kinetics of MTs[22,23]. In spherical GUVs with high membrane tension, temperature-induced MT growth led to centrosome decentering and the formation of semi-asters[21] (Fig. 1c; Supplementary Fig. 1e; Supplementary Video 1). In contrast, in GUVs with a deformable membrane, the system reorganized from a morphology with isotropically distributed microtubules to a stable polar protrusion morphology. The centrosome decentered as transient protrusions converged to a single one at the opposite pole (Fig. 1d; Supplementary Fig. 1f, Supplementary Video 1). This indicated that growing MT bundles in protrusions provided an initially random directional push to the centrosome, which further polarized the GUV by a recursive process of protrusion coalescence and centrosome decentering. Protrusion coalescence (Fig. 1d, Supplementary Fig. 1f) indicated that this process was driven by SIC, where initial MT-induced membrane deformations served as capture sites for neighboring MTs (Fig. 1e), thereby enlarging these deformations. Similar filamental organization at the membrane has previously been described for actin[24].

To investigate the basic rules underlying this dynamical transition to polar protrusion morphology, we described non-equilibrium fluctuations in MT-growth kinetics[25,26] and the cooperative bundling mechanism upon MT-induced membrane protrusions using an agent-based model. The model was implemented for a constant MT number[27] using Monte-Carlo simulations in a 1D circular geometry (Methods). For a defined number of nucleated MTs that evolve from fixed initial positions, the simulations showed that cooperative attraction of MTs by neighboring MTs in protrusions results in global organization of the MTs into two stable protrusions at opposed poles, as experimentally observed (Supplementary Fig. 1g, top, Supplementary Video 2, left). This suggested that the self-organization towards a polar morphology in the MT-membrane subsystem can be formally abstracted through a stigmergic[28–30] substrate-depletion model (Fig. 1f), where SIC of MTs into protrusions depletes free MTs. We calculated the kymograph representing the density of SIC-bundled MTs using a reaction–diffusion description of this substrate-depletion model in a simplified circular geometry with periodic boundary conditions, representing the angular distance along the circular GUV membrane (Fig. 1e, right). The results indeed reflected the formation of a stable polar pattern when starting from uniformly distributed free MTs (Fig. 1g, Supplementary Fig. 1h). The Monte-Carlo and the reaction–diffusion model additionally showed that formation of a star-like pattern formation would be possible for increased number of nucleated MTs (Supplementary Fig. 1g bottom, Supplementary Video 2, right and Supplementary Fig. 1i), corroborating the equivalence in the qualitative description of

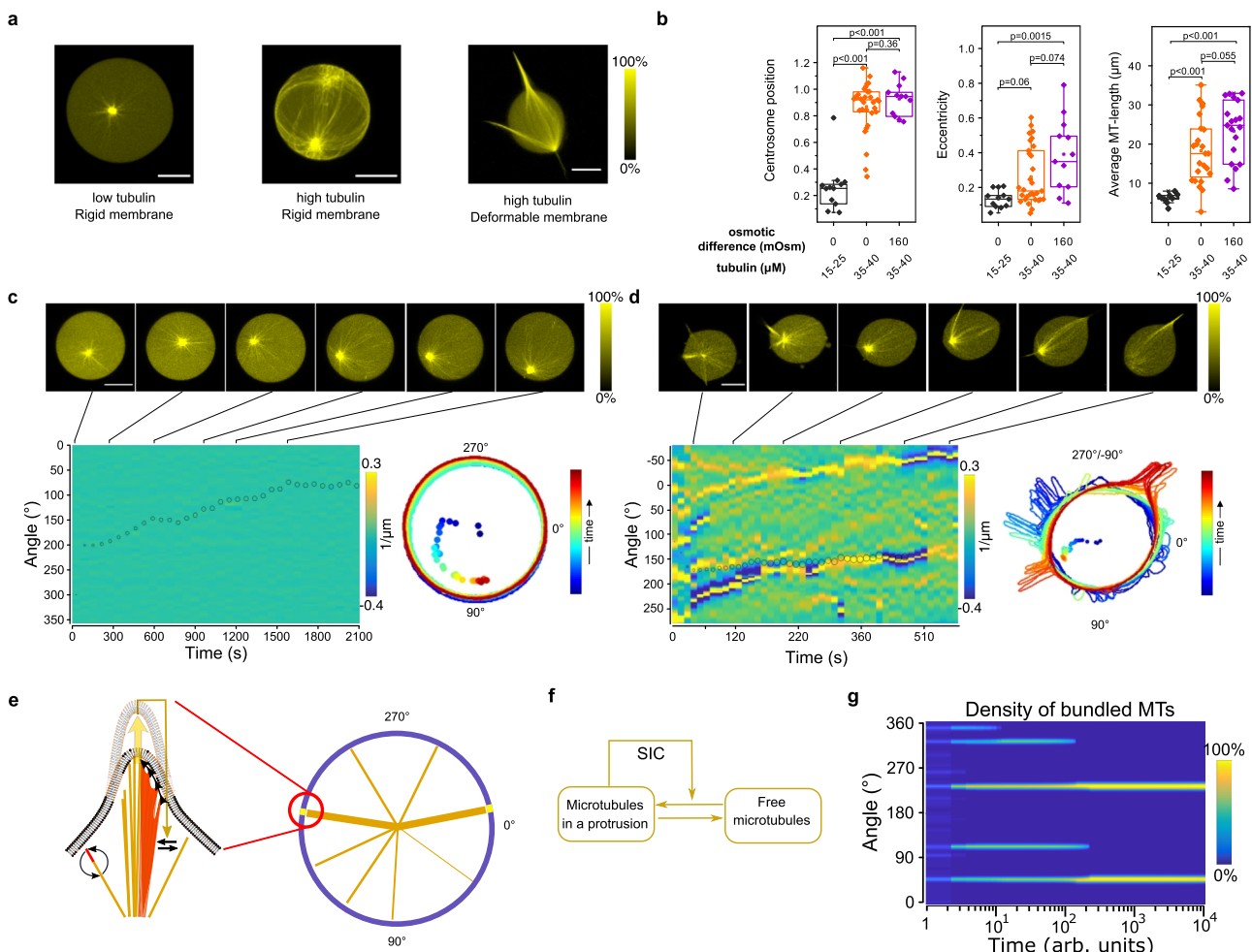

**Fig. 1 Self-organized morphologies of liposomes with encapsulated MT-asters. a** Representative 3D projections of encapsulated asters imaged by Confocal Laser Scanning Microscopy (CLSM) of tubulin[568] fluorescence (color bar: normalized intensity). Left to right: low tubulin (15–25 µM), rigid membrane (iso-osmotic, 0 mOsm); high tubulin (35–40 µM), rigid membrane; high tubulin, deformable membrane (hyperosmotic, 160 mOsm). **b** Morphometric and average microtubule (MT)-length quantification under conditions represented in **a**: low (black: 12, 12, 9 GUVs, respectively) or high tubulin (orange: 30, 30, 24) at iso-osmotic, and high tubulin at hyperosmotic (magenta: 12, 12, 19) conditions. Centrosome position: 0—centered, 1—membrane proximal, eccentricity: 0—perfect circle, >0—deformed circle (Methods). Boxplots: individual GUV (diamond), 25th and 75th percentile (box), 1.5 interquartile range (whiskers), median (line), and mean (square). *p*-values from two-sample Kolmogorov–Smirnov test with 95% confidence interval. **c** Top: CLSM time-lapse of temperature-change-induced aster growth in a GUV with rigid membrane (color bar: normalized fluorescence intensity). Bottom left: angular membrane curvature kymograph overlaid with centrosome position (small circle: centered, large circle: membrane proximal). Color bar: inverse radius (1/µm) of inscribed circle (Methods). Lines indicate times of connected micrographs. Bottom right: GUV contours during time-lapse, color coded by time (colored dots: corresponding centrosome positions). **d** CLSM time-lapse of temperature-change-induced aster growth in a GUV with deformable membrane (color bar: normalized fluorescence intensity). Bottom left: angular membrane curvature kymograph overlaid with centrosome position as in **c**. Bottom right: GUV contours during time-lapse, color coded by time progression as in **c**. See additional examples in Supplementary Fig. 1e, f. **e** Left pictogram: self-induced capture (SIC) of MTs (colored lines). MT-induced membrane deformations promote further capture of neighboring MTs (yellow feedback arrow onto black interconversion arrow), which bundle by sliding into the protrusion (arrowheads). Curved black arrows: MT-dynamic instability. Right: scheme of angular density of bundled MTs, simplified on a 1D circular geometry. **f** Pattern-generating MT-depletion model. Horizontal arrows: conversion between free and bundled MTs. Angled arrow: positive feedback mediated by SIC. **g** Angular density kymograph (color bar: normalized density) of MT-bundling described by model in **f**, simulated in 1D circular geometry (**e** right). Parameter settings; total amount of MTs: $c_1 = 1300$, SIC feedback strength: $\gamma_1 = 1$ (Methods). See also Supplementary Fig. 1h, i. Scale bars: 10 µm.

the MT-membrane subsystem dynamics within both frameworks. These results therefore suggest that the fundamental principle of SIC of dynamic MTs into protrusions together with their depletion elsewhere can describe the formation of distinct MT-induced shape patterns, as experimentally observed.

**Regulation of microtubule growth by the microtubule regulator stathmin.** In cells, the 17-kD cytoplasmic protein stathmin negatively regulates MT-dynamics by sequestering tubulin and thereby effectively decreases the concentration of free

αβ-heterodimers available for polymerization[17]. In response to extracellular stimuli, stathmin becomes phosphorylated on up to four residues (pStathmin), which inactivates its inhibitory effect on MT growth by reducing the affinity for soluble tubulin[31]. We quantified the effects of stathmin phosphorylation on reconstituted MT-dynamics by single-filament TIRF assays (Methods). Increase in stathmin concentration linearly decreased MT-growth speed and abruptly increased catastrophe frequency that could be reversed upon increasing the fraction of pStathmin at constant total Stathmin and tubulin concentration (Fig. 2a, b,

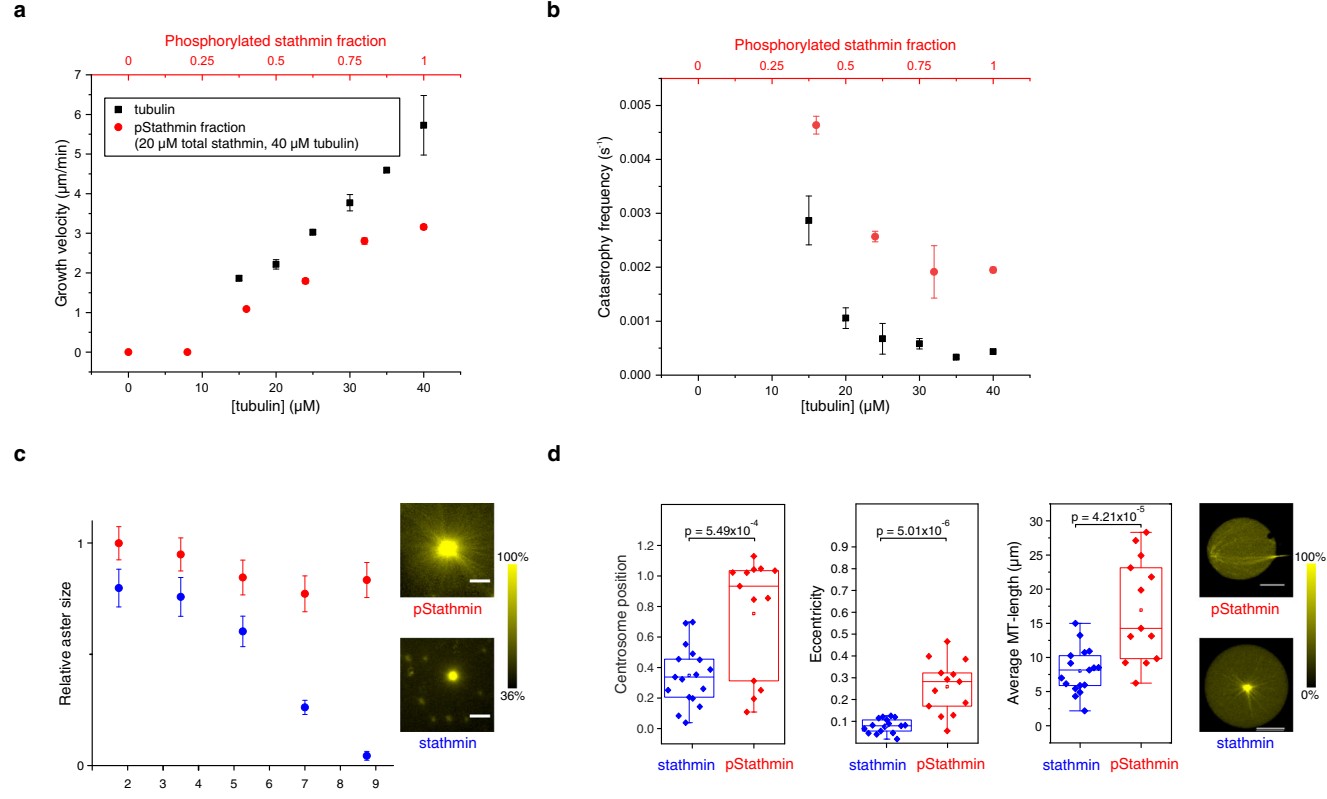

**Fig. 2 Regulation of MT-dynamics and aster size by phosphorylation of stathmin. a** Dependence of MT-growth velocity, and **b** catastrophe frequency on tubulin concentration (black squares, black axis on bottom) and fraction of phosphorylated stathmin (pStathmin, red dots, 40 μM tubulin and the fraction of phosphorylated stathmin was varied keeping a fixed total concentration of 20 μM stathmin, red axis on top) as determined by single-filament TIRF microscopy assays (mean ± S.E.M, 78 tracked filaments per condition from $N = 3$ independent experiments). See also Supplementary Fig. 2a, b. **c** Size of MT-asters on glass surface as a function of stathmin (blue dots) and pStathmin (red dots) concentration, at 35 μM tubulin. Error bars: Standard Error of the regression (Methods). Right: examples of multiple overlaid asters on glass surface (33 asters per condition from $N = 2$ independent experiments, yellow color bar denotes normalized tubulin[568] fluorescence intensity) in the presence of 7 μM pStathmin (top) or stathmin (bottom). **d** Morphometric and average MT-length quantification (as in Fig. 1b) of GUVs containing 5 ± 1 μM stathmin (17 GUVs, $N = 2$) or pStathmin (13 GUVs, $N = 3$), at 40 ± 7 μM encapsulated tubulin. Centrosome position: 0—centered, 1—membrane proximal, eccentricity: 0—perfect circle, >0—deviations from circle (Methods). Boxplots: individual GUV (diamond), 25th and 75th percentile (box), 1.5 interquartile range (whiskers), median (line), and mean (square). $p$-values from two-sample Kolmogorov–Smirnov test with 95% confidence interval. Right: examples of 3D confocal stack projections of tubulin[568] fluorescence (yellow color bar denotes normalized intensity) depicting GUV morphologies in the presence of pStathmin (top) or stathmin (bottom). Scale bars: 10 μm.

Supplementary Fig. 2a, b). Interaction of fluorescently tagged stathmin with MT-plus ends was not observed, concluding that stathmin affects MT-dynamics purely through sequestering free tubulin[17,32]. In this manner, increasing stathmin decreased the average size of reconstituted free asters on glass surfaces, whereas pStathmin hardly affected aster size in the same concentration range (Fig. 2c). Furthermore, encapsulation of 5 ± 1 μM stathmin and 40 ± 7 μM tubulin (final concentration considering encapsulation efficiency, Supplementary Fig. 2c, d) in GUVs with a deformable membrane resulted in decreased MT-aster size leading to a spherical morphology comparable to that obtained at lower tubulin concentrations. However, encapsulation of 5 ± 1 μM pStathmin and 40 ± 7 μM tubulin resulted in the formation of a polar GUV morphology as reflected in the centrosome positioning, increased MT length and GUV eccentricity (Fig. 2d). This was analogous to the GUV morphology observed with high tubulin (35–40 μM) alone (Fig. 1b). These results show that MT-induced GUV morphology can be biochemically controlled by a signaling system that phosphorylates stathmin and thereby reduces its capacity for tubulin sequestration. We therefore reconstituted a signal actuation system based on light-induced stathmin-kinase translocation to GUV membranes.

**Dimensionality reduction by light-induced kinase-translocation mimics intracellular signal actuation**. To capture the dimensionality reduction principle[7–9] of Rho-GTPase-activated kinase signaling, we encapsulated fusion constructs of improved Light-Inducible Dimer/Stringent starvation protein B (iLID/SspB) optical dimerizer system[33]. iLID was associated with the membrane via a fused C2 phosphatidylserine-binding domain (C2-iLID), while the SspB domain was fused to stathmin phosphorylating kinase AuroraB[34] (SspB-AuroraB). Encapsulation of both constructs enables translocation of AuroraB to a GUV membrane in response to 488 nm light stimuli. Step-wise increase of the 488 nm light dose induced repartitioning of fluorescent SspB-AuroraB[488] (Alexa488-SspB-AuroraB) to the membrane, being saturated at moderate light-doses (Fig. 3a). This saturable binding to C2-iLID limits maximally achievable SspB-AuroraB activity on the membrane and residual binding in the dark determines the basal activity as compared to the lumen (Supplementary Fig. 3a–c). Therefore, low encapsulation efficiency of C2-iLID with cDICE was the limiting factor on controlling light-induced SspB-AuroraB translocation amplitude as well as the main cause for variance in SspB-AuroraB translocation efficiency in different GUVs (Supplementary Fig. 3d, e).

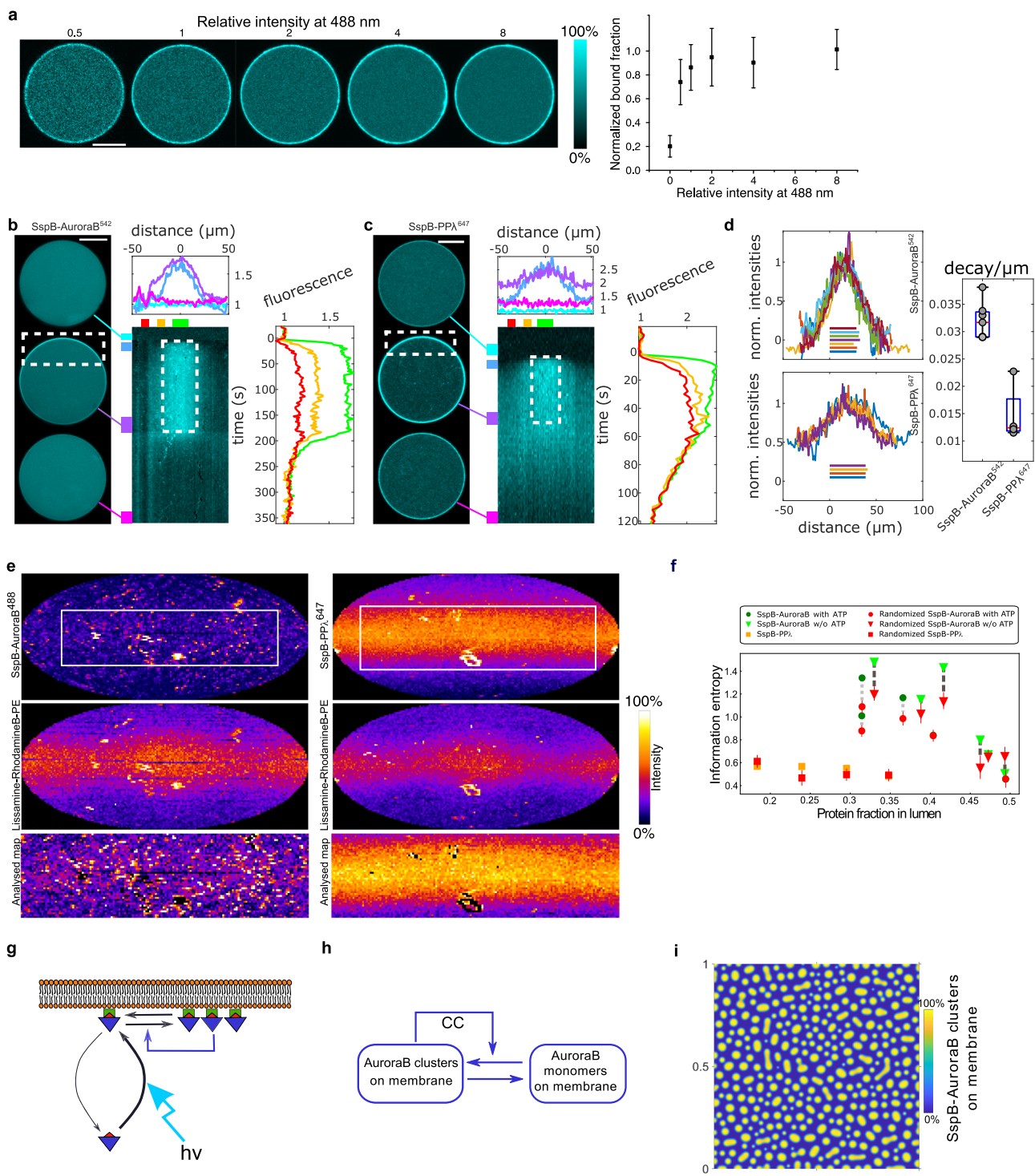

We next investigated if SspB-AuroraB translocation could be localized to a 488 nm irradiated area on the GUV membrane. In order to also observe translocation outside irradiated membrane areas, as well as the kinetics of dissociation upon 488 nm light removal, SspB-AuroraB[542] was labeled with Atto542 (Atto542-SspB-AuroraB, Methods), which could be excited separately from C2-iLID with 542 nm light. Continuous 488 nm light irradiation in a locally confined area resulted in a pronounced SspB-AuroraB[542] translocation to that area within ~10 s. During further 488 nm illumination, a gradient became apparent along the membrane extending from the irradiated membrane region that reached steady state after ~50 s. This gradient had a maximal

amplitude in the center of the irradiated area and prestimulus levels on the opposite side of the GUV. SspB-AuroraB[542] on the membrane reverted to prestimulus levels when the illumination was stopped (Fig. 3b, d). The ensuing gradient steepness therefore results from a convolution of SspB-AuroraB dissociation rate from C2-iLID in dark areas, and lateral diffusion of C2-iLID/SspB-AuroraB on the membrane. These results demonstrate that upon light irradiation, the C2-iLID/SspB-AuroraB system can mimic a spatially confined signaling response to a localized extracellular morphogen signal.

We also fused another protein, protein phosphatase lambda (PPλ), to SspB (SspB-PPλ). In this case, equivalent localized

**Fig. 3 Light-induced control of SspB-AuroraB membrane translocation and patterning. a** Light-dose-dependent translocation of encapsulated SspB-AuroraB$^{488}$ (4.9 ± 0.9 μM) to C2-iLID (7 ± 4 10$^{-1}$ pmol/cm$^2$) on the membrane. Relative intensity normalized to 65 mW/cm$^2$. Left: representative CLSM fluorescence images (color bar: normalized fluorescence intensity), right: quantification (mean ± S.D., $n = 5$, $n = 17$ dark-binding: 0 on $x$-axis, approximated by initial frame). See also Supplementary Fig. 3a, b. **b** Locally irradiated GUV (encapsulated: C2-iLID, SspB-AuroraB$^{542}$). Normalized frames (left) -connected with lines to fluorescence kymograph (middle, cyan-to-magenta boxes indicate time-averaged frames). $X$-axis: distance along membrane from top; $y$-axis: time, dashed boxes: 488-nm-irradiated duration/position. Profiles top/next to kymograph are averaged time/space slices marked by colored boxes (time: cyan-to-magenta; space: green-to-red). Fluorescence normalized to average intensity/frame. **c** Identical to **b** but with encapsulated C2-iLID and SspB-PPλ$^{647}$. **d** Peak normalized spatial profiles of last photo-recruitment frame for SspB-AuroraB$^{542}$ (top, 7 GUVs, $N = 3$) and SspB-PPλ$^{647}$ (bottom, 4 GUVs, $N = 2$). Colored horizontal lines: irradiation area. Box-plot: decay/μm of photo-recruitment from linear regression (left flank's slope); datapoints (dots), 25th and 75th percentile (blue box), min-max (whiskers), median (red line). **e** Representative translocated SspB-AuroraB$^{488}$ (top left) or SspB-PPλ$^{647}$ fluorescence map (top right) in GUVs (color bar: normalized intensity). Middle: corresponding Lissamine-RhodamineB-PE fluorescence-distribution. Bottom: cropped maps (white rectangular areas top row), excluding lipid irregularities apparent in middle row (methods). **f** Cluster-pattern recurrence quantified by information entropy as function of luminal-protein-depletion (Supplementary Fig. 4a–d). Dark green: SspB-AuroraB$^{488}$ with 2 mM ATP, light green: without ATP, orange: SspB-PPλ$^{647}$, red: information entropy upon numerical cluster-pattern randomization ($n = 50$, mean ± S.D., connected with dashed lines). **g** Cooperative clustering (CC) of SspB-AuroraB. Light ($h\nu$) induced SspB-AuroraB (blue triangles, red SspB) translocation (curved black arrow) to C2-iLID (green squares) on membrane drives cluster formation (black arrows), amplified by a positive feedback mechanism (blue arrow). **h** Scheme of the depletion model where CC causes depletion of SspB-AuroraB monomers on the membrane thereby generating cluster patterns. Horizontal arrows: monomer-to-cluster interconversion. Angled arrow: CC mediates positive feedback. **i** Self-organized SspB-AuroraB pattern in 2D Euclidean geometry from numerical simulations of the model in **h**. Color bar: normalized density of clustered SspB-AuroraB. Equations in Methods. See also Supplementary Fig. 4e–h. Scale bars: 10 μm.

irradiation resulted in a half-as-steep gradient of the encapsulated SspB-PPλ$^{647}$ (Alexa647-SspB-PPλ) in comparison to the one of SspB-AuroraB$^{542}$ (Fig. 3c, d), reverting to the basal state on the same timescale upon switching off irradiation. A slow-down of diffusion by AuroraB oligomerization could therefore explain the steep steady state SspB-AuroraB gradient along the membrane upon local irradiation.

We therefore investigated if SspB-AuroraB forms clusters on the membrane upon 488 nm irradiation by obtaining fluorescence intensity membrane surface distributions from 3D confocal micrographs of GUVs encapsulating iLID/SspB-AuroraB$^{488}$ (Fig. 3e). The SspB-AuroraB$^{488}$ construct enabled simultaneous C2-iLID activation and SspB-AuroraB$^{488}$ fluorescence visualization, thereby liberating spectral windows and was used in all further experiments. A trace amount of fluorescent Lissamine-RhodamineB-PE lipid was incorporated in the GUVs to verify if lipid irregularities cause SspB-AuroraB clusters on the membrane. These irregularities coincided with large SspB-AuroraB patches and were preferentially formed in the absence of ATP, indicating that these were misfolded, inactive aggregates. However, outside of these irregularities, small SspB-AuroraB$^{488}$ clusters were formed on the membrane upon its light-induced translocation, irrespective of ATP and therefore kinase activity. This clustering was a specific property of AuroraB and not of the iLID/SspB proteins, since it did not occur for encapsulated iLID/SspB-PPλ$^{647}$ upon 488 nm irradiation outside areas containing lipid distribution irregularities (Fig. 3e; Supplementary Fig. 4a–c).

We next investigated if a self-organized process underlies the formation of SspB-AuroraB clusters on the membrane by quantifying the regularity in SspB-AuroraB$^{488}$ cluster distributions outside of the areas with lipid irregularities (Fig. 3e, Supplementary Fig. 4a, b). For this, we computed cluster-pattern recurrences and quantified these by means of information entropy[35] (Methods) (Fig. 3f; Supplementary Fig. 4d). The spatial order of cluster distributions increased for higher SspB-AuroraB$^{488}$ recruitment on the membrane (coincident with depletion from lumen), which did not occur for SspB-PPλ$^{647}$ (Fig. 3e, f; Supplementary Fig. 4c). This increase in spatial order was irrespective of ATP, showing that it was not driven by ATP hydrolysis, but instead by the light-dependent dimerization of iLID/SspB that determines the steady-state concentration of SspB-AuroraB on the membrane. Randomization of the spatial localization of the SspB-AuroraB$^{488}$ clusters significantly

decreased the information entropy values above a threshold translocation, whereas this was not the case for SspB-PPλ$^{647}$ (Fig. 3f). This shows that a positive feedback on SspB-AuroraB cluster formation on the membrane together with monomer depletion underlies the self-organized regularity of the SspB-AuroraB cluster patterns (Fig. 3g). The positive feedback is likely a concentration dependent cooperative clustering (CC) mechanism based on weak multivalent interactions among SspB-AuroraB monomers.

Simulations of the model of interconversion between clusters and monomers at the membrane aided by the cooperative clustering effect on a 2D Euclidean grid with no-flux boundary conditions showed the formation of SspB-AuroraB patterns (Fig. 3h, i, Methods). These self-organized patterns could be generated above a threshold concentration of SspB-AuroraB on the membrane through a symmetry-breaking bifurcation (Supplementary Fig. 4e). Given that a fixed amount of SspB-AuroraB is translocated to the membrane limited by the amount of available iLID; considering only the membrane-bound SspB-AuroraB species resulted in an equivalent transition to a patterned solution as when explicitly modeling the SspB-AuroraB monomer translocation from the lumen (Supplementary Fig. 4f, g). Similarly as in the experiments, the stable self-organized SspB-AuroraB cluster pattern (Fig. 3i) was characterized by high entropy values, which were decreased upon randomization of the cluster distributions; in contrast to the randomization at initial timepoints of the simulation, before the pattern was stabilized (Supplementary Fig. 4h). The simulations thereby corroborate that the measured cluster regularities are based on a self-organized process of self-amplified SspB-AuroraB clustering together with monomer depletion. Thus formally, the basic principle of self-organized dynamical transitions in the MT-membrane as well as the SspB-AuroraB signaling subsystem equivalently relies on self-amplification through substrate-depletion.

**A membrane-proximal stathmin phosphorylation cycle concentrates free tubulin.** In order to generate a synthetic signaling system based on a stathmin (de)phosphorylation cycle, we encapsulated stathmin together with SspB-AuroraB$^{488}$ and PPλ. To first determine if this cycle can establish a steady-state of stathmin phosphorylation, we measured kinetics of stathmin phosphorylation in solution by sequential addition of

SspB-AuroraB[488] and PPλ (Supplementary Table 1) in the presence of soluble tubulin (20 μM). Stathmin phosphorylation was monitored using an organic dye-containing variant of the stathmin phosphorylation FRET-sensor COPY[18] (Atto532-stathmin-Atto655: COPY[o] (organic)) (Supplementary Fig. 5a, b). Ratiometric quantification of COPY[o] (10 μM) phosphorylation after sequential addition of 2 μM SspB-AuroraB[488] and 0.5 μM PPλ, demonstrated that (de)phosphorylation cycles can maintain a steady-state phosphorylation level of stathmin (~20%) in the presence of ATP in solution (Fig. 4a; Supplementary Fig. 5c). This steady-state phosphorylation level of stathmin corresponds well with the ~16% as calculated from kinetic parameters obtained from stopped-flow experiments (Supplementary Fig. 5d, e, Supplementary Table 1), which shows that the higher overall dephosphorylating activity of PPλ ($k_{cat}/K_M$: 22 $10^3 s^{-1} M^{-1}$) with respect to the overall phosphorylating activity ($k_{cat}/K_M$: 11 $10^2 s^{-1} M^{-1}$) of SspB-AuroraB[488] tends to maintain a low pStathmin level in solution.

To next investigate if the SspB-AuroraB[488] kinase translocation can increase the steady-state phosphorylation level of stathmin in GUVs, we encapsulated COPY[o] ($4 \pm 0.8$ μM) together with the iLID/SspB-AuroraB/PPλ cyclic signaling system ([SspB-AuroraB] = $5 \pm 0.9$ μM, [PPλ] = $5 \pm 1$ $10^{-1}$ μM, estimated final concentration). FLIM-FRET measurements showed that COPY[o] was maintained in a mostly dephosphorylated steady state in the lumen of GUVs in the presence of PPλ phosphatase activity (Fig. 4b), in agreement with the estimated steady-state pStathmin level (~33%) obtained from kinetic parameters. Upon light-induced translocation of SspB-AuroraB[488], a steep pStathmin gradient (~0.5 μm decay length) could be observed by ratiometric FRET-imaging that emanated from the GUV membrane. This gradient only occurred in the presence of PPλ (Fig. 4c, d; Supplementary Fig. 5f–i), showing that it was dynamically maintained by a (de)phosphorylation cycle (Fig. 4e). Furthermore, 3D projections of ratiometric confocal stacks exhibited bright spots indicating that stathmin phosphorylation on the membrane is highest in discrete locations that most likely originate from high-activity SspB-AuroraB clusters (Supplementary Fig. 5j).

In order to understand how a dynamic (de)phosphorylation cycle of stathmin affects tubulin release from and sequestration on stathmin, we simulated the interaction between tubulin and stathmin with a reaction–diffusion model using measured enzymatic and association/dissociation parameters in a radial 1D geometry (Supplementary Fig. 6a–k; Supplementary Table 1; Methods). This confirmed that recruiting SspB-AuroraB activity to the membrane can locally overcome the luminal PPλ activity to yield a pStathmin gradient with a decay length comparable to the experimental observations (Fig. 4f). However, the numerical simulations also revealed that the release of tubulin from phosphorylated stathmin results in a tubulin gradient (Fig. 4f; Supplementary Fig. 6i, j). This steady-state gradient can only be generated if dephosphorylation of stathmin establishes a tubulin release/sequestration cycle, causing a flux of tubulin to the membrane that is countered by its diffusional equilibration into the lumen of the GUV (Fig. 4g). A signal-induced recruitment of a kinase to the membrane thus leads to a dynamically maintained enhanced level of pStathmin as well as tubulin near the membrane.

**A light-responsive encapsulated microtubule aster-signaling system.** To test if the light-induced membrane-proximal tubulin concentration can induce MT growth, we encapsulated the signaling system together with tubulin[647] (10% Alexa647-tubulin) and centrosomes in GUVs. We refer to encapsulated MT-signaling systems as Synthetic Morphogenic Membrane System (SynMMS). In SynMMS with a rigid membrane (iso-osmotic), light-induced SspB-AuroraB[488] translocation (quantified relative to the lumen: R, Methods), resulted in centrosome decentering (Fig. 5a–c, Supplementary Video 3), reflected in the enhanced Relative Centrosome Displacement (RCD, Fig. 5d, Methods) as compared to before translocation. The centrosome reverted to a central positioning upon light removal. Furthermore, quantification of SspB-AuroraB[488] fluorescence as function of time in angular kymographs demonstrated speckles of SspB-AuroraB[488] fluorescence, which indicated that transient interactions of dynamic MTs with the membrane briefly enhanced SspB-AuroraB[488] clustering (Fig. 5e). However, in control SynMMSs with a rigid membrane that lacked stathmin (SynMMS[-stat]), relative centrosome movement did not change upon light-induced activation of SspB-AuroraB[488] (Fig. 5d, Supplementary Fig. 7a–c), showing that stathmin is essential for coupling SspB-AuroraB translocation to enhanced MT growth near the membrane.

On the other hand, in an osmotically strongly deflated SynMMS (160 mOsm) with polar protrusions from a large aster, light-induced translocation of SspB-AuroraB[488] induced a net astral-MT growth that drastically elongated and reoriented the liposome (Fig. 5f–i, Supplementary Video 4). In this case however, the SspB-AuroraB[488] fluorescence kymograph demonstrated that SspB-AuroraB[488] accumulated in the main protrusions over time (Fig. 5j). Overall, this shows that the light-induced membrane-proximal tubulin concentration can induce MT growth and thereby induce a change in initial SynMMS morphology.

**Initial morphological states of SynMMS are determined by MT-density and basal signaling.** SynMMS exhibited different initial morphologies, which could be divided in three general classes: in addition to a spherical morphology with small asters and a polar protrusion morphology with large asters, a substantial population exhibited a star-like morphology with axially distributed protrusions (Fig. 5k, Supplementary Fig. 7d). All protrusions were enveloped by membrane without perforating the bilayer (Supplementary Fig. 7e). In some cases, SynMMSs incorporated small spherical structures, which were identified as lipid vesicles (Supplementary Fig. 7f). The star morphology hardly occurred for encapsulated asters alone with equivalent tubulin concentration (Fig. 5k, Methods). This indicates that basal SspB-AuroraB[488] signaling from the membrane has a role in stabilizing protrusions such that star-like morphologies can be generated. The polar as well as the star SynMMS were characterized by both, thin and large spiking protrusions (SPs) in which several MTs converged into a single protrusion, characteristic of SIC (Fig. 5l). We also found protrusions within which bundles of strongly bent MTs formed loops, supporting a pair of membrane sheets around them, which we termed membrane sheet protrusions (MSPs, Fig. 5l). MSPs only occurred in SynMMS, and not in the system with encapsulated asters alone, indicating that basal signaling by SspB-AuroraB on the membrane is required for their formation. Furthermore, astral-MT density as determined by the relative ratio of centrosomal tubulin surface over GUV surface (Relative Centrosome Surface: RCS, methods) was related to initial morphological states, increasing from spherical to polar to star (Fig. 5m). Thus, SspB-AuroraB basal signaling together with astral-MT density determines the type and distribution of protrusions on the membrane surface.

**Light-induced de novo formation of membrane protrusions.** To investigate if de novo membrane deformations can be formed by

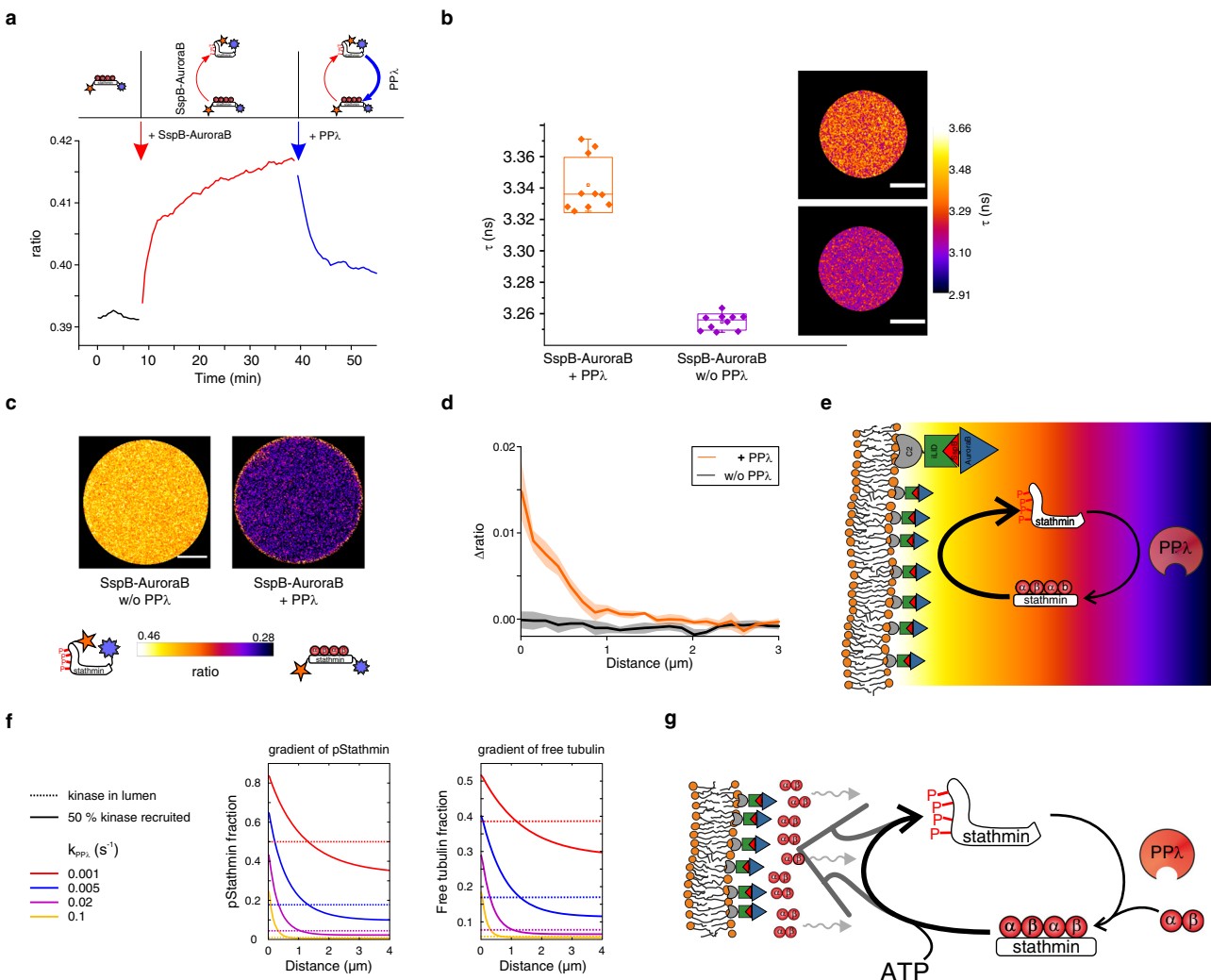

**Fig. 4 A light-inducible stathmin (de)phosphorylation cycle generates a membrane-proximal tubulin gradient. a** Stathmin phosphorylation in solution (20 μM tubulin, 2 mM ATP) measured by ratiometric imaging of FRET-sensor COPYᵒ (10 μM). Enzyme-addition indicated by arrows (red: 2 μM SspB-AuroraB; blue: 0.5 μM PPλ, final concentrations). See also Supplementary Fig. 5c. Top: cumulative scheme of establishing the (de)phosphorylation cycle. Left, COPYᵒ: stathmin conjugated to Atto532 (orange) and Atto655 (violet). Middle: phosphorylation by SspB-AuroraB (red arrow) releases αβ-tubulin heterodimers (red circles), inducing conformational change that increases FRET efficiency. Right: PPλ dephosphorylates COPYᵒ (blue arrow) closing the cycle. **b** Average donor (Atto532) fluorescence lifetimes ($\tau$) of COPYᵒ (4 μM) in GUVs (40 ± 7 μM tubulin, 2 mM ATP, 5 ± 3 10⁻¹ pmol/cm² C2-iLID, 5 ± 1 μM SspB-AuroraB) with (orange, $n = 10$, 5 ± 1 10⁻¹ μM) or without PPλ (purple, $n = 10$). Right: representative images (top: with, bottom: without PPλ, color bar: $\tau$ in ns; lower $\tau$: increased COPYᵒ phosphorylation). Boxplots: individual GUV (diamond), 25th and 75th percentile (box), 1.5 interquartile range (whiskers), median (line) and mean (square). **c** Representative COPYᵒ ratiometric fluorescence-emission images after light-induced membrane translocation of SspB-AuroraB. Maximum-intensity-projections of ratiometric 8-slice-confocal z-stacks; without (left), with PPλ (right). Color bar: fluorescence-emission ratio with corresponding open and closed COPYᵒ conformations. **d** Corresponding baseline-subtracted COPYᵒ ratiometric profiles ($\Delta$ratio; mean ± S.E.M, with PPλ orange, $n = 4$; without: black, $n = 6$; Supplementary Fig. 5h–j). **e** Scheme of spatially segregated (de)phosphorylation cycle. Translocated SspB-AuroraB (red-blue triangles) overcomes cytosolic PPλ (red circle) dephosphorylating activity. Phosphorylated stathmin diffuses away from the membrane establishing a steady-state phospho-stathmin gradient (warm-to-cold colors) by homogenous PPλ activity. **f** 1D reaction–diffusion simulations of stathmin phosphorylation (left) and free tubulin (right) yield gradients upon rebalancing (de)phosphorylation cycles by kinase translocation. Profiles plotted before ($k_{kin} = 4\ 10^{-3}\ s^{-1}$, dashed lines) and after kinase-recruitment (50% translocation, $k_{kin} = 1\ s^{-1}$, solid lines) for varying $k_{PP\lambda}$ (Methods). **g** Stathmin (de)phosphorylation cycle maintains enhanced membrane-proximal tubulin concentration by SspB-AuroraB (red-blue triangles) mediated release of tubulin (red circles) from pStathmin (dark gray arrow towards membrane), countered by tubulin diffusion (wavy arrows). Luminal PPλ dephosphorylates pStathmin to rebind tubulin heterodimers, closing the ATP-driven tubulin-deposition cycle. Scale bars: 10 μm.

light-induced SspB-AuroraB⁴⁸⁸ translocation, spherical SynMMS with low astral-MT density (RCS = 0.2–0.9) that were smaller than the GUV were globally irradiated by 488 nm light. These asters were poorly visible against the background signal of fluorescent tubulin (Fig. 6a). Upon strong light-induced translocation of SspB-AuroraB⁴⁸⁸ (Fig. 6a, c), multiple isotropically distributed MSPs appeared at the membrane, exhibiting both tubulin⁶⁴⁷ as well as

SspB-AuroraB⁴⁸⁸ fluorescence signals (Fig. 6a, b, Supplementary Video 5). The occurrence of MSPs and not SPs indicated that MT growth of single MTs or small bundles was strongly accelerated near the membrane upon strong light-induced signaling. Although the connection between astral-MTs and these MSPs was hardly visible due to the high background of free tubulin (Fig. 6a), the enhanced RCD after light-induced activation further demonstrated that these

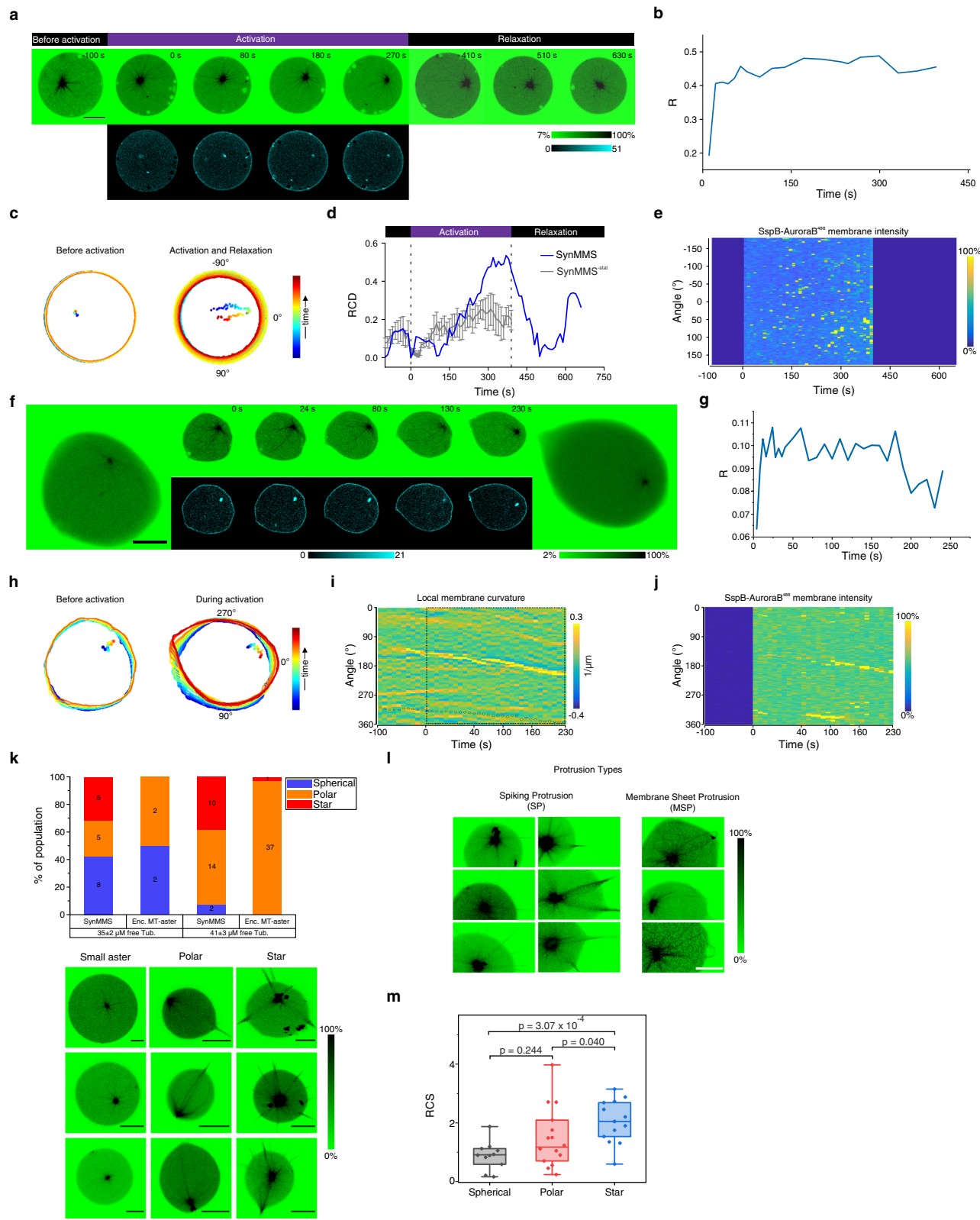

originated from astral-MTs (Fig. 6d). Quantification of membrane deformations as function of time in angular kymographs (Fig. 6e), as well as the analysis of the trajectory of these protrusions from the corresponding overlay of fluorescence and transmission images (Supplementary Fig. 8a), showed that MSPs, once formed, were stable. These protrusions also exhibited directional movement on

the membrane surface, which was coupled to centrosome movement (Fig. 6e). They strongly accumulated SspB-AuroraB[488] as apparent from angular kymographs of SspB-AuroraB[488] fluorescence and the 3D reconstruction of SynMMS after light-induced SspB-AuroraB[488] translocation (Fig. 6a, f). Furthermore, the presence of MSPs was neither detected in SynMMS-stat (Fig. 6g, h,

**Fig. 5 Light-responsive encapsulated signaling system affects morphological states by regulating astral microtubule growth. a** Selected images at indicated times of CLSM time-lapse of normalized tubulin[647] fluorescence (upper row, inverted green color bar) and corresponding SspB-AuroraB[488] translocation (lower row, cyan color bar) of a SynMMS with a rigid membrane, before, during and after global 488 nm irradiation. **b** Corresponding SspB-AuroraB[488] translocation (R) during irradiation, and **c** SynMMS-contours and centrosome positions (dots), color coded by time. **d** Relative Centrosome Displacement RCD (blue) of time-lapse represented in **a** compared to controls without stathmin (gray, mean ± S.E., 6 GUVs, N = 3; Supplementary Fig. 7a–c). **e** Angular kymograph of SspB-AuroraB[488] membrane fluorescence (color bar: normalized intensity) for SynMMS in **a**. **f** Selected images of CLSM time-lapse of normalized tubulin[647] fluorescence (upper row, inverted green color bar) and SspB-AuroraB[488] translocation (lower row, cyan color bar) of a SynMMS with a deformable membrane during global 488 nm irradiation. Maximum-intensity-projections of tubulin[647] fluorescence z-stack before (left), and after (right) 488 nm irradiation. **g** Corresponding SspB-AuroraB[488] translocation (R) and, **h** contours and centrosome positions (dots), color coded by time. **i** Angular membrane curvature kymograph overlaid with centrosome position (small circle: centered, large circle: membrane proximal). Color bar: inverse radius (1/μm) of an inscribed circle (Methods). Global 488 nm irradiation starts at t = 0 s (dashed box). **j** Corresponding angular kymograph of SspB-AuroraB[488] membrane fluorescence (color bar: normalized intensity). **k** Occurrence frequency of three stable morphologies (Spherical, Polar, Star; number of GUVs given in the bars) for SynMMS and encapsulated microtubule (MT)-asters with a deformable membrane for two estimated free-tubulin concentrations (methods). Representative morphologies shown below (inverted green color bar: normalized tubulin[647] fluorescence intensity). **l** Representative MT-induced protrusion types in SynMMS (inverted green color bar: normalized fluorescence intensity). **m** Relative centrosome surface (RCS) distributions for the three stable morphologies. Boxplots: RCS of individual GUV (diamond), 25th and 75th percentile (box), min-max (whiskers), median (line) and mean (square). p-values from two-sample Kolmogorov–Smirnov test with 95% confidence interval. Tubulin[647] fluorescence images were enhanced by histogram equalization, directional filtering and unsharp masking (methods). SspB-AuroraB[488] translocation images were corrected for luminal fluorescence contribution (methods). Scale bars: 10 μm.

Supplementary Fig. 8b, Supplementary Video 5), nor was there an increase in the RCD after light-induced translocation of SspB-AuroraB[488] (Fig. 6d), which confirms that stathmin is essential for enhancing astral-MT growth near the membrane. In SynMMS[-stat] SspB-AuroraB cluster patterns could be observed, exhibiting little or no aggregation (Supplementary Fig. 8c).

To investigate if membrane protrusions could be specifically generated in locally irradiated areas, we consecutively irradiated opposed areas of a slightly elliptical SynMMS with a sparsely populated aster (RCS = 0.90) (Fig. 6i). Local irradiation on the right side resulted in strong SspB-AuroraB[488] translocation (Fig. 6k) that triggered amplified growth of astral-MTs that bent as they deformed the membrane and caused a stable MSP (Fig. 6i, j, l) that accumulated SspB-AuroraB[488] (Fig. 6m). The resulting membrane deformation eventually enabled the capture of other dynamic MTs that evolved to a spiking protrusion after the irradiation was switched to the other side. The irradiation on the opposite side also induced the formation of MSPs that similarly enabled the subsequent capture of astral-MTs resulting in a long dynamic SP after switching the irradiated area to the lower part (Fig. 6i, j). In this third area, no stable protrusions were visible, indicating that most MTs were already redirected to the previously irradiated areas (Supplementary Video 6). In contrast, a morphologically analogous control without stathmin (SynMMS[-stat]) did not form protrusions in any of the sequentially irradiated areas (Supplementary Fig. 8d–h).

These experiments show that de novo protrusions can be formed from low density MT-asters upon strong light-induced translocation of SspB-AuroraB. However, the low density of astral-MTs together with their accelerated growth near the membrane, favors MSP over SIC-mediated SP formation.

**Reciprocal coupling of SspB-AuroraB signaling and MT self-induced capture in membrane deformations.** Since we observed that SspB-AuroraB was recruited to de novo generated MT-induced membrane deformations, we next addressed if SspB-AuroraB recruitment preferentially occurs in preformed protrusions. For this, we measured SspB-AuroraB[488] recruitment as function of light dose in both, a polar SynMMS with a strong SIC protrusion (Fig. 7a–d), as well as a star-like SynMMS with preformed MSPs and SPs (Fig. 7e–h). In addition to the light-induced recruitment of SspB-AuroraB[488] to the membrane, a stronger recruitment to the preformed protrusions could be observed at the lowest light dose in both the polar (Fig. 7a, c, d)

as well as the star SynMMS (Fig. 7e, g, h). Step-wise increasing the light dose further enhanced this recruitment in the preformed protrusions, visibly coinciding with increased astral-MT growth (Fig. 7a, e). The resulting net force on the centrosome pushed it to the periphery in both SynMMS (Fig. 7a, b, e, f), also reflected in the enhanced RCD as compared to control SynMMS[-stat] (Fig. 7i). This light-induced MT growth induced a global morphological change from a star to a spherical SynMMS shape with cortical MT bundles (Fig. 7e, f), whereas the growth of MTs in the main protrusion of the polar SynMMS did not change its morphology (Fig. 7a, b). A gradual broadening of the main protrusion due to the enhanced signaling by SspB-AuroraB[488] was observed that occurred mostly from the neck of the protrusion in which the membrane has a negative curvature (Fig. 7a) and unlimited diffusional access to the lumen. Such geometry transiently increases membrane recruitment upon irradiation[8,36]. In addition, increased membrane recruitment inside protrusions (Fig. 7e, SspB-AuroraB[488] panel) likely relies on weak interactions between MTs and SspB-AuroraB on the membrane, causing an effective slow-down of its lateral diffusion. The resulting increase in clustering combined with the limited amount of stathmin and PPλ within the protrusions, further strengthen the amplitude of the tubulin gradient (Supplementary Fig. 8i). Taken together, a positive feedback between MT-induced membrane deformation and SspB-AuroraB recruitment occurs at the membrane that reinforces and stabilizes the growth of otherwise dynamically instable MTs within protrusions (Fig. 8a).

**A generic theoretical description of the bidirectional interaction between signaling and MT-membrane systems.** To investigate which of the experimentally identified inter- and intrasystem links (Fig. 8a) dictate the formation of initial star or polar morphologies and to which extent these determine the response to morphogen signals, we numerically analyzed the dynamics of the coupled MT-membrane/SspB-AuroraB system (Eq. 24, Methods). As discussed above, the behavior of each subsystem could be equivalently described through a reaction–diffusion model, where self-amplification of local structures (SIC: Fig. 1f; CC: Fig. 3h) causes the depletion of their constituents (free MTs or AuroraB monomers) (Fig. 8a). An important intersystem link between membrane deformation and SspB-AuroraB clustering (Fig. 8a, lower arrow) was revealed by the enhanced SspB-AuroraB cluster formation in preformed

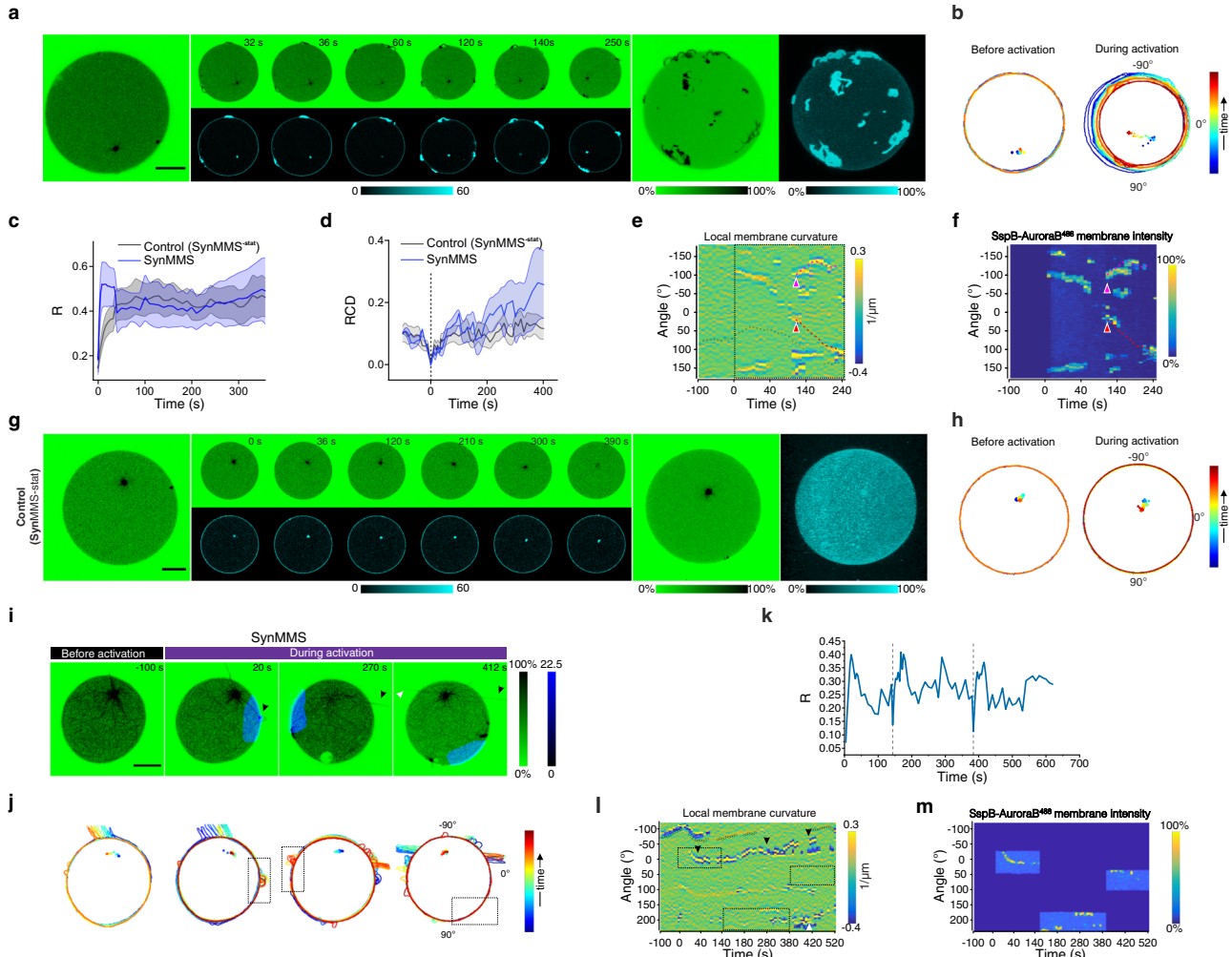

**Fig. 6 Light-induced de novo formation of astral-MT-membrane protrusions. a** Selected images at indicated times of CLSM time-lapse of normalized tubulin[647] fluorescence (upper row, inverted green color bar) and SspB-AuroraB[488] translocation (lower row, cyan color bar) of SynMMS with a small MT-aster during global 488 nm irradiation. Maximum-intensity projections of tubulin[647] and SspB-AuroraB[488] fluorescence z-stack before (left), and after (right) irradiation. **b** Corresponding contours and centrosome positions (dots), color coded by time. **c** SspB-AuroraB[488] translocation (R) for SynMMS with small asters as in **a** (blue, mean ± S.E.M, 3 GUVs, $N = 3$) compared to SynMMS[-stat] (gray, mean ± S.E.M, 13 GUVs, $N = 5$), during 488 nm irradiation. **d** Corresponding Relative Centrosome Displacement (RCD). **e** Angular membrane curvature kymograph overlaid with centrosome position (small circle: centered, large circle: membrane proximal). Color bar: inverse radius ($1/\mu m$) of an inscribed circle (Methods). Irradiation starts at $t = 0$ s (dashed box). **f** Corresponding angular kymograph of SspB-AuroraB[488] membrane fluorescence (color bar: normalized intensity). Red and magenta arrowheads: initial position of protrusions that migrate (dashed lines) over the GUV surface (Supplementary Fig. 8a). **g** Images of CLSM time-lapse as in **a** for a SynMMS[-stat] control, and **h** corresponding contours and centrosome positions, before and during activation (Supplementary Fig. 8b, c). **i** Selected images at indicated times of CLSM time-lapse of normalized tubulin[647] fluorescence (inverted green color bar) overlaid with SspB-AuroraB[488] translocation (blue color bar) before and during multiple local irradiation phases with 488 nm light of a SynMMS with a sparse aster. **j** Corresponding contours and centrosome position for each phase color coded by time (rectangles: irradiation areas), and **k** SspB-AuroraB[488] translocation (R) (dashed lines: change of irradiation phase). **l** Angular membrane curvature kymographs overlaid with centrosome position as in **e**. Local irradiation (dashed rectangles) starts at $t = 0$, arrowheads: protrusions marked in **i**. **m** Corresponding angular kymograph of SspB-AuroraB[488] membrane fluorescence (color bar: normalized intensity). See also Supplementary Fig. 8d-h. Tubulin[647] fluorescence images were enhanced by histogram equalization, directional filtering and unsharp masking (methods). SspB-AuroraB[488] translocation images were corrected for luminal fluorescence contribution (methods). Scale bars: 10 μm.

MT-induced membrane deformations (Fig. 7). The reverse intersystem link between SspB-AuroraB clusters and protrusion growth (Fig. 8a, upper arrow) is given by the tubulin gradient induced MT growth (Fig. 5a–j). These interacting subsystems thereby form a single dynamical system that can be analyzed with a reaction–diffusion model in a simplified 1D circular geometry (as in Fig. 1g). The coupled system formally breaks symmetry above a threshold of total SspB-AuroraB for a given MT amount (Supplementary Fig. 9a, b), and the symmetry-broken solution is stable in a wide parameter range (Supplementary Fig. 9c). Different initial states of the full system can be

generated, depending on which intrasystem feedbacks are dominant.

The numerical simulations demonstrated that strong SIC ($\gamma_1$) relative to CC ($\gamma_2$) generated a self-organized polar state (Fig. 8b, top, Supplementary Fig. 9d, f). In contrast, if CC is dominant over SIC, a pattern with multiple MT bundles (star-like) was stabilized (Fig. 8c, top, Supplementary Fig. 9e). This could not be obtained under weak CC, even for a strong intersystem link of MT-induced membrane deformation promoting SspB-AuroraB clustering (Supplementary Fig. 9g). This corroborates the role of basal SspB-AuroraB recruitment

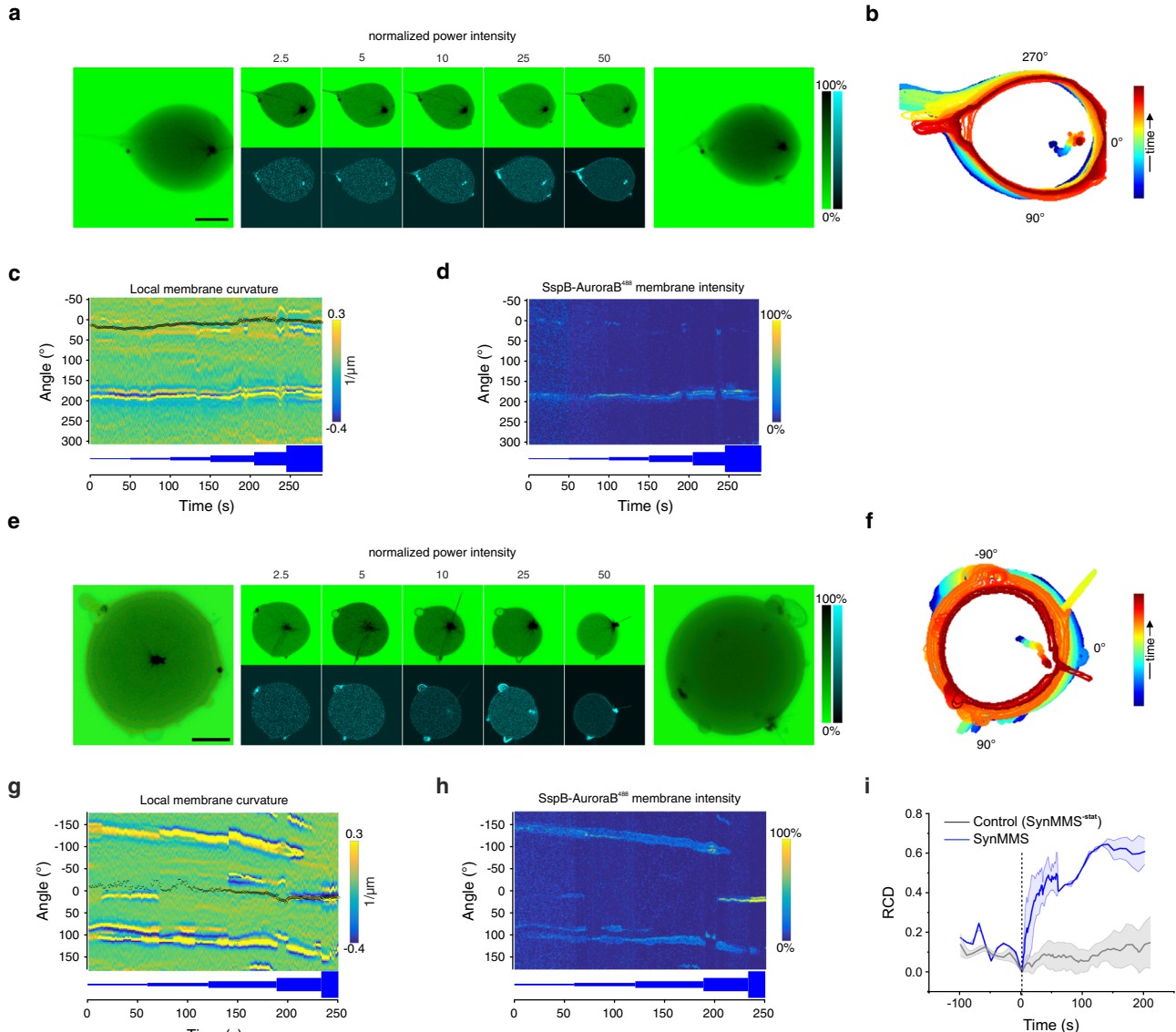

**Fig. 7 SspB-AuroraB is preferentially recruited to preformed protrusions upon light-induced translocation. a** CLSM images of normalized tubulin[647] fluorescence (upper row, inverted green color bar) and normalized SspB-AuroraB[488] translocation (lower row, cyan color bar) of a SynMMS with a strong polar protrusion during step-wise increase of global 488 nm irradiation (at indicated normalized power intensity). For each light dose an average of three consecutive frames is shown. Maximum-intensity-projections of tubulin[647] fluorescence z-stack before (left), and after (right) 488 nm irradiation. Light power normalized to 1.8 mW/cm². **b** Corresponding contours and centrosome positions (dots), color coded by time. **c** Angular membrane curvature kymograph overlaid with centrosome position (small circle: centered, large circle: membrane proximal). Color bar: inverse radius (1/μm) of an inscribed circle (Methods). **d** Corresponding angular kymograph of SspB-AuroraB[488] membrane fluorescence intensity normalized to total intensity/frame (color bar). Relative light dose is represented below the kymographs. **e–h** Same as above for a star-like SynMMS. **i** Relative Centrosome Displacement (RCD) during global irradiation of star-like SynMMS (blue, mean ± S.E.M., 3 GUVs, N = 3) compared to SynMMS[-stat] (gray, mean ± S.E.M., 4 GUVs, N = 3). Scale bars: 10 μm.

and CC in the stabilization of a star-like morphology. The initially stable protrusion distributions were maintained upon global stimulation (Fig. 8c; top, Supplementary Fig. 9f), in line with the experiments (Fig. 7).

Conversely, the responsiveness to local stimuli strongly depended on the initial stable state. When an initial polar pattern was formed through strong SIC with respect to CC, a stimulus localized away from the MT-bundle had no effect on the overall state (Fig. 8b, top). However, a stimulus proximal to the bundle caused its reorientation towards the stimulus (Fig. 8b, bottom). Contrary to this case, local stimulation of star-like initial state in which CC was dominant over SIC caused multiple MT bundles to

coalesce into the main bundle that was within or close to the stimulation region (Fig. 8c, bottom). This resulted in a transition from star to polar pattern that was oriented in the direction of the stimulus. This shows that a local stimulus enhances trapping of dynamically exchanging MTs in a pre-existing MT-bundle, thereby locally amplifying it at the expense of other MT bundles. The resulting polar pattern was robust to a second stimulus localized away from the main MT-bundle (Fig. 8c, bottom), as for an initial polar pattern (Fig. 8b, top). Thus, a high radial density of distributed MT bundles as in star-like states allows for a precise directional transition towards a localized stimulus. On the other hand, polar patterns are robust to stimulus-induced transitions

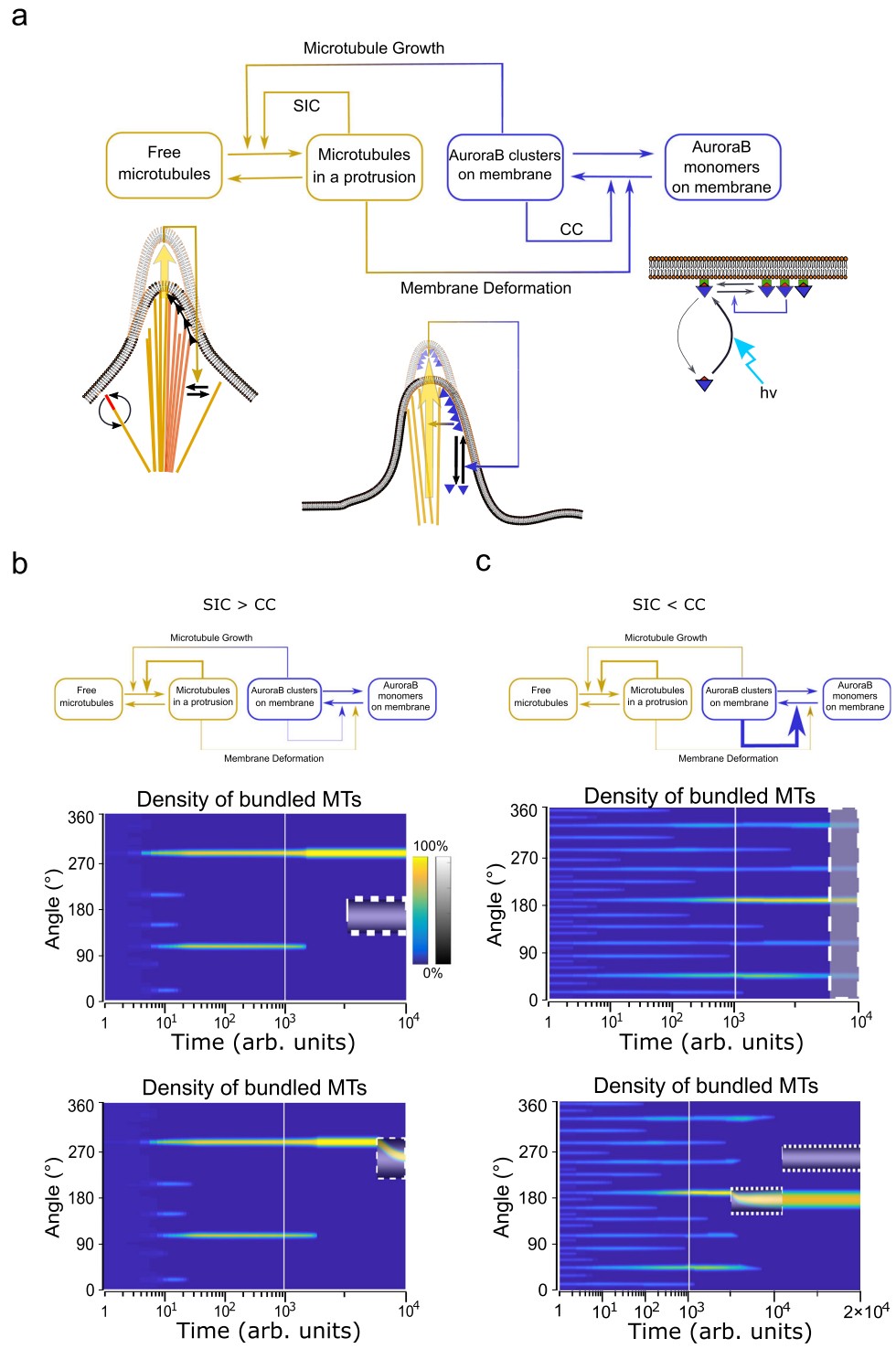

and their directional response is limited by the proximity of the stimulus.

**Light-induced morphological changes towards local signaling sources depend on initial morphological states.** In the context of our minimal theoretical abstraction, we next investigated how initial SynMMS states relate to morphogenic plasticity and directionality to external cues. We first locally stimulated a star-like SynMMS with initially axially distributed small dynamic SIC protrusions (Fig. 9a, left panel), indicative of CC dominating over SIC. This star-like SynMMS exhibited basal levels of SspB-

AuroraB[488] on the membrane that was strongly enhanced upon local irradiation (Fig. 9b), suggesting the possibility for strong CC-induced signal amplification. Upon local continuous irradiation of this SynMMS in the upper right quadrant (Fig. 9a), dynamically maintained protrusions converged into this area (Fig. 9a, c, d). As observation of tubulin[647] fluorescence was restricted to the confocal plane, out-of-focus protrusions highlighted by membrane curvature could be tracked in transmission images (Supplementary Video 7). During this process, the dynamics of MTs that transiently deformed the membrane increased in this area, showing the rapid exchange of MTs

**Fig. 8 In silico dynamics of the coupled MT-membrane/signaling system. a** Schematic representation of the coupled microtubule (MT)-membrane/ signaling system incorporating experimentally identified intra- and intersystem causal links. Intrasystems links; SIC self-induced capture, CC cooperative clustering. Intersystem links; top arrow: AuroraB clustering on the membrane cause MT growth, which promotes protrusions by capture of free MTs, bottom arrow: MT-induced membrane deformation enhances SspB-AuroraB clustering. Lower pictograms: intrasystem links as in Fig. 1f and Fig. 3h. Middle, intersystem links: MT growth deforms the membrane (large yellow arrow) that increases SspB-AuroraB (triangles) clusters in the protrusion (yellow-blue feedback arrow onto black arrow) that in turn enhances MT growth (horizontal yellow arrow) and respective MT-capture in a positive feedback loop. **b** Top; SIC > CC: self-amplified MT-capture ($\gamma_1 = 1$) dominates over SspB-AuroraB cooperative clustering ($\gamma_2 = 0.5$). Middle: density of bundled MTs kymograph showing that self-organized polar state is robust to a local stimulus distant from the protrusion (dashed box). Bottom: The MT-bundle can be reoriented in the direction of a proximal stimulus (dashed box). **c** Top; SIC < CC: SspB-AuroraB cooperative clustering ($\gamma_2 = 5$) dominates over self-amplified MT-capture ($\gamma_1 = 1$). Middle: kymograph of density of bundled MTs showing the evolution to a star-like state that is stable upon global stimulation (dashed box). Bottom: local stimulation (dashed box) causes a transition to a polar pattern that is robust to a second distal stimulus. The color bar in **b** is representative for all kymographs and indicates normalized density of bundled MTs per angular bin, gray-scale bar: normalized stimulation strength. The combined logarithmic/linear timescales are implemented to depict the initial pattern starting from random initial conditions and the transition to a stable pattern.

between instable protrusions that were assimilated in a stable main protrusion. These astral-MT-induced protrusions accumulated SspB-AuroraB[488] (Fig. 9e), demonstrating the feedback between signaling and MT growth. The resulting amplification stabilized the main polar protrusions and led to an explosive sprouting of additional microprotrusions (Fig. 9a, Supplementary Video 7). This resulted in an overall morphing process from star to polar morphology towards the irradiated region (Fig. 9a: left and right 3D projections, Fig. 9c), demonstrating that star-like SynMMS with strong CC relative to SIC can globally reorganize their morphology in the direction of a localized stimulus. This star-to-polar morphological transition is consistent with the numerical analysis for SIC < CC (Fig. 8c).

We compared these results with the morphogenic plasticity of a star-like SynMMS with low SspB-AuroraB[488] translocation (Fig. 9f, g) and therefore a weak potential for CC-induced signal amplification. In this system, the initially radially distributed dynamic SPs reoriented towards the shallow gradient of SspB-AuroraB[488] translocation emanating from the lower irradiated quadrant (Fig. 9h, i). This eventually resulted in an anisotropic redistribution of the initially isotropically distributed fluctuating SPs towards the light-induced signaling gradient (Fig. 9f, left and right 3D projections). Thus, for low SspB-AuroraB[488] translocation, the feedback between localized signaling and SIC cannot be effectively established at the membrane. As a result, protrusions cannot be stabilized, but rather fluctuating SPs are only redistributed towards the signal source.

We next investigated the morphogenic plasticity of a polar SynMMS with a stable dominant protrusion and dynamic ones at the opposite pole, indicative of SIC dominating over CC (Fig. 9j). Local irradiation in the left flanking region to the main polar protrusions resulted in SspB-AuroraB[488] translocation (Fig. 9j, k) and net MT growth towards the irradiated area, thereby rotating the SynMMS in the direction of the light signal (Fig. 9l). Light-induced translocation (Fig. 9j, k) in the opposite flanking area generated visible MSPs (Fig. 9j, m) enriched in SspB-AuroraB[488] (Fig. 9n), indicating feedback between signaling and MT-induced membrane deformation. However, light-induced MT growth led to sliding of the MSPs towards the main polar protrusion, accompanied by the loss of a dynamic protrusion at the opposite pole. Subsequent global irradiation showed that these SspB-AuroraB[488]-enriched MSPs indeed accumulated into the main polar protrusion (Fig. 9m, Supplementary Video 8). This shows that irradiation of a polar SynMMS away from the main protrusion results in the redistribution and capture of astral-MTs into the main protrusion, stabilizing its polar morphology. Similarly, local irradiation proximal to the protrusion of a polar SynMMS caused it to reorient into the irradiated area without overall morphological reorganization (Supplementary Fig. 10).

Equivalent reorientation of the main MT-bundle in the signal direction was also obtained numerically, for SIC > CC (Fig. 8b). These results therefore demonstrate that polar morphologies are robust to stimulus-induced morphological transitions and their directional response is limited by the proximity of the stimulus.

## Discussion

To distill basic principles of morphogen-guided cell morphogenesis, we reconstituted dynamic microtubule asters inside GUVs together with a light-responsive signaling system that mimics morphogen signal transduction to the cytoskeleton in cells. The reconstituted system incorporated regulatable (de) phosphorylation reaction cycles, which are a fundamental feature of cells to maintain their responsiveness to extracellular signals. In the absence of stimulus, both stathmin kinases and phosphatases are active and reside in the cytoplasm. In this state, phosphatase activity is higher than that of the kinases, maintaining a low steady state of phosphorylated stathmin, and thus suppressed MT growth. Morphogen-induced activation of Rac in cells via activated guanine nucleotide exchange factors (GEFs) recruits stathmin kinases to the stimulated membrane area. This dimensionality reduction (3D-to-2D) locally enhances kinase concentration relative to cytoplasmic phosphatases that in turn rebalances the cycle towards phosphorylation of stathmin in the membrane-proximal region. In analogy, we demonstrated how light-exposed C2-iLID on the membrane recruits the stathmin-kinase SspB-AuroraB to locally enhance its concentration relative to cytoplasmic $\lambda$−phosphatase, rebalancing the cycle towards phosphorylation of stathmin in the membrane-proximal region. In both systems, diffusion of pStathmin away from the membrane exposed to morphogen/light towards the lumen with high relative phosphatase activity then establishes a steady-state pStathmin gradient. However, it is the concurrent release of tubulin from inactivated pStathmin and the thereby generated higher tubulin concentration in the membrane vicinity that promotes directional growth of the MTs that deform the membrane (Fig. 10).

On the other hand, MT growth can indirectly enhance signaling via membrane deformation, which establishes bidirectional causality between SspB-AuroraB signaling and the MT-cytoskeletal system. Each of these subsystems already has self-organizing pattern-forming properties based on the principle of local matter coalescence, where self-amplification depletes the substrate from the surrounding: cooperative clustering (CC) for SspB-AuroraB and self-induced capture (SIC) for MTs. The process of generating protrusion morphologies through indirect interactions mediated by the deformable membrane, is analogous to Grassé's stigmergic concept of matter coalescence in regular patterns through indirect communication of social insects[8,28–30]. We found that the bidirectional coupling of the subsystems not

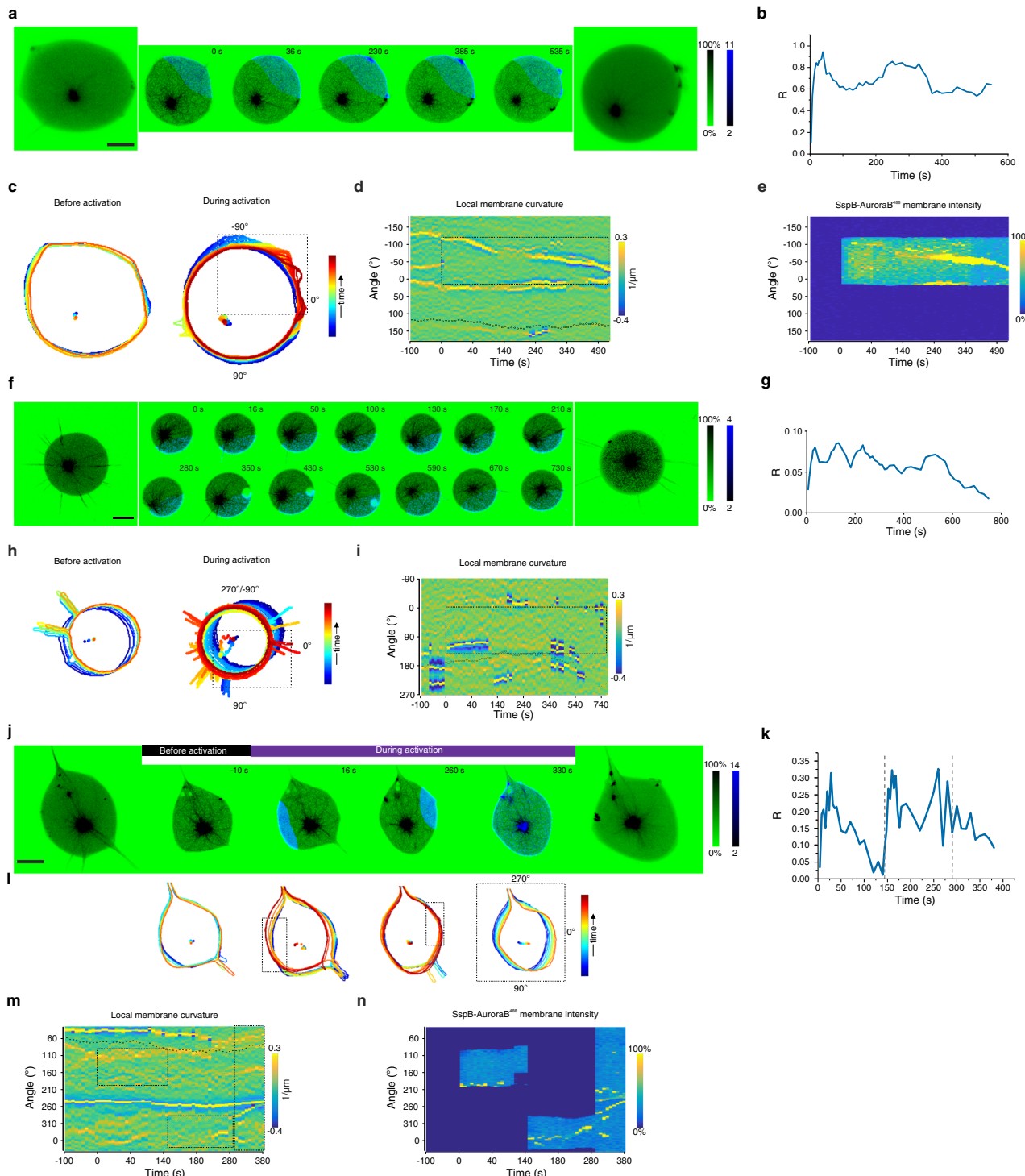

only defines stable initial SynMMS morphologies but also their global morphogenic response to local light stimuli. Plasticity in this morphing process and directionality towards an external cue is dictated by the initial morphological state that for a given MT density arises from the balance between SIC and CC. When CC dominates over SIC, collective behavior leads to star-like SynMMS with multiple small protrusions that convey the highest plasticity and directional response towards external cues (Fig. 10, right pictograms). When SIC, however, dominates, initial states with a large polar protrusion are generated that convey

morphological robustness in response to external signaling cues. Consequently, the transition from star into a polar shape by external cues is irreversible, which leads to the question how cells can switch from robust polar to plastic star-like states. Although many layers of biochemical regulation contribute to both morphological plasticity and stabilization in cells[37,38], the interplay between actin and MTs may be of fundamental importance, where actin would generate rapid and local reversible morphological dynamics at the cell periphery, while the MT-cytoskeleton would guide and stabilize the global shape. This could be further

**Fig. 9 Morphological responses towards local light cues depends on initial states. a** Selected images at indicated times of CLSM time-lapse of tubulin647 fluorescence (inverted green color bar: normalized intensity) overlaid with SspB-AuroraB488 translocation (blue color bar) during local 488 nm irradiation of a star-like SynMMS with small protrusions. Maximum-intensity-projections of tubulin647 fluorescence z-stack before (left), and after (right) 488 nm irradiation. **b** Corresponding SspB-AuroraB488 translocation (R) during irradiation, and **c** contours and centrosome positions (dots), color coded by time. **d** Angular membrane curvature kymograph overlaid with centrosome position (small circle: centered, large circle: membrane proximal). Color bar: inverse radius (1/μm) of an inscribed circle (Methods). Local irradiation (dashed rectangle) starts at t = 0 s. **e** Corresponding angular kymograph of SspB-AuroraB488 membrane fluorescence intensity (color bar: normalized intensity). **f** CLSM time-lapse of tubulin647 overlaid with SspB-AuroraB488 as in **a** during local 488 nm irradiation of a star-like SynMMS with low SspB-AuroraB488 translocation. Maximum-intensity-projections of tubulin647 fluorescence z-stack before (left), and after (right) 488 nm irradiation. **g** Corresponding SspB-AuroraB488 translocation (R) during irradiation, and **h** contours and centrosome positions (dots), color coded by time. **i** Angular membrane curvature kymograph as in **d**. Local irradiation (dashed rectangle) starts at t = 0 s. **j** Selected images at indicated times of CLSM time-lapse of normalized tubulin647 fluorescence overlaid with SspB-AuroraB488 translocation before and during multiple local 488 nm irradiation phases followed by global irradiation of a polar SynMMS as in **a**. Maximum-intensity projections of tubulin647 fluorescence z-stack before (left), and after (right) 488 nm irradiation. **k** SspB-AuroraB488 translocation (R) (dashed lines: change of irradiation phase). **l** Contours and centrosome position for each phase color coded by time (rectangles: irradiation areas). **m** Angular membrane curvature kymograph as in **d**. Local irradiation (dashed rectangles) starts at t = 0 s. **n** Corresponding angular kymograph of SspB-AuroraB488 membrane fluorescence (color bar: normalized intensity). Tubulin647 fluorescence images were enhanced by histogram equalization, directional filtering and unsharp masking (methods). SspB-AuroraB488 translocation images were corrected for luminal fluorescence contribution (methods). Scale bars: 10 μm.

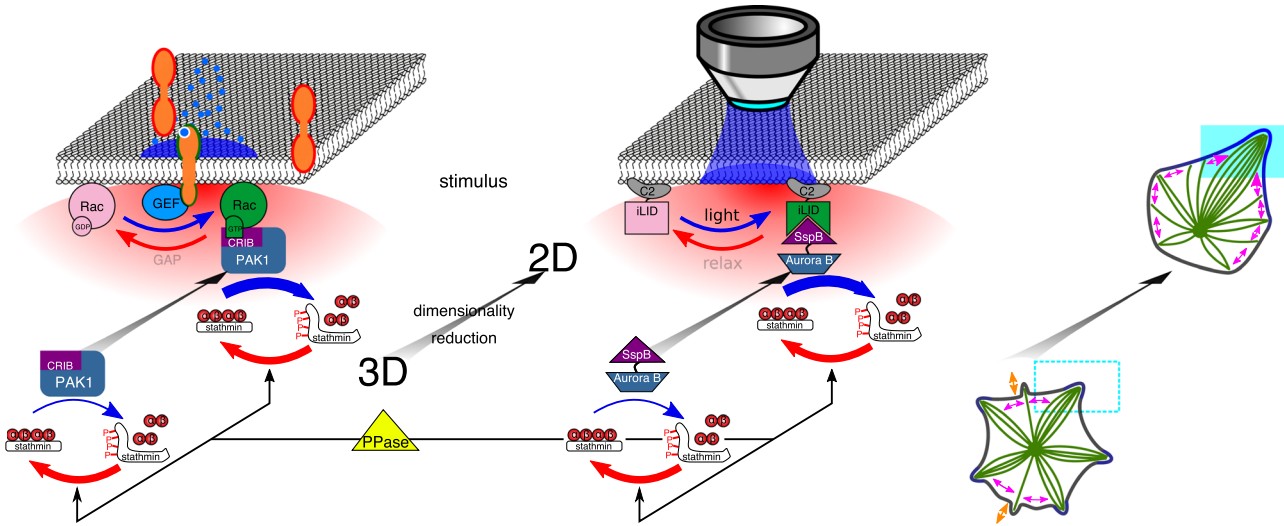

**Fig. 10 Biomimetic principles of SynMMS.** Scheme of natural (left) and synthetic (right) signal transduction to MT-cytoskeleton. Lower cycles: before stimulus, cytosolic kinases (natural: PAK1, synthetic: SspB-AuroraB) phosphorylate stathmin (blue curved arrow) in a cyclic reaction countered (red curved arrow) by stronger activity of cytosolic phosphatases (PPase, synthetic: PPλ). This maintains a low steady state of phosphorylated stathmin (P-), which suppresses MT growth by sequestering tubulin (red circles on stathmin). Lower right SynMMS pictogram: before stimulus (dashed box), dynamic instability of MTs (double-headed orange arrows) enables self-induced capture to generate MT bundles (green lines) that together with basal signaling (blue) stabilize protrusions. The exchange of MTs between protrusions (double-headed magenta arrows) results in dynamically maintained self-organized morphologies (here: star-like). Upper cycles: during stimulus, cells (left) respond to extracellular morphogen (blue dots) by receptor (orange) activity on the locally stimulated membrane (blue area) that recruits and activates guanine nucleotide exchange factors (GEFs), which activates the GTPase Rac by nucleotide exchange of GDP to GTP (pink-to-green) and is reversed by GTPase activating protein (GAP) activity. The activated GTPase recruits the cytosolic stathmin-kinase PAK1 via its CRIB-domain to the stimulated membrane. Analogously, in SynMMS (right) a localized light stimulus (blue area at the membrane) activates C2-iLID (pink-to-green, membrane-associated via C2-domain), recruiting SspB-AuroraB stathmin-kinase via its SspB domain to the stimulated membrane. In both systems, recruitment results in a dimensionality reduction from 3D (cytosolic) to 2D (membrane) that increases the local concentration of the kinase on the stimulated part of the membrane. This increased local kinase activity shifts the cyclic reaction towards phosphorylated stathmin that thereby enhances its continuous tubulin release (red circles) in the membrane-proximal region (red haze). Upper right SynMMS pictogram: during stimulus (solid cyan box), the signaling induced tubulin gradient is preferentially amplified in preformed protrusions, generating a membrane-proximal environment in which MT growth and further capture is promoted (large magenta arrowhead) at the expense of other protrusions that exchange MTs (magenta double-headed arrows). This results in a global reorganization of the cytoskeleton towards the external stimulus (here: polar).

elucidated using synthetic systems that build upon SynMMS from the progress made in reconstituting actin networks on and within synthetic membranes[39,40].

In spite of good encapsulation reproducibility for most components, two major factors currently limited the controlled generation of initial morphologies. These were variable encapsulation of C2-iLID, as well as variable nucleation of MTs on centrosome

clusters. The former determines both the basal as well as maximal level of signaling and thereby CC, whereas the latter determines the astral-MT density. Purified centrosomes could be replaced by crosslinking motors in combination with nucleating factors[41], whereas methods to incorporate protein after GUV formation, such as picoinjection[42] or electroporation[43] could improve incorporation of lipophilic proteins. However, the phenotypic

variability of SynMMS was used in an informative way because experimentally identified causalities allowed numerical investigation of the relation between initial states and response properties that were verified experimentally.

The short life span of SynMMS is constrained by the encapsulated fuel store (ATP, GTP). By introducing an artificial photosynthetic membrane[44], light energy could be converted into ATP or even GTP chemical potential[45] that could maintain the nonequilibrium state over prolonged periods. Finally, recent progress in reconstituting self-replicating DNA by its encoded proteins in liposomes[46], bacterial cell division[47] and lipid metabolism[48] suggests that self-replicating properties could be conferred to synthetic systems such as SynMMS in the future.

At the higher scale, steps have been made to generate synthetic tissues where porous proto-cells communicate by diffusive protein signals to form larger tissue-mimicking arrays[49]. The communication in these synthetic systems was based on distinct signal emitter and receiver synthetic cells. The intrinsic self-organizing morphological dynamics of SynMMS has the potential to establish recursive communication among them[50], enabling the investigation of basic principles of self-organized tissue formation.

## Methods

**Preparation of recombinant proteins**. Amino acid sequences of all used constructs are presented in Supplementary Table 2. All proteins were expressed in *E. coli* BL21 DE3 RIL (CN: 230245, Agilent, Santa Clara, CA, United States) over night at 18 °C after induction with 0.2 mM IPTG. SspB-AuroraB was bicistronically expressed with the INCENP in-box from a modified pGEX6P-2rbs plasmid, which was a gift from A. Musacchio. All other proteins were expressed from a pET24 plasmid as fusions to an N-terminal His$_{10}$-tag followed by a TEV protease site. 4–6 L of LB medium were used to express Stathmin, iLID, and $\lambda$-phosphatase ($\lambda$-PPase), whereas 10–12 L of TB medium were used for SspB-AuroraB. Cells were lysed by two passes through an Emulsiflex C5 (Avestin, Mannheim, Germany), and the lysate was cleared by 45–60 min centrifugation at 48,000 × *g* in an A27-8 × 50 rotor (Sorvall/Thermo Fisher Scientific, Dreieich, Germany).

*Stathmin and iLID*. Gly-Stathmin-Cys, Gly-iLID-tRac1, or Gly-C2-iLID were purified via Ni-NTA Superflow (Qiagen, Hilden, Germany) in 50 mM NaP$_i$ pH 8, 300 mM NaCl, 2 mM MgCl$_2$, 0.1 mM β-mercaptoethanol, 10 mM imidazole. The column was washed with 60 mM imidazole, and the protein eluted by a step to 500 mM imidazole. The eluate was supplemented with 2–4 mg TEV protease and dialyzed over night against 2 L of 50 mM Tris pH 8, 100 mM NaCl, 1 mM EDTA, 0.1 mM βmercaptoethanol. The cleaved protein was adjusted to 10 mM imidazole (60 mM for Gly-C2-iLID), and passed again through the Ni-NTA column, followed by gel filtration on a HiLoad 26/600 Superdex 75 pg column (GE Healthcare, Solingen, Germany) in 50 mM Hepes pH 7.5, 200 mM NaCl, 2 mM MgCl$_2$, 2 mM DTT. For iLID constructs, the sample was adjusted to a saturated concentration of flavin mononucleotide prior to gel filtration. Gly-iLID-tRac1 was subsequently geranylgeranylated[51]. Three milligrams of the target protein and a 1:10 molar ratio of GGTase was mixed with one vial of Geranylgeranyl pyrophospate (~460nmol, Sigma–Aldrich/Merck) to a total volume of 3 mL in prenylation buffer (50 mM Hepes pH 7.5, 50 mM NaCl, 2 mM MgCl2, 10% Glycerin, 10 μM ZnSO4, 2% CHAPS, and 2 mM DTT) and incubated at 4 °C over night. The geranylated protein was further purified by gel filtration on a HiLoad 16/600 Superdex 75 pg column (GE Healthcare, Solingen, Germany) in prenylation buffer with 0.5% CHAPS followed by detergents removal by a IllustraTM NAPTM-5 column (GE Healthcare, Solingen, Germany) and a Pierce® Detergent Removel Spin Column (Thermo Fisher Scientific, Munich, Germany). We refer to geranylgeranylated iLID as iLID_G.

*$\lambda$-phosphatase*. The protein was purified via cobalt-loaded HiTrap Chelating HP columns (GE Healthcare) in 50 mM NaP$_i$ pH 8, 500 mM NaCl, 2 mM MgCl$_2$, 5% v/v glycerol, 0.1 mM β-mercaptoethanol, 10 mM imidazole. The washing/elution steps were 60, 200, 500 mM imidazole. The eluted protein was quickly supplemented with 10 mM DTT and digested with TEV protease as above, while dialyzing against 2 L of 50 mM Tris pH 8, 500 mM NaCl, 5% v/v glycerol, 2 mM MgCl$_2$, 10 mM DTT. The cleaved protein was buffer exchanged back into the sodium phosphate buffer via a HiPrep Desalting 26/10 column (GE Healthcare), adjusted to 10 mM imidazole, and passed again over the chelating column. Finally, the protein was gel filtered on a HiLoad 26/600 Superdex 75 pg column (GE Healthcare) in 50 mM Hepes pH 7.5, 500 mM NaCl, 2 mM MgCl$_2$, 5 mM DTT, 10% v/v glycerol.

*SspB-AuroraB*. Gly-SspB-AuroraB(45-344)/INCENP(834-902) was purified via Glutathione Sepharose 4 Fast Flow (GE Healthcare) in 50 mM Hepes pH 7.3, 500 mM NaCl, 2 mM MgCl$_2$, 5 mM DTT, 5% v/v glycerol. The column with bound protein was washed with buffer containing 1 mM ATP, then the protein was eluted with buffer containing 20 mM glutathione and additional DTT up to 10 mM. The protein was cleaved over night with TEV protease while remaining in the elution buffer. The cleaved protein was buffer exchanged via a HiPrep Desalting 26/10 column (GE Healthcare) to remove glutathione, and passed again through the Glutathione Sepharose column. Gel filtration was carried out on a HiLoad 26/600 Superdex 200 pg column (GE Healthcare) in 50 mM Hepes pH 7.3, 500 mM NaCl, 2 mM MgCl$_2$, 5 mM DTT avoiding concentrating the protein before gel filtration. Finally, the protein was concentrated in an Amicon Stirred Cell (Merck Millipore, Darmstadt, Germany), adjusted to 20% glycerol and hard spun at 400,000 × *g* prior to shock freezing.

**Preparation of tubulin and centrosomes**. Tubulin purification was performed according to a standard method[52]. Pig brains were lysed in depolymerization Buffer (50 mM MES pH 6.6, 1 mM CaCl$_2$) in the ratio of 1.5 buffer/brain (l/kg). The lysate was centrifuged for 60 min (4 °C) at 28,000 × *g* in SLA-1500 (Sorvall/Thermo Fisher Scientific). The supernatant was collected, supplemented with: PIPES, ATP and GTP to the final concentration of 333, 1.5, and 0.50 mM, respectively, and incubated for 60 min at 37 °C. The polymerized supernatant was centrifuged for 30 min (37 °C) at 200,000 × *g* in Type 45 TI (Beckman Coulter, Krefeld, Germany). The pellets were resuspended in depolymerization Buffer on ice. The depolymerized solution was centrifuged for 20 min (4 °C) at 140,000 × *g* in Type 45 TI (Beckman Coulter, Krefeld, Germany). After repeating polymerization and depolymerization steps, purified tubulin concentration was adjusted to the of 200 μM with BRB80 (80 mM PIPES pH 6.8, 1 mM MgCl2, 1 mM EGTA) and stored at −150 °C after flash freezing in liquid nitrogen.

Tubulin was labeled with NHS-Alexa568, −488 or −647 (Thermo Fisher Scientific) or EZ-link NHS-biotin (Thermo Fisher Scientific) according to a standard procedure[53]. Typically, 50 mg of tubulin were labeled with a 7.5-fold molar excess of dye assuming 70% tubulin recovery after the initial polymerization step. Following deviations from the published procedure were made: the dye was added in two steps, each followed by a 15 min incubation, totaling 40 μl DMSO in 1 ml buffer. The reaction was not specifically stopped, and all ultracentrifugation steps were performed at an increased speed of 400,000 × *g* in an MLA-80 rotor (Beckman Coulter, Krefeld, Germany). Labeled tubulin was adjusted to 200 μM with BRB80 and stored at −150 °C.

Centrosomes were isolated from KE-37 (DSMZ no.: ACC 46, Human T cell leukemia, References: 14473,15220) cells following a standard method[54]. 2.5-3 liters of $1.5 \times 10^6$ cell/mL were treated with 0.33 μM nocodazole and incubated at 37 °C for 60 min. Cells were centrifuged for 5 min (4 °C) at 800 × *g* in F10-6x500y (Sorvall/Thermo Fisher Scientific) and resuspended in half of the starting volume in TBS (20 mM Tris pH 7.4, 150 mM Nacl). This was followed by centrifugation for 5 min (4 °C) at 800 g in F10-6x500y (Sorvall/Thermo Fisher Scientific) and resuspended in half of the starting volume of this step in TBS 1/10–8Suc. (2 mM Tris pH 7.4, 15 mM Nacl, 8% w/v Sucrose). After centrifugation for 5 min (4 °C) at 800 × *g* in SLA-1500 (Sorvall/Thermo Fisher Scientific), cells were resuspended in TBS 1/10–8Suc. and were lysed (cell density was adjusted to $10^7$cells/mL) in Lysis Buffer (1 mM HEPES pH 7.2, 0.5% vol. IGEPAL, 0.5 mM Mgcl$_2$, 0.1% vol. β-mercaptoethanol, 1 μg/mL Leupeptin, 1 μg/mL pepstatin, 1 μg/mL aprotinin, 1 mM PMSF). The lysate was centrifuged for 10 min (4 °C) at 2500 × *g* in SLA-1500 (Sorvall/Thermo Fisher Scientific). The supernatant was collected, supplemented with 10 mM PIPES and 1 μg/mL DNase I and incubated at 4 °C for 30 min. Centrosomes were centrifuged onto 60% sucrose Cushion (10 mM PIPES pH 7.2, 60% w/w sucrose, 0.1% vol. Triton X-100, 0.1% vol. β-mercaptoethanol) for 30 min (4 °C) at 10,000 × *g* in swing out P28S (Eppendorf Himac Technologies). They were further centrifuged through a discontinuous 70%, 50%, and 40% sucrose Cushions for 60 min (4 °C) at 40,000 × *g* in swing out P28S (Eppendorf Himac Technologies). The centrosomes were collected from the bottom of the tube in 0.5 mL fractions and stored in −80 °C after flash freezing in nitrogen liquid.

**Protein labeling**. Proteins were specifically labeled by sortagging[55]. The LPETGG peptide was conjugated to NHS-Alexa488, -Alexa647, -Atto532, or -Atto542 over night at 30 °C (20 mM peptide, 40 mM dye (Thermo Fisher Scientific for Alexa dyes, Atto-tec (Siegen, Germany) for Atto532 dyes) in DMSO), the reaction stopped by 100 mM Tris pH 8. Typically, 1–3 mg protein (ideally around 300 μM final concentration in the reaction), were mixed with a 4-fold excess of conjugated peptide, ca. 100 μM Sortase A (from *S. aureus*, gift from P. Bieling), and 6 mM CaCl$_2$. For iLID constructs, the sample was additionally adjusted to a saturated concentration of flavin mononucleotide. The reaction was allowed to proceed over night at 18 °C or 4 °C (for $\lambda$-PPase and SspB-AuroraB), and the mixture was separated by gel filtration on a Superdex 75 10/300 GL column (GE Healthcare) in appropriate protein gel filtration buffer.

**Protein encapsulation in GUVs by cDICE**. Encapsulation of proteins in GUVs was achieved by continuous Droplet Interface Crossing Encapsulation following the original method[19] with relevant parameters noted below. Empty GUVs used for experiments in the outside configuration were also produced by this method.

0.36 mM lipids were prepared in mineral oil (M3516, Sigma–Aldrich/Merck, Darmstadt, Germany). The mixture consisted of eggPC (Sigma–Aldrich/Merck) with 15 mol% DOPS and 0.05 mol% DOPE-biotin (both from Avanti Polar Lipids, Alabaster, U.S.A.). Additionally, 0.05 mol% Lissamine-RhodamineB-PE was added for experiments to analyze Alexa488-SspB-AuroraB or Alexa647-SspB-PPλ patterning on GUV membranes. Glass capillaries were typically 10–15 μm wide, and pressure was applied using an MFCS-EZ control system (Fluigent, Jena, Germany). cDICE chambers were 3.5 cm wide and were rotated at 1500–1800 rpm. Eight-well LabTek chambers (Thermo Fisher Scientific) or 18-well μ-slides (ibidi, Martinsried, Germany) were coated with biotin and functionalized with streptavidin for GUV immobilization by interaction with DOPE-biotin.

The base cDICE buffer contained 80 mM PIPES pH 6.8, 75 mM KCl, 2 mM MgCl$_2$, 1 mM EGTA, 1 mM trolox, 0.1 mg/ml β-casein. Additionally, the inner buffer contained 300 mM sucrose and 80 mM glucose, while the outer buffer contained 380 mM glucose. Whenever an osmotic deflation of GUVs to decrease membrane tension was desired, the sucrose concentration inside was lowered by 60 mM and the outside glucose concentration was increased by 160 mM (i.e. Figure 1a, d; Fig. 5f; Figs. 6, 7, 9; Supplementary Fig. 1f, Supplementary Fig. 7d, f; Supplementary Fig. 8a, c, Supplementary Fig. 10).

The encapsulation efficiency of tubulin and stathmin was determined by a reference concentration outside of GUVs (Supplementary Fig. 2c, d), and the used protein concentration in cDICE was adjusted to yield desired final concentrations, as indicated in respective experiments. For encapsulation of centrosomes, the samples typically contained 10 vol% of a purified centrosome fraction.

Prior to cDICE, the samples were centrifuged for 15 min at $434{,}000 \times g$ in a TLA-100 rotor (Beckman Coulter). The centrosomes were never centrifuged, while the tubulin stock was centrifuged separately for 30 min at $200{,}000 \times g$ in an RP80-AT2 rotor (Sorvall/Thermo Fisher Scientific).

For encapsulated asters alone, the buffer contained 80 mM PIPES pH 6.8, 75 mM KCl, 2 mM MgCl$_2$, 1 mM EGTA, 1 mM trolox, 0.1 mg/ml β-casein, 2 mM GTP. In all experiments involving SynMMS (controls as well as full system) the buffer contained 80 mM PIPES pH 6.8, 75 mM KCl, 4 mM MgCl$_2$, 0.8 mM MnCl$_2$, 1 mM trolox, 0.1 mg/ml β-casein, 2 mM GTP, 2 mM ATP. EGTA is consistently replaced with MnCl$_2$ because PPλ is a metal dependent phosphatase, and MgCl$_2$ was increased to ensure nucleotide binding. In stathmin phosphorylation gradient experiments that require the presence of soluble tubulin, only 60 μM GTP was included to avoid polymerization and prevent tubulin degradation.

### Imaging of MT-asters and morphological states in GUVs.

To produce MT-asters in GUVs, soluble tubulin and centrosomes were encapsulated as described in "Protein encapsulation in GUVs by cDICE". Stathmin was included as indicated for individual experiments. The following protein concentrations were used in the full reconstituted SynMMS: 40 or 44 μM (final) tubulin (10% Alexa647-tubulin), 4 μM (final) stathmin, 5 μM Gly-C2-iLID, 4.9 μM (final) Alexa488-SspB-AuroraB/INCENP, 0.5 μM (final) PPλ. Alexa568-tubulin was used for experiments without the signaling system at indicated concentrations. Alexa488-tubulin was used in experiments to establish dependency between membrane curvature and Alexa647-SspB-PPλ membrane recruitment.

Imaging was done on a Leica SP8 confocal microscope (Leica Microsystems, Wetzlar, Germany) equipped with Leica HyD hybrid detectors at 33 °C in a controlled environment chamber (Life Imaging Services, Basel, Switzerland) using an HC PL APO 63x/1.2NA motCORR CS2 water objective (Leica Microsystems) using the Leica LasX software. The elevated temperature promoted tubulin polymerization, the asters reaching steady-state size in ca. 7 min, after which single-plane images or z-stacks (1 or 0.5 μm spacing) were taken. For temperature ramp experiments, GUV-containing LabTek chambers were incubated at 4 °C for 10 min in the precooled Stable Z Lab-Tek Stage (Bioptechs, Butler, U.S.A.). Then, the stage was moved onto the microscope equilibrated at room temperature and reached RT (~21 °C) in 10 min. The measurements indicated as "cold" were done under this condition. After that the Stable Z Lab-Tek Stage controller was used to heat the sample to 34 °C within 20 min and images were taken every minute.

Alexa568 or Alexa488 were excited with a 470–670 nm white light laser (white light laser Kit WLL2, NKT Photonics, Köln, Germany) at 578 nm (270–600 mW/cm$^2$) or 488 nm (220–340 mW/cm$^2$), respectively. Detection of fluorescence emission was done in standard or counting mode, restricted with an Acousto-Optical Beam Splitter (AOBS) to 588–640 nm or 498–540 nm, respectively. Images were taken with 2–4× line averaging at 400 or 200 Hz scanning speed. The pinhole was set to 1 Airy unit.

In SynMMS, Alexa647 was excited at 650 nm (200–700 mW/cm$^2$) at 1000 Hz scanning speed to reduce photodamage. Detection of fluorescence emission was done in counting mode, restricted to 650–750 nm and further restricted by LightGate time-gating to minimize signal from laser reflection and background. Activation of iLID and imaging of Alexa488-SspB-AuroraB was done with 488 nm (220 mW/cm$^2$) excitation from a white light laser source, restricting detection to 498–580 nm and by LightGate. Time series images and stacks (1 μm spacing) were taken with 8x line accumulation in line sequential scanning mode. For local activation, 488 nm irradiation was restricted to a manually drawn ROI.

### Single-filament TIRF-M assay and data analysis.

GMPCPP-stabilized microtubule seeds were prepared as follows: 40 μL of 40 μM tubulin mixture (25%

tubulin, 25% Alexa568-tubulin, 50% biotinylated tubulin) was polymerized in seed buffer (80 mM PIPES pH 6.8, 1 mM MgCl$_2$, 1 mM EGTA, 1 mM GMPCPP (Jena Bioscience), 5 mM β-mercaptoethanol) at 37 °C. After 30 min, 400 μL of pre-warmed BRB80 (80 mM PIPES pH 6.8, 1 mM MgCl$_2$, 1 mM EGTA) was added to the tubulin mixture, which was then spun down in a table top centrifuge at $21{,}000 \times g$ for 8 min at RT. The microtubule pellet was thoroughly resuspended in 40 μL of BRB80.

TIRF flow chambers were assembled from a biotin-PEG functionalized glass coverslip attached to a PLL-PEG passivated cover slide using double-sided tape[56]. To prevent potential unspecific binding, the flow chamber was first flushed and incubated with 35 μL of blocking buffer (100 μg/mL κ-casein, 1% w/v Pluronic F-127 in assay buffer (40 mM PIPES pH 6.8, 75 mM KCl, 4 mM MgCl$_2$, 0.4 mM EGTA, 1 mM GTP, 100 μg/mL β-casein, 1 mM trolox, 20 mM β-mercaptoethanol). After 5 min, the flow chamber was washed two times with 35 μL of assay buffer. Then, the chamber was flushed and incubated with 35 μL of 150 μg/mL Neutravidin (Thermo Fisher Scientific) in assay buffer for another 5 min. The flow chamber was washed two times with 35 μL of assay buffer, and then the chamber was flushed and incubated with 35 μL of diluted (1:750) GMPCPP-stabilized microtubule seeds. After 5 min, the chamber was washed two times with 35 μL of assay buffer.

After the seeds were immobilized on the surface, 40 μL of protein mixture (10–40 μM tubulin (7.5% Alexa568-labeled), 0–20 μM stathmin, 0–30 μM phospho-stathmin, oxygen-scavenging system[56] (0.125 mg/mL Glucose-oxidase (22778.02, Serva, Heidelberg, Germany), 0.02 mg/mL catalase (C40, Sigma–Aldrich/Merck, Darmstadt, Germany), 40 mM Glucose), 0.2% w/v methylcellulose) in assay buffer was introduced into the flow chamber and the two sides were sealed. Oxygen scavengers were used to prevent photobleaching.

Imaging was performed at RT with a custom-built TIRF microscope (Olympus IX81 base) with a 60x Olympus APON TIRF objective with TOPTICA IBeam smart 560 s laser and a Quad-Notch filter (400-410/488/561/631-640). Temperature was kept at 34 °C by a collar objective heater (Bioptechs). Image acquisition was done with an EM CCD Andor iXon 888 camera controlled by Micromanager 1.4 software. Fiji ImageJ was used for data analysis. For each condition, the average MT-growth velocity and MT-catastrophe frequency were determined from ≥75 microtubules by kymograph analysis.

### Determination of MT-aster size in vitro.

Twenty microliter of protein mixture (10–40 μM tubulin (7.5% Alexa568-labeled), 0–20 μM stathmin, 0–30 μM phospho-stathmin, 2 μL Silica beads (SS06N, Bangs Laboratories, Fishers, U.S.A.)) was dispersed on a glass cover slide blocked with PLL-PEG[56] and covered with a glass coverslip pre-coated with PLL-PEG. The edges were sealed with transparent nail polish. Imaging of asters at 33 °C or in temperature ramp experiments were done as described in "Imaging of microtubule asters and morphogenic state transitions in GUVs". Twelve-micrometer thick z-stacks (1 μm spacing) were taken in the experiments at 33 °C, but single planes were taken during the temperature ramp to maximize the number of imaged fields of view.

For each individual condition, the average-projection of z-stacks for each field of view were aligned to the centrosome, and average-projected again using a custom macro in ImageJ. The radial intensity profile of these images (>30 asters per condition) was extracted with a custom ImageJ macro. The aster size was determined as the 99% decay length of a monoexponential fit to the radial intensity profile in Origin (OriginLab, Northhampton, U.S.A.).

### Morphometric analysis of GUVs

*Centrosome position and eccentricity.* A mask of the GUVs was obtained by thresholding the fluorescence intensity of fluorescently labeled tubulin at the plane of the centrosome and an ellipse was fitted to the mask in ImageJ. The centrosome position relative to the center of mask was calculated by:

$$\text{Relative centrosome position} = \frac{\left| \bar{r}_c - \bar{r}_{\text{guv}} \right|}{R_{\text{guv}}} \tag{1}$$

where $\bar{r}_c = (x_c, y_c)$ and $\bar{r}_{\text{guv}} = \left( x_{\text{guv}}, y_{\text{guv}} \right)$ are the position of the centrosome and the geometric center of the fitted ellipse obtained in ImageJ, respectively, and $R_{\text{guv}} = (M + m)/4$ where $M$ and $m$ are the major and minor axis of the ellipse. The Relative Centrosome Displacement was then calculated as the absolute difference of the centrosome positions to the first timepoint of activation.

The eccentricity of the fitted ellipse was calculated as:

$$e = \sqrt{1 - \frac{m^2}{M^2}} \tag{2}$$

*Morphometric classification.* A 3D mask of the GUV shape was made in the fluorescently labeled tubulin confocal Z-stack, by performing a 2-pixel Mean Filter and thresholding. Masks were imported to MATLAB 2018b. The centrosome position was calculated as above, except that in 3D, $R_{\text{guv}} = (PA1 + PA2 + PA3)/6$ where $PA1, 2, 3$ are the Principal Axis Lengths and $\bar{r}_c$ is the GUV solid mask centroid. These were obtained by applying the in-built function *regionprops3* to the GUV solid mask. The diameter D of the minimal bounding sphere was calculated using the *minboundsphere* function by John D'Errico available at the MathWorks

File Exchange. The diameter of the minimal inbound sphere was calculated by taking the maximum of the distance transform of the points contained in the GUV mask, to the GUV surface (edge of the mask) by means of the in-built function *bwdist*.

*Local curvature kymographs.* At each time point of the time-lapse experiments, a mask of the GUV shape was made in the fluorescence image of labeled tubulin by performing a 2-pixel Mean Filter and thresholding in Fiji. A spline was fitted to the Region of Interest (ROI) and then converted into a segmented line format. ROIs and CLSM images were imported to MATLAB 2018b. The points in the segmented line were ordered by their angle with respect to the geometric center of the GUV mask. The curvature was calculated with the function LineCurvature2D written by D. Kroon (University of Twente, August 2011) and available at the MathWorks File Exchange. The curvature at a specific angle was calculated by reconstructing a polygon with its neighboring points at a distance of five pixels.

*Fluorescence intensity angular kymographs.* For the Alexa488-SspB-AuroraB fluorescence intensity kymographs, the GUV mask at each time point was eroded and dilated using *imerode* and *imdilate*, respectively, forming a band following the shape of the GUV. The mask was segmented into 230 angular bins, and the maximum intensity at each of them was taken.

**Quantification of SspB-AuroraB membrane translocation.** Binary masks of the GUVs were obtained for the fluorescence intensity image of Tubulin (Alexa647-Tubulin) by thresholding in ImageJ. The pixels corresponding to the membrane were defined as those 7 pixels away from the edge of mask. This was obtained by eroding the mask by 7 pixels. Since the optical resolution was ~250 nm, as compared to the ~4 nm width of a lipid membrane, the fluorescence signal at pixels, which include part of the membrane contain contributions from both luminal as well as membrane located SspB-AuroraB$^{488}$. To correct for the luminal fraction contribution, the Tubulin channel was used as a reference. The fluorescence contribution of luminal SspB-AuroraB$^{488}$ can thus be estimated as

$$AuB_{lum} \frac{Tub_{mem}}{Tub_{lum}} = AuB_{lum} * \varepsilon \quad (3)$$

where $Tub_{mem}$ and $Tub_{lum}$ are the average fluorescence intensities of the tubulin$^{647}$ channel at the membrane pixels and the lumen pixels, respectively, while $AuB_{lum}$ is the average fluorescence intensity of the SspB-AuroraB$^{488}$ channel in the lumen.

The amount of translocation relative to the lumen was then calculated as

$$R(t) = \frac{AuB_{mem}(t) * (1 - \varepsilon(t))}{AuB_{lum}(t)} \quad (4)$$

To obtain translocation images of SspB-AuroraB$^{488}$, this operation was performed on a pixel basis. The analysis was performed using custom written code in MATLAB 2019b.

**Quantification of nucleation strength by relative centrosome surface.** A two-dimensional gaussian function was fitted to the centrosome on the fluorescence image of tubulin$^{647}$at the z-stack plane where the centrosome fluorescence intensity was maximal. The function followed the form

$$I(x, y) = B_1 + B_2 * e^{-\left[\frac{(x-x_0)^2}{2\sigma_x^2} + \frac{(y-y_0)^2}{2\sigma_y^2}\right]} \quad (5)$$

The centrosome size was then estimated as the area of the ellipse of axis lengths of $\sigma_x$ and $\sigma_y$, normalized by the area spanned by the GUV at the same plane. In the case multiple nucleation centers were observed, which were close together but did not form a single peak, the same fit was performed for each of them and their respective areas were added. The obtained number was scaled by a multiplicative factor of 100 for convenience. The analysis was performed with custom-made code in MATLAB 2019b.

**Image processing of fluorescent asters.** The visibility of the MT bundles in the tubulin$^{647}$ images was limited by the high background signal from free tubulin. The signal of MT bundles with respect to the background on the tubulin$^{647}$ fluorescence images was enhanced in Fiji with a custom-made macro. First, contrast enhancement was applied ("Enhance Contrast", 0.3% of saturated pixels). Then, two consecutive directional filters were applied (MorphoLibJ library, Type = Max, Operation = Mean, Direction = 32, Line = 3 and Line = 20, respectively), which are commonly used for enhancement of thin curvilinear structures, followed by an unsharp mask (Radius = 5 pixels, Mask Weight = 0.5). Lastly, the resulting image with enhanced curvilinear structures was multiplied by the original image.

**Imaging and analysis of light-induced translocation.** Imaging of light-induced translocation of SspB-AuroraB (Alexa488- or Atto542-labeled) or Alexa647-SspB-PPλ to membrane-bound C2-iLID (Alexa488-labeled or unlabeled) was done on a Leica SP8 confocal microscope (see "Imaging of microtubule asters and morphogenic state transitions in GUVs"). C2-iLID was activated by the 458 nm Argon laser line. The following laser powers were used to determine the dose-response properties of the iLID/SspB system: near maximal recruitment was observed at

30 mW/cm$^2$, so that 65 mW/cm$^2$ was used to observe recruitment kinetics and to activate the full morphogenic SynMMS system. Alexa647, Atto542, and Alexa488 were excited with 650, 542, and 495 nm white line laser lines, respectively. Detection of fluorescence emission was done in photon counting mode, restricted with an Acousto-Optical Beam Splitter (AOBS; 505–560 nm for Alexa488, 552–582 nm for Atto542, 660–700 nm for Alexa647), and further restricted by LightGate time-gating to minimize signal from laser reflection and background. In the presence of Alexa647- or Atto542-labeled proteins, images were taken in sequential frame mode where 488 nm was used to activate C2-iLID. In the presence of C2-iLID$^{488}$, 488 nm excitation was also used for fluorescence imaging. For the translocation of SspB-AuroraB$^{488}$, unlabeled C2-iLID was used, and only images of the 488 nm channel were taken to simultaneously activate C2-iLID and image SspB-AuroraB$^{488}$. Typically for activation, it was sufficient to take an image in the 488 nm channel every 3–4 s, but sometimes higher framerates (ca. 0.6 s) were used to fully resolve kinetics. Atto542-SspB-AuroraB was only used when a clear separation of activation and observation was desired, but due to bleed through, higher membrane partitioning and lower construct stability not used in further experiments in SynMMS.

For experiments on the outside membrane leaflet of the GUVs, varying protein concentrations were added to immobilized GUVs as indicated in individual figure legends. We mainly used C2-iLID, but we found no general differences in the behavior of the prenylated iLID_G in respect to recruitment and unbinding kinetics as well as to achievable surface densities of the effector proteins. The defined protein concentration in the bulk was used to calculate the protein densities on the membrane using a semi-automated tool in Matlab MathWorks, Natick, U.S.A.)[57]. Bulk depletion effects could be neglected because only 10–50 GUVs were present in each 50 µl sample. Binding and unbinding kinetics were exponentially fit in Origin (OriginLab).

For inside GUV experiments, defined starting concentrations are indicated in individual figure legends. To quantify the encapsulation efficiency and membrane densities, a defined protein concentration was added outside of the GUVs as a reference. Analysis was done in the same manner as for outside GUV experiments.

Local light-induced translocation resulted in a lateral gradient of membrane-bound proteins due to lateral diffusion, which was characterized as follows. A GUV encapsulating Atto647-SspB-AuroraB and Alexa647-SspB-PPλ and iLID was imaged for 380 and 140 s, respectively, every 2 s. Irradiating a region of interest encompassing roughly 25% of the membrane for 180 and 50 s, respectively (Fig. 3b, c). The resulting redistribution of SspB in the GUV was fit by 2D CA-simulation with the following model: SspB has a minor affinity for the membrane as discernable by the difference in contrast in the images before photo-recruitment with a partitioning resulting from association and dissociation rate constants $k_{assoc}$/$k_{dissoc}$; irradiation "creates" a tightly membrane-bound fraction at a rate constant $k_{recruit}$ that is released according to $k_{release}$, but due to the amount of iLID encapsulated concentration of SspB at the membrane is capped at $c_{max}$ (normalized intensity); both membrane-bound iLID and SspB can diffuse laterally ($D_{lat}$) and SspB in solution diffuses with a coefficient $D_{sol}$. The time-lapse of the experiment was normalized to account for bleaching and segmented to improve signal-to-noise. The simulation utilized a time resolution of 0.02 s and matched the pixel size of $0.181 \times 0.181$ µm of the confocal micrographs. By varying the parameters of the simulation via function lsqnonlin (Matlab), a best fit was achieved for $D_{lat} = 26.4$ µm$^2$s$^{-1}$; $k_{recruit} = 4.6$ s$^{-1}$; $k_{release} = 0.016$ s$^{-1}$; $k_{assoc}/k_{dissoc} = 0.38$; $c_{max} = 4.2$arb.units. The time resolution of the experiment was too slow to resolve the very fast diffusion of SspB in solution, so any $D_{sol} > 200$µm$^2$s$^{-1}$ fits the data well. Any discrepancy in kinetic values stems from the fact that the apparent binding and unbinding reaction fit in Fig. 3a is a convolution of diffusion, membrane association/dissociation and photo-recruitment kinetics.

**Quantifying regularity of SspB-AuroraB clusters on the membrane.** To quantify the regularity of SspB-AuroraB clusters on the GUV surface (Fig. 3e, f and Supplementary Fig. 4a-d) we employed generalized recurrence quantification analysis[35,58,59]. A recurrence plot (RP) is an advanced technique of nonlinear data analysis that represents a two-dimensional binary diagram indicating the recurrence that occur in an $m$-dimensional phase space within a fixed threshold $\varepsilon$ at different times $i,j$. For one-dimensional time series, the RP is expressed as a square matrix of zeros (nonoccurrences) and ones (occurrences) of states $\vec{x}_i$ and $\vec{x}_j$ of the system: $R_{ij} = \Theta(\varepsilon - ||\vec{x}_i - \vec{x}_j||)$, where $\vec{x}_i \in R^m$, $i, j = 1,..N$, N is the number of measured states $\vec{x}_i$, $\Theta(\cdot)$ is the step function and $|| \cdot ||$ is a Euclidean norm. To analyze the regularity of SspB-AuroraB on the surface of the GUV, we used an extension of this analysis for spatial data[58,59]. For a $d$-dimensional Cartesian space, $2 \times d$ dimensional spatial recurrence matrix can be defined by: $R_{\vec{i}\,\vec{j}} = \Theta(\varepsilon - ||\vec{x}_{\vec{i}} - \vec{x}_{\vec{j}}||)$, where $\vec{x}_{\vec{i}} \in R^m$, $\vec{i}, \vec{j} \in N^d$ (example projections in Supplementary Fig. 4d). The recurrence quantifications are done based on the diagonal hypersurfaces defined as surfaces of dimension $N^d$ which are parallel to the hypersurface of identity (defined by, $R_{\vec{i}\,\vec{j}} \equiv 1 \forall \vec{i} = \vec{j}$). The size of the diagonal hypersurfaces, $l(\vec{l} = (l, \dots, l), \vec{l} \in N^d)$ defined by $(1 - R_{\vec{i}-\vec{1},\vec{j}-\vec{1}})(1 - R_{\vec{i}+\vec{l},\vec{j}+\vec{l}})$ $\prod_{k_1,\dots,k_d}^{l-1} R_{\vec{i}+\vec{k},\vec{j}+\vec{k}} \equiv 1$, were calculated, and a distribution of surface sizes ($P(l)$)

was used to calculate the Information entropy. The Information entropy (ENTR) is defined by $ENTR = -\sum_{l=l_{min}}^{N} p(l)*\ln(p(l))$, where $\ln(.)$ is the natural logarithm, $p(l)$ is the probability of a diagonal hypersurface with length $l$, such that $p(l) = \frac{P(l)}{N_l}$ and $N_l$ is the total number of diagonal hypersurfaces with length greater than a fixed minimum length $l_{min}$.

The 2D maps of the GUV surface were projected from confocal stacks using the Map3-2D software[60]. From the spatial map of the SspB-AuroraB on the GUV surface (Fig. 3e, Supplementary Fig. 4a-c), a rectangular region was selected. Regions with lipid defects or strong SspB-AuroraB aggregations were excluded by thresholding and filled with NaN values. Spatial recurrence quantification analysis with $\varepsilon = 0.3$ and $l_{min} = 3$ was performed on the corresponding rectangular images. Same procedure and parameters ($\varepsilon$, $l_{min}$) were used for quantification of the SspB-PPλ spatial distributions, in the respective SspB-AuroraB and SspB-PPλ randomization controls (Supplementary Fig. 4 h), as well as the quantification of the numerically obtained patterns (Supplementary Fig. 4 h).

To quantify the SspB-AuroraB (SspB-PPλ) fraction in lumen, the 3D z-stack of the GUVs were used. For each stack, the membrane and the lumen were separately masked. The mean intensity per pixel in the 3D lumen $\left(C_{lumen_{mean}}\right)$ and on the 2D membrane $\left(C_{membrane_{mean}}\right)$ were calculated using imageJ. SspB-AuroraB fraction in the lumen $\left(C_{lumen/total}\right)$ was calculated using: $\left(C_{lumen/total} = \frac{C_{lumen_{mean}}}{C_{lumen_{mean}} + C_{membrane_{mean}}}\right)$. For the significance testing, the spatial AuroraB pattern of each GUV was shuffled uniformly at random, and the information entropy of the obtained pattern was calculated. The results in Fig. 3f show the mean ± sd from 50 independent runs.

**COPY$^o$ characterization**. Gly-Stathmin-Cys was sequentially labeled with two organic dyes to generate the sensor COPY$^o$ ($^o$=organic). First, the acceptor dye was conjugated to the C-terminal cysteine. Typically, 3 mg of Gly-Stathmin-Cys were incubated with 5 mM TCEP for 1 h at room temperature under an argon atmosphere before buffer exchange via a NAP5 column (GE Healthcare) into degassed 50 mM Hepes pH 7.5, 200 mM NaCl, 2 mM MgCl$_2$. A 10-fold molar excess of Atto655-maleimide (Atto-tec) was added and the mixture incubated for 3 h at room temperature under an argon atmosphere. The reaction was stopped by adding 5 mM DTT. Next, without further purification, the donor dye Atto532 (Atto-tec) was conjugated to the N-terminus as described in the section "Protein labeling". For enzyme kinetic measurements in bulk solution another sensor version was constructed with Atto647N (Atto-tec) as acceptor dye, which we termed COPY$^{o2}$.

Initial characterization was carried out by evaluating sensitized emission (Supplementary Fig. 5a and 6a). Fluorescence-emission spectra were measured with a QuantaMaster fluorescence system (PTI) (exc. 520 nm, em. 530–720 nm). 1 μM sensor was in 80 mM PIPES pH 6.8, 75 mM KCl, 1 mM EGTA, 2 mM MgCl$_2$, 20 mM β-mercaptoethanol. 10 μM tubulin was added to measure the decrease of sensitized emission.

Detailed characterization of the FRET properties was done by analysis of donor fluorescence lifetime. Confocal FLIM experiments were performed on a Leica SP8 confocal microscope (see "Imaging of microtubule asters and morphogenic state transitions in GUVs") equipped with the FALCON (FAst Lifetime CONtrast) system. Atto532 was excited at 532 nm with a pulse frequency of 40 MHz and emission was collected with three Leica HyD detectors restricted with an Acousto-Optical Beam Splitter (AOBS) to 542–563, 568–591, and 591–620 nm. Data were collected for the donor-only labeled Stathmin, free COPY$^o$ and COPY$^{o2}$ (3 μM), and tubulin-bound sensors (+20 μM tubulin) in cDICE base buffer (see "Protein encapsulation in GUVs by cDICE") +60 μM GTP, 300 mM sucrose, and additional 1 mM trolox. Photons were collected up to ~3000 photons per pixel in the sum of all three detectors. Global analysis of FLIM-FRET was implemented in a custom program written in Python according to a described process[61,62], yielding $\tau_D$ (ns) and $\tau_{DA}$ (ns) as well as FRET efficiency. Additionally, all pixel values of the output lifetime images were averaged to obtain the absolute lifetimes of the donor for the free and tubulin-bound sensor states (Supplementary Fig. 5b; Supplementary Fig. 6b).

Due to the higher signal amplitude obtainable with COPY$^{o2}$, we preferred this variant for stopped-flow and most kinetic measurements. However, since Atto647N (acceptor dye in COPY$^{o2}$) is positively charged and is known to interact with membranes[63], we used COPY$^o$ for the measurement of stathmin phosphorylation gradients in GUVs.

**Stopped-flow measurements**. Association and dissociation of COPY$^{o2}$ and tubulin were measured in a stopped-flow apparatus (Applied Photophysics, Leatherhead, U.K.) with excitation at 535 nm and a 670/30 nm band-pass filter to monitor sensitized emission changes at 25 °C in 80 mM PIPES pH 6.8, 75 mM KCl, 2 mM MgCl$_2$, 20 mM β-mercaptoethanol, 60 μM GTP. COPY$^{o2}$ or phosphorylated COPY$^{o2}$ (see "Time-resolved measurements of AuroraB and λ-PPase activity") was kept constant at 200 nM for the tubulin titration. Dissociation was measured at 200 nM COPY$^{o2}$ and 4 μM tubulin, displacing COPY$^{o2}$ with 60 μM unlabeled stathmin. Data were fit in Origin (OriginLab), and error propagation for derived

constants was calculated by Gaussian equation taking the errors of the fit as a starting point.

**Determination of enzymatic kinetic parameters**. Stathmin phosphorylation and dephosphorylation kinetics were measured using the fluorescence ratiometric signal or lifetime change of COPY$^o$ or COPY$^{o2}$ as indicated in the respective figure legends. Confocal FLIM measurements were performed as described for "Stathmin FRET-sensor generation and characterization" with few modifications. Excitation pulse frequency was 20 MHz. Per timepoint 1–2 million photons were collected in a $256 \times 256$ pixels image. During analysis, 4× binning was applied. Measurements were conducted in cDICE buffer (see "Protein encapsulation in GUVs by cDICE") + 60 μM GTP, 300 mM sucrose, −EGTA, and +0.8 mM MnCl$_2$ when PPλ was used.

To measure dephosphorylation kinetics, COPY$^o$ was pre-phosphorylated (COPY$^o$-P) with 500 nM AuroraB by incubating with 4 mM ATP for 8 h at room temperature and used without further purification. Residual kinase activity was neglected because AuroraB partially loses its activity during the incubation, and also adding PPλ deactivates AuroraB in the final sample for the kinetic measurement. Kinetic traces of dephosphorylation were obtained at varying concentrations of COPY$^o$-P (with tubulin present in at least two-fold excess) and 200 nM PPλ. Values for $k_{cat}$ and $K_m$ were determined from a fit to the Michaelis–Menten equation. Reported errors are errors of the fit, and the error for $k_{cat}/K_m$ was derived by Gaussian error propagation.

Since phosphorylation of stathmin under conditions suitable for Michaelis–Menten analysis was too slow to yield sufficiently stable and analyzable signal amplitudes, we estimated $k_{cat}/K_m$ from monoexponential fits of single-progress curves. 10 μM COPY$^{o2}$ (+20 μM tubulin) was phosphorylated by 500 nM or 1 μM SspB-AuroraB, and the data were fit mono-exponentially. The apparent rate constant was converted to a value corresponding to $k_{cat}/K_m$ by taking the used kinase concentrations into account and averaging the two obtained values. Fit errors were propagated by the Gaussian equation.

Note that the derived kinetic parameters for PPλ and SspB-AuroraB do not consider the presence of multiple phosphorylation sites on stathmin or how many of them need to be phosphorylated to induce tubulin dissociation. The parameters apply for the catalysis of the overall transition between free and tubulin-bound stathmin, which is our direct readout and also the relevant state change for the derivation of the signaling gradient.

**Calculations of steady-state pStathmin fraction and effective tubulin concentration**. The steady-state phosphorylated stathmin fraction was calculated with mass action kinetics and the kinetic parameters obtained from Michaelis–Menten analysis described above:

$$[S] = \frac{k_{-1} \times [S_t] \times [PP\lambda]}{k_1 \times [A] + k_{-1} \times [PP\lambda]} \tag{6}$$

with

$$[S_t] = [S] + [S_p] \tag{7}$$

where: $[S_t]$ is the total stathmin concentration, $[S]$ the unphosphorylated and $[S_p]$ the phosphorylated stathmin concentration. $[A]$ is the SspB-AuroraB and $[PP\lambda]$ the PPλ concentration with $k_{-1}$ being the $k_{cat}/K_M$ of the dephosphorylation and $k_1$ the $k_{cat}/K_M$ of the phosphorylation reaction taken from Supplementary Table. 1. For SspB-AuroraB a concentration of 7 μM and for PPλ of 0.5 μM was used consistent with the concentration in the experiments.

The tubulin concentration sequestered by stathmin was calculated by considering a phosphorylated stathmin fraction of 16% and using the four dissociation constants obtained from stop-flow (Supplementary Table. 1). The tubulin-bound fraction of stathmin per binding side was determined:

$$\text{Bound fraction} = \frac{[tub]}{k_D + [tub]} \tag{8}$$

where, $[tub]$ is the total tubulin concentration in the system. In the next step the effective tubulin concentration was calculated by subtracting the bound fraction of tubulin from the total concentration of tubulin considering a phosphorylation level of 16%:

$$[tub_{eff}] = [tub] - 0.16 \times [stat]\left(bf_1^{pStat} + bf_2^{pStat}\right) - 0.84 \times [stat]\left(bf_1^{stat} + bf_2^{stat}\right) \tag{9}$$

where, $[stat]$ is the stathmin concentration of 4 μM and bf are the bound fraction of tubulin per respective binding site and phosphorylated or unphosphorylated stathmin is indicated in superscript ($^{pStat}$, $^{stat}$).

**Measurement of stathmin phosphorylation gradients**. Spatial COPY$^o$ phosphorylation gradients were imaged and quantified by computing the ratio of Atto655/Atto532 fluorescence-emission intensities. First, we derive the relationship between the fluorescence FRET ratio to the fraction of tubulin-bound stathmin.

The fluorescence intensity detected at the acceptor channel $F_A$ can be separated into three contributions:

$$F_A = k_A[S] + K_{FA}^{pS}[pS] + K_{FA}^{tS}[tS] \tag{10}$$

where: $k_A$ is a constant proportional to the fluorophore brightness, $[S]$ is the total stathmin concentration, and its product represents the intensity in absence of donor; $K_{FA}^{pS}$ and $K_{FA}^{tS}$ are the contributions to the intensity due to FRET when stathmin is either phosphorylated or tubulin bound, while $[pS]$ and $[tS]$ are their concentrations, respectively. We assume that stathmin can only adopt these two states and hence $[S] = [pS] + [tS]$.

The fluorescence at the donor channel $F_D$ can be written as:

$$F_D = [S]Q = \frac{k_D[S]}{k_D + \alpha K_{FD}^{pS} + (1-\alpha)K_{FD}^{tS}} \tag{11}$$

where Q is the quantum yield, $k_D$ is a constant proportional to the donor fluorophore brightness, and $K_{FD}^{pS}$, $K_{FD}^{tS}$ are the contributions to donor quenching by FRET, in the corresponding stathmin states, and $\alpha = [pS]/[S]$ is the fraction of tubulin-bound stathmin

Normalizing the acceptor fluorescence intensity to the total stathmin concentration yields:

$$F_A = \left[k_A + K_{FA}^{pS}\alpha + K_{FA}^{tS}(1-\alpha)\right][S] \tag{12}$$

After rearranging terms, we obtain:

$$F_A = \left[(k_A + K_{FA}^{tS}) + \alpha(K_{FA}^{pS} - K_{FA}^{tS})\right][S] \tag{13}$$

Taking the FRET fluorescence ratio of acceptor fluorescence to donor fluorescence can be then expressed as:

$$R = \frac{F_A}{F_D} = \frac{1}{k_D}\left[(k_A + K_{FA}^{tS}) + \alpha(K_{FA}^{pS} - K_{FA}^{tS})\right]\left[k_D + \alpha K_{FD}^{pS} + (1-\alpha)K_{FD}^{tS}\right] \tag{14}$$

where $R = F_A/F_D$ is the fluorescence FRET ratio. Rearranging:

$$R = \frac{F_A}{F_D} = \frac{1}{k_D}\left[(k_A + K_{FA}^{tS}) + \alpha\left(K_{FA}^{pS} - K_{FA}^{tS}\right)\right]\left[(k_D + K_{FD}^{tS}) + \alpha(K_{FD}^{pS} - K_{FD}^{tS})\right] \tag{15}$$

Grouping the remaining constants between parenthesis as,

$$\begin{aligned} K_1 &= k_A + K_{FA}^{tS} \\ K_2 &= K_{FA}^{pS} - K_{FA}^{tS} \\ K_3 &= k_D + K_{FD}^{tS} \\ K_4 &= K_{FD}^{pS} - K_{FD}^{tS} \end{aligned} \tag{16}$$

Simplifies the expression to:

$$R = \frac{1}{k_D}[K_1 + \alpha K_2][K_3 + \alpha K_4] \tag{17}$$

Distributing the product yields:

$$R = \frac{1}{k_D}\left[K_1 K_3 + \alpha(K_1 K_4 + K_2 K_3) - \alpha^2 K_2 K_4\right] \tag{18}$$

Since FRET is negligible in the tubulin-bound state, $K_{FA}^{tS}$ and $K_{FD}^{tS}$ are negligible, and because the FRET efficiency of COPY$^o$ is <30%, the product $K_{FD}^{pS}K_{FA}^{pS}$ is also negligible since both magnitudes depend on the intrinsic FRET efficiency. Hence, the quadratic term becomes negligible, so we can solve for $\alpha$ as:

$$\alpha = \frac{k_D R - K_1 K_3}{K_1 K_4 + K_2 K_3} \tag{19}$$

Therefore, the fraction of phospho-stathmin $\alpha$ is an approximate linear function of the fluorescence FRET ratio $R$.

The ratiometric intensity is more photon efficient than fluorescence lifetime measurements and therefore provides the sensitivity to detect spatial gradients. Confocal imaging was done on a Leica SP8 microscope (see "Imaging of microtubule asters and morphogenic state transitions in GUVs"). COPY$^o$ was excited at 532 nm, and detection was done in photon counting mode, restricted to 542–560 nm and 640–750 nm for donor and acceptor, respectively. The imaging focus was maintained by Leica Adaptive Focus Control. Imaging conditions were identical when measuring outside of or inside GUVs. The following protein concentrations were used for encapsulation: 40 μM (final) tubulin, 4 μM (final) COPY$^o$, 5 μM C2-iLID, 12 μM SspB-AuroraB (unlabeled), 1 μM λ-PPase. Outside GUVs the following concentrations were used: 20 μM tubulin, 20 μM COPY$^o$, 1 μM C2-iLID, 1.5 μM SspB-AuroraB (unlabeled), 1 μM λ-PPase 5 μM C2-iLID, 12 μM SspB-AuroraB (unlabeled) and varying λ-PPase concentrations as indicated.

For the "dark" state (no C2-iLID activation) 40 images at the equatorial plane were taken with 3 s intervals. This spacing was necessary to prevent unwanted C2-iLID activation by 532 nm light.

Activation of C2-iLID was done with the 458 nm Argon laser line for the duration of 3 min with 3 s intervals. This sustained activation was performed to allow SspB-AuroraB to undergo full autoactivation and the steady state of stathmin

phosphorylation to be reached. C2-iLID fluorescence of flavin mononucleotide was detected between 520 and 568 nm to be used as a membrane marker during the analysis. After this C2-iLID activation, 40 images with minimal intervals were taken for COPY$^o$ imaging. A sequential line scan was performed between C2-iLID activation and COPY$^o$ imaging in order to maintain SspB-AuroraB translocation.

In case there was any GUV movement during dark or activated image series acquisition, the frames were aligned in ImageJ using the MultiStackReg plugin based on StackReg[64–67]. Then, all images of the series were summed up (more precise than averaging, and made possible by photon counting). We observed an optical xy-aberration between the donor and acceptor channels, which led to asymmetric edge effects around the GUVs in the ratio image. To avoid these, the channels were manually registered to yield a symmetric ratio image calculated as acceptor/donor. Radial profiles were extracted from the ratio images by a custom macro in ImageJ using flavin mononucleotide fluorescence of C2-iLID as a marker to determine the starting position at the membrane and align the profiles of individual GUVs.

Data obtained in this manner were sufficient to describe gradients arising outside of GUVs. To visualize the gradient amplitude in cluster vicinity, we measured 7–8-μm thick z-stacks of the ratio (1 μm spacing, 10 images per slice) on sufficiently large GUVs at their equatorial plane, so that the GUV diameter would not significantly change throughout the stack. Maximum-intensity projections along the z-axis were done to highlight gradient hotspots.

**Monte-Carlo simulations of MT-membrane interactions**. To simulate the collective organization of the astral-MT-population through self-induce capture and depletion of free MTs as experimentally observed (Fig. 1), we implemented an agent-based model describing the nonequilibrium fluctuations in MT-growth kinetics[25,26] and the cooperative bundling mechanism upon MT-induced membrane protrusions, using Monte-Carlo simulations in 1D circular geometry.

For this, a defined number of MTs are nucleated and evolve from fixed initial positions, mimicking a growth from a MT-organizing center. This initial MT–distribution reflects an angular distribution around the MT-organizing center [0°, 360°] represented in a 1D line, which is periodically connected, and the distance of the membrane is set to 25 μm, approximating an average GUV radius. Similarly as in a previous study[26], it is assumed that: (1) MTs are straight elements composed of a linear chain of subunits representing the length of the MT without incorporating molecular details; however, the correct dimension was retained that 1 μm of MT length is made up of 1624 dimers; (2) MT-growth rate is linearly dependent on the free-tubulin concentration; (3) tubulins released upon catastrophe events are immediately competent to polymerize without delay. According to Supplementary Fig. 1b, (4) the MT-catastrophe rate $k_C$ nonlinearly decreases with increase in tubulin concentration, $k_c \sim c_3 e^{-c_4[\text{Tub}]}$. The concentration of total tubulin was set to $[\text{Tub}] = 35$ μM, and the number of nucleation sites was set to 20 or 80, which also determines the maximum number of MTs. (5) Complete MT depolymerization immediately opens up that nucleation site for a new nucleation event. When a MT reaches the membrane, we assume that it pushes against it and generates local deformations. At this point, (6) the membrane exerts an elastic force on the MT proportional to the length of the protrusion ($F_e \sim c_1 \cdot \Delta h$) that effectively slows down the growth velocity ($v_g \sim c_2$ $[\text{Tub}]\, e^{-Fe}$). The membrane deformation caused by protruding MTs can capture free MTs, thereby generating a stigmergic effect in which MT bundles are self-amplified and the deformations are stabilized (Fig. 1e). (7) We assume that each protruding MT will attract free neighboring radial MTs at a distance that scales with the length of the protrusion with an intensity of attraction $I_a \sim \frac{\Delta h}{h}\left(1 - \frac{x}{\sigma(\Delta h)}\right)$. This dependency was derived from generic stigmergic models[68], where $\sigma$ is the size of the attracting neighborhood corresponding to the size of the membrane deformation, x—position of the free radial MTs to the deformation source, $\Delta h$—the extent of the deformation-, and h—the total MT length. (8) Free radial MTs that are initially positioned in two overlapping neighborhoods will be assigned to the deformation with a higher $I_a$. Keeping the position of the deforming MT fixed, the positioning of the free radial MTs is thereby updated to mimic formation of a MT-bundle. (9) Formed MT bundles can also attract a MT-bundle that is localized in the attracting neighborhood if they have a higher intensity of attraction.

Python-generated plot (Supplementary Fig. 1g) depicts the evolution of the system towards polar ($\text{MT}_{\text{total}} = 20$, top row) or star-like MT-bundle distribution ($\text{MT}_{\text{total}} = 80$, bottom row), when all remaining parameters in the simulation are fixed. The simulation parameters were taken as in ref. [26]: $c_1 = 0.06$ pN/μm (such that on average, the pushing force is in same order of magnitude as reported[69]); $c_2 = 0.005$ μm/s·μM; $c_3 = 0.04\ \text{s}^{-1}$; $c_4 = 0.1\ \mu\text{M}^{-1}$, shrinkage velocity $v_s = 0.218$ μm/s, rescue frequency $k_{res} = 0.175\ \text{s}^{-1}$; total simulation time was set to $N = 2000$ time steps. $\sigma(\Delta h)$ is linearly proportional to the extent of the deformation with a scaling factor 0.05.

In the algorithm and correspondingly Video 2, the bundling is mimicked by overlaying the positioning of the attracted MTs to the position of the MT-bundle, and thereby the thickness of the lines does not represent the number of MTs in a bundle. The respective thickness of the MT bundles is depicted only in Supplementary Fig. 1g, where the individual MTs constituting a particular bundle are shown.

**Paradigmatic model of morphogenesis.** The dynamic behavior of the two subsystems, the self-organized morphogenesis of the encapsulated MT-aster as well as the spatial organization of SspB-AuroraB clusters on the GUV membrane, when isolated, can be conceptualized with paradigmatic reaction–diffusion models, where self-amplification of local structures triggers depletion of their free substrate. Generically, the dynamics of each of the subsystems can be represented as:

$$\frac{\partial u_i}{\partial t} = f(u_i, \nu_i) + D_{u_i} \nabla^2 u_i$$
$$\frac{\partial \nu_i}{\partial t} = -f(u_i, \nu_i) + D_{\nu_i} \nabla^2 \nu_i \qquad (20)$$

with i = 1,2 corresponding to the MT-membrane (Fig. 1f) and the SspB-AuroraB (Fig. 3h) subsystems, respectively; and $f(u_i, \nu_i) = (k_0 + \gamma_i u_i^2)\nu_i - \omega_i u_i$. For the MT-membrane subsystem, $u_1$ corresponds to the density of SIC-bundled MTs, whereas $\nu_1$ represents the density of single MTs in the membrane vicinity, $u_1 + \nu_1 = c_1 = const.$, and $\nabla^2 = \frac{1}{R^2}\frac{\partial^2}{\partial\theta^2}$.

For the SspB-AuroraB subsystem, $u_2$ corresponds to the density of clustered SspB-AuroraB on the membrane, $\nu_2$ –density of SspB-AuroraB monomers on the membrane, $u_2 + \nu_2 = c_2 = const.$, and $\nabla^2 = \frac{\partial^2}{\partial x^2} + \frac{\partial^2}{\partial y^2}$.

For Fig. 1g and Supplementary Fig. 1h-i, $\omega_1 = 0.5$, $k_0 = 0.067$. For Supplementary Fig. 4e, $\gamma_2 = 5$, $\omega_2 = 5$, $k_0 = 0.067$, in Fig. 3i and Supplementary Fig. 4h, $c_2 = 5$, $\gamma_2 = 10$, $\omega_2 = 5$, $k_0 = 0.067$.

The MT-membrane subsystem, similarly to the Monte-Carlo simulations, captures the self-organization into SIC-bundled MTs at a fixed number of MTs. Although the RD abstraction does not take into consideration the stochastic nature of MTs, the qualitative behavior in terms of formation of stable patterns can be equally well described using this abstraction (compare Supplementary Fig. 1g top with Fig. 1g, and Supplementary Fig. 1g bottom with Supplementary Fig. 1i). The simulations are performed in 1D circular geometry with periodic boundary conditions, starting from random initial conditions uniformly distributed around the homogeneous steady state. We use a fixed radius of R = 1 arb. units, and θ is the angle in degrees.

For the SspB-AuroraB experimental system, under continuous light illumination, a fixed amount of SspB-AuroraB is translocated to the membrane that is limited by the amount of available iLID. Thus, the dynamics of the system can be well described by Eq. (20) (i = 2). To demonstrate that this subsystem has a stable interval in which pattern formation is observed, 2D simulations of SspB-AuroraB pattern formation on a Euclidean grid with no-flux boundary conditions is shown in Fig. 3i, when starting from random initial conditions uniformly distributed around the homogeneous steady state.

The dynamics of the SspB-AuroraB subsystem can be equivalently described using a 3-component model where the AuroraB monomers in the lumen are explicitly considered:

$$\frac{\partial u_2}{\partial t} = f(u_2, \nu_2) + D_{u_2} \nabla^2 u_2$$
$$\frac{\partial \nu_2}{\partial t} = f_1(u_2, \nu_2, w_2) + D_{\nu_2} \nabla^2 \nu_2 \qquad (21)$$
$$\frac{\partial w_2}{\partial t} = f_2(\nu_2, w_2) + D_{w_2} \nabla^2 w_2$$

with $f_1(u_2, \nu_2, w_2) = -f(u_2, \nu_2) + k_{on}w_2 - k_{off}\nu_2$, $f_2(\nu_2, w_2) = -k_{on}w_2 + k_{off}\nu_2$; $u_2$ and $\nu_2$ as above, and $w_2$—SspB-AuroraB monomers in the lumen, $\nabla^2 = \frac{\partial^2}{\partial x^2} + \frac{\partial^2}{\partial y^2}$.

For Supplementary Fig. 4g, $c_2 = 8$, $\gamma_2 = 5$, $\omega_2 = 5$, $k_0 = 0.067$, $k_{on} = 0.1$, $k_{off} = 0.2$.

To provide quantitative description of the symmetry breaking in both subsystems, we applied Linear Perturbation Analysis (LPA)[70,71] to Eq. (20) that in turn allows to analyze the bifurcation structure of a reaction–diffusion (RD) system. In brief, the method takes advantage of the diffusion discrepancy $D_{u_i} \ll D_{\nu_i}$ and considering the limit $D_{u_i} \to 0, D_{\nu_i} \to \infty$, the stability of the homogenous steady state can be probed with respect to a 'local' perturbation in the form of a localized peak of $u_i$ with a negligible total mass, at some location in the domain. In this limit, the localized peak of $u_i(u_{L,i})$ does not influence the surrounding background $u_i$ levels ($u_{G,i}$), whereas due to the 'infinitely fast' diffusion, $\nu_i$ can be described by a single global variable ($\nu_{G,i}$) that is approximately uniformly distributed in space. This allows writing a set of ODEs that describe the evolution of the perturbation peak as:

$$\frac{du_{L,i}}{dt} = g(u_{L,i}, c_i - u_{G,i})$$
$$\frac{du_{G,i}}{dt} = g(u_{G,i}, c_i - u_{G,i}) \qquad (22)$$

with

$$g(u_{L,i}, c_i - u_{G,i}) = (k_0 + \gamma_i \cdot u_{L,i}^2)\cdot(c_i - u_{G,i}) - \omega_i \cdot u_{L,i}$$
$$g(u_{G,i}, c_i - u_{G,i}) = (k_0 + \gamma_i \cdot u_{G,i}^2)\cdot(c_i - u_{G,i}) - \omega_i \cdot u_{G,i} \qquad (23)$$

Dynamics of these ODE systems (Eq. 22) under variation of $c_i$ and $\gamma_i$ was numerically explored using the Xppaut bifurcation software[72] to produce

Supplementary Fig. 4e. For the simulations presented in Figs. 1, 3 and Supplementary Figs. 1, 4, the total AuroraB on the membrane or MT amount was fixed after the symmetry-breaking point.

In SynMMS, a joint dynamical system is established, where the SIC protrusion-induced change in the membrane geometry enhances SspB-AuroraB translocation and thereby its activity, that in turn locally promotes the MT growth (Fig. 8a). The joint system is thus described with:

$$\frac{\partial u_1}{\partial t} = h(u_1, \nu_1, u_2) + D_{u_1} \nabla^2 u_1$$
$$\frac{\partial \nu_1}{\partial t} = -h(u_1, \nu_1, u_2) + D_{\nu_1} \nabla^2 \nu_1$$
$$\frac{\partial u_2}{\partial t} = h(u_2, \nu_2, u_1) + D_{u_2} \nabla^2 u_2 \qquad (24)$$
$$\frac{\partial \nu_2}{\partial t} = -h(u_2, \nu_2, u_1) + D_{\nu_2} \nabla^2 \nu_2$$

$h(u_i, \nu_i, u_j) = (k_0 + k_{3/4} \cdot u_j + \gamma_i \cdot u_i^2)\cdot \nu_i - \omega_i \cdot u_i$, when i = 2, j = 2 then $k_{3/4} = k_3$, when i = 2, j = 1 then $k_{3/4} = k_4$. For i = 1,2, $\nabla^2 = \frac{1}{R^2}\frac{\partial^2}{\partial\theta^2}$.

Using the LPA analysis in this case yields a set of ODEs that describe the evolution of the perturbation peak as:

$$\frac{du_{L,1}}{dt} = h(u_{L,1}, u_{L,2}, c_1 - u_{G,1})$$
$$\frac{du_{G,1}}{dt} = h(u_{G,1}, u_{G,2}, c_1 - u_{G,1})$$
$$\frac{du_{L,2}}{dt} = h(u_{L,2}, u_{L,1}, c_2 - u_{G,2}) \qquad (25)$$
$$\frac{du_{G,2}}{dt} = h(u_{G,2}, u_{G,1}, c_2 - u_{G,2})$$

where,

$$h(u_{L,1}, u_{L,2}, c_1 - u_{G,1}) = (k_0 + k_3 \cdot u_{L,2} + \gamma_1 \cdot u_{L,1}^2)\cdot(c_1 - u_{G,1}) - \omega_1 \cdot u_{L,1}$$
$$h(u_{G,1}, u_{G,2}, c_1 - u_{G,1}) = (k_0 + k_3 \cdot u_{G,2} + \gamma_1 \cdot u_{G,1}^2)\cdot(c_1 - u_{G,1}) - \omega_1 \cdot u_{G,1}$$
$$h(u_{L,2}, u_{L,1}, c_2 - u_{G,2}) = (k_0 + k_4 \cdot u_{L,1} + \gamma_2 \cdot u_{L,2}^2)\cdot(c_2 - u_{G,2}) - \omega_2 \cdot u_{L,2}$$
$$h(u_{G,2}, u_{G,1}, c_2 - u_{G,2}) = (k_0 + k_4 \cdot u_{G,1} + \gamma_2 \cdot u_{G,2}^2)\cdot(c_2 - u_{G,2}) - \omega_2 \cdot u_{G,2}$$
$$(26)$$

The RD simulations in Fig. 8 and Supplementary Fig. 9 are performed in 1D circular geometry with periodic boundary conditions, starting from random initial conditions uniformly distributed around the homogeneous steady state, with $c_1 = 1000$, $c_2 = 2750$, $k_0 = 0.067$, $\omega_1 = 0.5$, $\omega_2 = 5$, $k_3 = 1$, $k_4 = 0.5$ (except in Supplementary Fig. 9g, where $k_4 = 10$).

As above, the dynamics of this ODE system (Eq. 25) under variation of the total SspB-AuroraB concentration ($c_2$) was explored, demonstrating that the coupled system can break symmetry in a broad parameter range (Supplementary Fig. 9a-c).

For the reaction–diffusion simulations in Fig. 8b, c and Supplementary Fig. 9 f, an external signal source corresponding to local/global light irradiation is applied that increases the amount of free SspB-AuroraB monomers on the membrane and thereby enhances the cooperative clustering. This is implemented by including ρ, which is proportional to the signal strength as $\rho \cdot \gamma_2 \cdot u_2^2$. To model the influence of local signal, a Gaussian distribution was used.

All RD simulations were performed using a custom-made Python code, with $D_{u,i} = 0.1, D_{\nu,i} = 40$, except in Fig. 3i, Supplementary Fig. 4h $D_{u2} = 1, D_{\nu2} = 50$, and in Supplementary Fig. 4g $D_{u2} = 1, D_{\nu2} = 1, D_{w2} = 50$. The integration time was set to $1.5 \cdot 10^8$, the time step – $7.5 \cdot 10^{-5}$, for a total of $10^3$ spatial bins with step size 0.1.

**Reaction-diffusion simulation of gradients.** We utilize numeric simulations of the reaction–diffusion system to visualize the gradients resulting from localizing a kinase to the membrane of a GUV. Our kinetic measurements confirm two tubulin binding sites on stathmin. The low-affinity site (KD = 2 μM for the unphosphorylated and 12 μM for the phosphorylated state) is predominantly occupied by a tubulin heterodimer with a high turnover rate, almost irrespective of phosphorylation. At the high affinity site however, KD changes from ~30 nM to 610 μM as reflected in the simulation by a drastic difference in association/dissociation rate constants. We therefore consider a simplified setup of five "species": (1) free-tubulin dimers and (2) free stathmin, (3) stathmin-tubulin complex (in a 1:1 stoichiometry), (4) phosphorylated stathmin-tubulin complex, and (5) phosphorylated free stathmin. Here, species 2–5 are actually weakly associated with a second tubulin dimer that slows down effective diffusion, but whose binding state is not strongly influenced by phosphorylation. We assume that the pairwise association/dissociation and kinase/phosphatase reaction can be considered first-order processes governed by a back and a fourth rate constant (equivalent to sub-saturation Michaelis–Menten-Kinetics) to reduce the number of parameters in the system. Considering a well-mixed volume (diffusion much faster than reaction), the balance of kinase/phosphatase reaction determines the steady-state levels of phosphorylated stathmin. For our measurements of $k_{cat}/k_M$ (PPase 0.02 μM$^{-1}$s$^{-1}$;

kinase 0.0011 $\mu M^{-1} s^{-1}$), at a stoichiometry of 1:1 about 5–10% of stathmin should be phosphorylated with a homogenously flat distribution in the GUV. By recruitment to the membrane, the concentration of the kinase is increased by a factor of up to 500. For this, a 5× increase of fluorescence intensity suffices, as the resolution of ~500 nm dilutes the signal of the "shell" of recruited kinase with a thickness of ~5 nm close to the membrane. Such recruitment leaves a significant fraction of kinase in solution, but due to the unchanged phosphatase activity resulting in a lowered phosphorylated fraction. In a see-saw-like manner, the high kinase activity at the membrane "inverts" phosphorylation after recruitment and leave a fraction of <4% unphosphorylated stathmin there. If the phosphatase activity was also recruited, but to the center of the GUV, the result would be a linear gradient completely independent of diffusion, the amplitude of which would solely be determined by the relative strength of kinase and phosphatase activity. In such a case, diffusion would only change the amount of phosphorylated protein transported and how fast the gradient is established. However, as the soluble phosphatase is distributed evenly, the shape of the gradient arises from the interplay of diffusion, phosphatase activity and association/dissociation kinetics of stathmin-tubulin. This defeats any attempt of analytical solutions in more than simplified geometries and requires solving the underlying differential equations numerically.

To this effect, 1D reaction–diffusion simulations were performed by the cellular automata approach[65,66,73] in circular geometry, with a resolution of 0.01 µm per space unit. This corresponds to the kinase homogenously localized to the membrane of a spherical GUV. For a diffusion coefficient of 20 $\mu m^2 s^{-1}$ of species 3 + 4), this translates to a diffusant radius of 3 space units and a time resolution of the simulation of 0.01 s per time step. The diffusant radius of species (1 + 2 + 5) is set to 4 space units (corresponding to a slightly too high D = 35 $\mu m^2 s^{-1}$, avoiding numerical inconsistencies). Initial conditions were set for all association/dissociation of all species in steady state without any kinase activity and setting kinase activity to its maximum instantaneously at $t_0$. Simulations continue until the relative change of the vectorial sum of all concentrations <$10^{-9}$ results in a steady-state distribution of all species. For dissociation/association reactions, the relevant rate constants (Supplementary Table 1) are: $k_{on,stath} = 1.6 \mu M^{-1} s^{-1}$, $k_{off,stath} = 4.8 \; 10^{-2} s^{-1}$, $k_{on,p-stath} = 1.28 \; 10^{-2} \mu M^{-1} s^{-1}$, $k_{off,p-stath} = 8 s^{-1}$ and remain fixed across all simulations. In most experimental conditions, 2–5 µM of soluble kinase are used with a corresponding activity of $k_{kin} = 0.004 s^{-1}$. We estimate the increase of concentration after translocation to the membrane by a factor of 200–500 to yield a kinase activity of $k_{kin} = 1 s^{-1}$ for a depletion of 50% soluble kinase. Conversely, the phosphatase activity at 1 µM can be estimated as $k_{PP\lambda} = 0.02 s^{-1}$. These parameters produce a spatial distribution of the species (1–5) depicted in Supplementary Fig. 6i (left panel). Higher diffusivity of phospho-stathmin as compared to the tubulin-stathmin-complex (molecular weight 220 kDa, D = 20 $\mu M^2 s^{-1}$) leads to a depletion of the total stathmin (free, phosphorylated and tubulin bound, cyan line in Supplementary Fig. 6i (left panel) at the membrane. This distribution is mirrored by the total tubulin due to its high association rate to stathmin.

Gradients of phosphorylated stathmin and free-tubulin result from the competition of kinase activity that effectively "deposits" tubulin near the membrane and entropic redistribution by diffusion as well as rebinding to dephosphorylated unbound stathmin as driven by dephosphorylation. An activity of lambda phosphatase of 0.02 $s^{-1}$ is sufficient to let the gradients decline within 1.5 µm distance from the membrane even at 10000 times faster kinase activity. Interestingly, varying the phosphatase activity for a kinase activity of 1 $s^{-1}$ in agreement with our measurements shows an optimally steep gradient for 100 times slower phosphatase activity. Low phosphatase activity can saturate the GUV with phospho-stathmin and therefore free tubulin due to the shift in partitioning and yields a flat gradient with a high offset. The highest phosphatase activity abolishes the phospho-stathmin gradient (zero amplitude and offset). Due to the association rate to (phospho-)stathmin being about 20 times faster than even kinase activity, the gradient of free tubulin is "locked" to the phospho-stathmin gradient (Supplementary Fig. 6j). We compared this simplified 5-species model to a more detailed representation of the 2:1 stoichiometry of tubulin:stathmin binding (low-affinity binding: $k_{on,stath} = 9 \; 10^{-1} \mu M^{-1} s^{-1}$, $k_{off,stath} = 1.86 s^{-1}$, $k_{on,p-stath} = 1.7 \; 10^{-1} \mu M^{-1} s^{-1}$, $k_{off,p-stath} = 2.3 s^{-1}$) with nine species (Supplementary Fig. 6k) and found little difference in the gradients. The availability of a low-affinity binding site with high turnover rate does not change the tubulin gradient, but results in additional ~4 µM free tubulin corresponding to ~60% occupancy of this site (Supplementary Fig. 6k).

We additionally performed simulations in 2D Euclidean geometry with no-flux boundary conditions and each pixel being either outside, lumen or membrane to assess the impact of local geometry of protrusions on the gradients of free tubulin (Supplementary Fig. 8i). Protrusions feature a much higher surface-to-enclosed-volume ratio compared to undeformed spherical GUVs. Assuming a homogeneous distribution of kinase activity per µm$^2$ of surface, we plot the gradient of free tubulin in a GUV with a radius of 32 µm, featuring protrusions with 250 nm diameter with varying lengths (0.5–22.5 µm). The kinetic parameters were kept identical to previous 1D simulation, specifically with the activities of phosphatase (0.02 $s^{-1}$) and kinase (0.004 $s^{-1}$ homogenously in the medium and 1 $s^{-1}$ homogenously at membrane).

**Statistics and reproducibility.** Examples represented in Fig. 1a were additionally observed in N = 8 independent experiments for low tubulin with rigid membrane, N = 30 independent experiments for high tubulin with rigid membrane and N = 12 independent experiments for high tubulin with deformable membrane. Morphometric parameters of individual GUVs and statistics are shown in Fig. 1b.

The temperature-induced aster growth in GUV with rigid membrane as exemplified in Fig. 1c and Supplementary Fig. 1e was observed in seven independent experiments. The temperature-induced aster growth in GUV with deformable membrane as exemplified in Fig. 1d and Supplementary Fig. 1f was observed in four independent experiments.

Light-actuated decentering of the centrosome in SynMMS with rigid membrane as shown in Fig. 5a was observed in four independent experiments. The corresponding control (SynMMS-stat) exemplified in Supplementary Fig. 7a, was measured in N = 3 independent experiments.

MT growth that changes morphology of SynMMS with large asters upon global illumination as shown in Fig. 5f was observed in five independent experiments.

Representative micrographs of initial SynMMS morphologies in Fig. 5k were observed as indicated in the bar plot quantification above, n = 45 across N = 15 independent experiments with 10 spherical, 19 polar and 16 star morphologies, respectively. These initial states exhibited types of protrusions as shown in Fig. 5l (n = 3 only MSPs, n = 13 only SPs, n = 9 MSPs and SPs, and n = 20 no protrusions).

Initiation of protrusions from a sparse (small) aster upon local activation of the signaling system as shown in Fig. 6i was observed in N = 3 independent experiments. The negative control as exemplified in Supplementary Fig. 8d measured for n = 6 across N = 2 independent experiments.

Enhanced SspB-AuroraB$^{488}$ accumulation in protrusions shown in Fig. 7a, e was observed in five independent light-dose-response experiments. SspB-AuroraB$^{488}$ accumulation in protrusions is additionally shown in Figs. 6a, i, 9a, j.

The reorientation of MTs towards a local stimulus as shown in Fig. 9a, f, j and Supplementary Fig. 10a was observed across eight independent experiments.

The lipid envelopment of protrusions as shown in Supplementary Fig. 7e was observed in three independent experiments. The bubbles occurring in some SynMMS were shown to be lipid vesicles as exemplified in Supplementary Fig. 7f for four independent experiments.

**Reporting summary.** Further information on research design is available in the Nature Research Reporting Summary linked to this article.

## Data availability
Data supporting the findings of this manuscript are available from the corresponding author upon reasonable request. A reporting summary for this Article is available as a Supplementary Information file. Source data are provided with this paper.

## Code availability
The custom code (python, c++, matlab scripts) used for simulations is available in a github repository under the link (https://github.com/akhileshpnn/SynMMS). Image analysis and postprocessing customization for ImageJ/Matlab is available upon request to the corresponding author.

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

## Acknowledgements

We thank Peter Bieling for reagents, help with setting up the TIRF-M assay, discussions and critically reading the manuscript, Roger Goody for assistance with the stopped-flow experiments, H. Schütz, K. Michel, M. Reichl, and S. Gentz for technical assistance and A. Krämer and A. Schindl for critical reading of the manuscript. This project was funded by the BMBF/MPG network MaxSynBio (031A359A).

## Author contributions

P.B. conceived and supervised the project and wrote the manuscript with help of K.G., B.S., F.G., H.S., M.S., and A.K. K.G. developed biochemical assays as well as COPYᵒ, K.G. and H.S. purified and encapsulated proteins, performed biochemical and imaging experiments. F.G. generated encapsulated asters, performed MT assays and imaging experiments, B.S. developed and performed image and morphometric analysis, A.N. and M.C.M. performed numerical simulations, M.S. developed and performed gradient reaction–diffusion simulations, A.K. and P.B. developed theoretical concepts.

## Funding

## Competing interests

The authors declare no competing interests.
