## [Peer Review File · Nature Communications]

Reviewer #1 (Remarks to the Author):

In the present manuscript the authors present a vesicle system, in which microtubule asters are encapsulated and the claim is that by a light switch the polymerization kinetics is locally altered which results in deformations of deflated vesicles.

The manuscript addresses an interesting question how local activations can result in local responses in cytoskeletal structures, yet it fails to present convincing results that this has been achieved. It is unclear to which extent the spatial control has really been achieved. The local activation is very inconclusive or even possible artefacts, especially as the experiments seem to be single vesicle observations. Overall the manuscript appears to be more of a series of bold claims without the necessary firm basis of relevant number of unambiguous observations. It is disturbing how many new and unnecessary buzz words are introduced – by this the overall writing is not clear enough. The reader is more mystified than enlightened. I do not see how this manuscript could be improved – a series of quantitative control measurements are needed and complete rewriting and refocusing is highly recommended.

Just to name a couple of major issues, which the authors should seriously consider to not harm their reputation or even the field:

- 1) It is completely unclear what benefit the comparison with the Rac pathway in Fig. 1 has to understand the experimental system here, which is based on the presence of bulk PPλ and its presumable local competition with the AuroraB as shown in Fig 3h. AuroraB is not activated at membranes in real word and clustering is autophosphorylation independent.
- 2) The membrane deformations seem to rather be bleb formation – often lacking even any microtubule signal. The control presented in Fig 4 is named SynMMS-stat. In the control in 4f it is unclear, if the localization of Aurora was achieved or not. Membrane curvatures are reported – yet in the absence of stat, all microtubules should be polymerized by the outset of the experiment. Why are not many more microtubules visible?
- 3) Important controls are missing – e.g. in Fig. 4a it needs to be reported how the mobility of the aster behaves over a longer time before activation. Also a light activation of the vesicle in the absence of AuroraB and presence of Stat is necessary, that not only local heating is appearing. An experiment with constitutively active AuroraB without light activation would be important too.
- 4) What is the baseline activity of AuroraB? Fig 2a suggest 20% - does this not then depend on the exact ratio of the compounds if it matters? Fig 5 and 6 have different aroaraB basal activity? I am very confused by the different buffers used – once with Mg once without, once with scavenger once without. I am unsure how this matters in terms of validity of controls.
- 5) In Fig. 4d I see many microtubules poking out of the membrane even in areas where no activation was performed or even before activation. Emergence and disappearance of protrusions during the same activation cycle (Fig 5 h, 6 e,j) are never explained. Why does in the presumed cycle the activation not result in visible microtubules? For all experiments it should be possible to resolve the polymerized microtubules clearly – which is not done and leaves me wondering why.
- 6) Fig 5. Why is there the clustering so visible and not in the previous experiments?
- 7) Fig. 5 where are the controls for deflated vesicles?
- 8) Their whole simulations part is based on the assumption of the interplay between they call “self-induced capture” of the microtubule protrusions, yet never show clear data of this happening or discuss this model and the clustering of AuroraB, whose cause is inconclusively explained and not really further investigated. Clustering of AuroraB is also very heterogeneously appearing in the different prestened pictures or movies.

9) What are the dark bubbles in the videos appearing within the vesicles? are these oil residuals from the production method? what control is there, to show that these do not influence measurements, protein activity and/or interpretations?

As these are only some of the major points to be raised, I highly recommend not further considering the manuscript.

Reviewer #2 (Remarks to the Author):

The manuscript by Gavriljuk et al. presents a synthetic morphogenic membrane system (SynMMS) reconstituted in vitro from purified components. The system is a cell-sized membrane bound compartment that encapsulates centrally organized microtubules and a light-gated signaling system. The signaling part consists in an out of equilibrium steady state system which involves constant cycling of Stathmin phosphorylation status thanks to the kinase AuroraB and the phosphatase PPlambda. The activity of AuroraB is controlled by the iLID-SSPB light-gated dimerizing system via light-induced partitioning of the kinase to the membrane. The authors propose that the behavior of the system is based on three feedbacks: 1) the stabilization of protrusions by diffusional capture of astral MT (SIC), 2) the enhanced clustering of membrane-bound AuroraB by already clustered AuroraB, and 3) the preferential recruitment of AuroraB to deformed membrane areas. Based on these three ingredients, they explore the dynamics and plasticity of SynMMS which presents three main morphologies (spherical, polar, and star-like). The work achieved is impressive. Given the difficulty to reconstitute out of equilibrium signaling systems, it is remarkable that they succeeded in actuating the organization of a proto-cytoskeleton. Overall the data presented are convincing, but the results are not fully consistent along the whole length of the manuscript which lead to uncertainties, especially regarding the feedbacks 2) and 3). Given clarifications on the points raised bellow, I would recommend its publication in Nat. Comm.

AuroraB clustering. The authors present evidences for AuroraB clustering on the membrane of GUVs. First, I was surprised to see that the very large clusters of AuroraB seems to correlate strongly with clusters of fluorescent lipids (fig 3c). Can the authors comment on that? Is it due to local defects of the membrane? If so, defects might play a very important role in the establishment and maintenance of protrusions and those defects are not considered at all. Second, whereas the dynamics of local light-induced gradients are well reported for PPlambda (sup fig3e), the local gradients of AuroraB are only presented at a single time point (fig 3b and 3i,j). How does the clustering of AuroraB affect the gradient extent and stability over time? From supp. fig3e, one can see that the PPlambda gradient is smoothed out rapidly (~50s) because of lateral diffusion. Does the same apply to AuroraB gradients? Or does the clustering sharpen the gradient over time because of coalescence? These points are important, for example to understand the data presented in fig 4d,e. Indeed, the induction of a first protrusion is clear and convincing, but the second one is less clear: given that it takes ~200s for the protrusion to appear, one would expect the gradient to be completely smoothed out by lateral diffusion on this time scale (as a side note, the authors should show the AuroraB recruitment heat map for this experiment). Third, the authors claim from fig3e that "light-induced translocation drives self-organized SspB-AuroraB pattern formation based on

substrate-depletion, thereby generating a persistent and patterned response". I don't see how the response is rendered more persistent since the activity of AuroraB on the membrane depends on the lifetime of the SSPB-iLID dimer and not on the clustering.

AuroraB recruitment to deformed membranes. An important part of the work is based on the fact that AuroraB is preferentially recruited to deformed membranes. However, this assumption does not appear consistent throughout the manuscript. In fig4h, the correlation is not obvious at all. Focusing on the early recruitment (5-10s), it looks like there is no correlation at all. The large deformation observed at 300° does not lead to AuroraB recruitment, and the largest patch of AuroraB at 150° correlate with a negatively deformed membrane patch. This seems to be in contradiction to authors conclusions (such as "This shows that SspB-AuroraB is preferentially recruited to MT-induced deformed membrane surfaces"). In Fig5f and 6g, the correlation is clear. Yet, in Fig 5i (first activation) and 6k (time 40-100s) one would expect a recruitment of AuroraB in the highly deformed membrane patches, which is not the case.

Star-like SynMMS. The fact that star-like morphologies can be due to either strong signaling or strong SIC lead to confusions when reading the last part of the manuscript and this last section (Dynamical features of SynMMS enables self-organized morphogenic responses) should be rewritten to be more didactical. Indeed, reading the first paragraph I thought that star-like morphologies were only observed when self-amplified clustering dominates SIC (fig 5b). But the third paragraph present the opposite: a star-like morphology where SIC dominates. An then a bit later we are back with the other limit case (fig 6e,g). At the end, I am wondering if the classification into a star-like morphologies is the good one or not. From the data, it looks like that the main difference is the protrusion morphology: pointed versus rounded. If the feedback in the signaling system dominates the protrusion is small and round; but on the contrary if SIC dominates the protrusion is long and pointed. These two morphologies seem to be mutually exclusive and I am not convinced by the possible plasticity between them (or at least I didn't see any data supporting this plasticity).

Reviewer #3 (Remarks to the Author):

Bastiaens and colleagues take a bottom up approach to build a synthetic «cell» capable of performing a minimal but already complex morphogenesis. This system is comprised of a GUV, a microtubule aster, an optogenetic synthetic signaling system that can be spatially controlled by local light illumination to regulate microtubules. This synthetic signaling system is inspired from a Rho GTPase network known to regulate microtubules. The authors perform a series of carefully designed experiments in which they incrementally test different components (mt aster, mt aster + synthetic signaling system, imposing different spatial cues by light on different non-equilibrium states that the microtubule lattice can generate). I think this work is a technical tour de force in bottom up cell free reconstitution systems. The experiments are also motivated by a strong modeling component (of which I cannot judge the quality due to absence of a strong modelling background).

The authors find that this synthetic system can adopt a number of different non-equilibrium cell morphogenesis states. An important lesson is that these morphogenetic states depend on the geometric state of the microtubule lattice within the GUV. These cell-intrinsic geometric states serve as an additional input that modulates the external spatially regulated signaling inputs imposed by optogenetic control. The study shows how the interplay of the intrinsic and external input co-operate to lead to the different non-equilibrium morphogenesis states. Understanding this interplay

provides causal evidence about how these different inputs regulate cell morphogenesis. Obtaining such causalities would be impossible with top-down approaches performed in real cells because it would be obscured by the large amount of feedback systems present in a cell. The clear value of this bottom up reconstitution approach is therefore to provide intuition about the emergent properties that a simple system can already adopt. I can see that this system can be equipped in the future with additional components (an actin cytoskeleton, an energy producing system, ...) which should be able to explore additional emergent properties of morphogenetic systems.

I really enjoyed reading this excellent work. I am ready to accept this work as soon as some key points have been addressed. The text is very dense, and sometimes I wished that the authors would go a bit more in depth in some experimental details rather than confining them in the methods section (I imagine this comes from previous paper submissions).

I do not have any major comments. Rather a lot of small things that have to be clarified:

1. first paragraph: what is the rationale for the low and high tubulin concentrations ? Does it come from the tubulin field, or does it result from empirical screening of a larger grid of tubulin concentrations.
2. Sspb-aurora-pattern: does the aggregation of this protein reflect any known in vivo property ? Or does it result because the recombinantly expressed protein might have some tendency to aggregate, which might be facilitated by recruitment to the PM, and local enhancement of its concentration. The authors bank on this property as an important input in the synthetic morphogenesis system. I would appreciate if the authors clearly state if this reflects a biological property of this protein, or is a simple consequence of poor folding of this protein. In the 2nd case, the authors should mention that this an "informative" artefact that is instructive about the prototype synthetic morphogenetic system.
3. Page 4, line 47: The authors mention: " these gradients mainly emanated from patterned structures,". They show Fig.3i. Please clarify how you see this ? I did not understand it clearly.
4. In figure 4, is the iLID optogenetic system orthogonal to the Alexa488-sspb-AuroraB visualization ? It seems to me that the absorption spectra are overlapping. Thus global 488 nm illumination might activate the iLID system ? This indeed might depend on the light intensity that is required for membrane recruitment of the AuroraB moiety.
5. For figures 5 and 6, how many times have the experiments been performed ? I missed a mention to the number of repeats. The authors can control precisely the concentrations of the components of their synthetic system, and I imagine they observe a characteristic population distribution of synMMS in different morphological states across different experiments. What is the phenotypical heterogeneity of this system given that its parameters have been fixed ? I am not asking for a lot of repeats of these complex experiments, rather for a strong comment the phenotypic heterogeneity of such systems (I am aware that noise in such experiments can depend on the quality of different batched of its purified components).
6. I would have appreciated that the authors would have come back in the discussion to the prototype Rho GTPase signaling network they have tried to mimic. They do not do come back to this at all, even if it is a strong selling point in the abstract. What can we learn about this signaling

network using the approach presented in this paper ? This would put the results in a perspective and make the work accessible to a wider audience. The authors have more space to discuss this now !

Reviewers' comments:

Reviewer #1 (Remarks to the Author):

In the present manuscript the authors present a vesicle system, in which microtubule asters are encapsulated and the claim is that by a light switch the polymerization kinetics is locally altered which results in deformations of deflated vesicles.

The manuscript addresses an interesting question how local activations can result in local responses in cytoskeletal structures, yet it fails to present convincing results that this has been achieved. It is unclear to which extent the spatial control has really been achieved. The local activation is very inconclusive or even possible artefacts, especially as the experiments seem to be single vesicle observations. Overall the manuscript appears to be more of a series of bold claims without the necessary firm basis of relevant number of unambiguous observations. It is disturbing how many new and unnecessary buzz words are introduced – by this the overall writing is not clear enough. The reader is more mystified than enlightened. I do not see how this manuscript could be improved – a series of quantitative control measurements are needed and complete rewriting and refocusing is highly recommended.

We thank the referee for the comments and criticisms that helped us to improve the manuscript. Following the suggestions, we have substantially restructured and re-written the manuscript to improve the clarity and avoid unnecessary jargon. In the amended version, we have included additional experiments/controls and corresponding quantifications as well as improved our theoretical analysis to clarify how local extracellular cues can lead to global morphological changes that depend on initial morphology.

Just to name a couple of major issues, which the authors should seriously consider to not harm their reputation or even the field:

It is completely unclear what benefit the comparison with the Rac pathway in Fig. 1 has to understand the experimental system here, which is based on the presence of bulk PPL and its presumable local competition with the AuroraB as shown in Fig 3h. AuroraB is not activated at membranes in real world and clustering is autophosphorylation independent.

The reason for the comparison with the Rac pathway is that the system was built to mimic signal transduction to the MT-cytoskeleton in a simplified manner that however still captures fundamental physicochemical processes of morphogen-induced morphogenesis in cells. We designed the system to respond to light cues instead of morphogens, while maintaining the basic principles of localized signal transduction in response to morphogens that generate intracellular MT-regulator signaling gradients. In that regard, we engineered SspB-AuroraB construct as a membrane recruitable stathmin kinase. We have now improved

the explanation behind the design of the system and elaborated on the analogy to the Rac pathway in the discussion, as well as summarized the conclusions in a new scheme presented in Fig. 10.

The membrane deformations seem to rather be bleb formation – often lacking even any microtubule signal. The control presented in Fig 4 is named SynMMS-stat. In the control in 4f it is unclear, if the localization of Aurora was achieved or not. Membrane curvatures are reported – yet in the absence of stat, all microtubules should be polymerized by the outset of the experiment. Why are not many more microtubules visible?

We have now thoroughly characterized the observed type of protrusions and show that these are caused by astral MTs (Fig. 5l, Fig. 6a,i, Fig. 7a,e, Fig. 9a,f,j). We have also quantified the translocation (R) of SspB-AuroraB for all presented cases that show that the localization was achieved (Fig. 5b,g, Fig. 6c,k, Fig. 9b,j,l).

In the absence of stathmin, there is a dynamic balance between free and polymerized tubulin that depends on the amount of tubulin in the system and therefore not all tubulin is polymerized. Stathmin changes this balance by sequestering free tubulin.

Important controls are missing – e.g. in Fig. 4a it needs to be reported how the mobility of the aster behaves over a longer time before activation. Also a light activation of the vesicle in the absence of AuroraB and presence of Stat is necessary, that not only local heating is appearing. An experiment with constitutively active AuroraB without light activation would be important too.

We have now added controls without stathmin where we compared the mobility of the aster before and after light-induced SspB-AuroraB translocation (Fig 5.d, Supplementary Fig. 6). We thereby show that in case of absence of stathmin, the aster mobility before and after activation does not change, whereas in the full system the mobility is enhanced after activation.

Visible light in biological fluorescence imaging of cells as used here has been shown to not induce any significant heating because of the negligible absorption of water in the visible spectrum (A. Schönle and S. Hell. *Opt. Lett.* , 23 (5), 325 – 327 (1998); W. J. Denk, D. W. Piston, and W. W. Webb, “Two-photon molecular excitation laser-scanning microscopy,” in *Handbook of Biological Confocal Microscopy*, J. B. Pawley, Ed., pp. 445–458, Plenum Press, New York (1995)).

Regarding SspB-AuroraB, the kinase is constitutively active in the presence of ATP in the lumen of the GUV, where it phosphorylates stathmin. However the higher relative activity of PP λ in the lumen maintains a low steady-state phosphorylation of stathmin (Fig. 4a,b). Actuation of the synthetic signaling system operates by locally concentrating AuroraB activity on the membrane upon light-induced translocation, thereby overcoming the luminal PP λ activity to generate a local (membrane proximal) phosphorylation gradient of stathmin (Fig. 4). This works even without an intrinsic activation mechanism of AuroraB activity

(conformational change, auto-phosphorylation), which could additionally enhance the amplitude of the pStathmin gradient, while maintaining the same decay length (Fig. 4f).

What is the baseline activity of AuroraB? Fig 2a suggest 20% - does this not then depend on the exact ratio of the compounds if it matters? Fig 5 and 6 have different aroaraB basal activity?

In Figure 3a (now Fig. 4a) we report an average and S.D. of the basal SspB-AuroraB translocation. This basal translocation mostly depends on the variable encapsulation efficiency of C2-iLID (Supplementary Fig.3d), which is reflected in the reported S.D. of translocation in the absence of light. We have now quantified both, the basal and the light-induced translocation for each experiment and used this to understand the response properties of the system (Fig. 5b,g, Fig. 6c,k, Fig. 9b,j,l).

I am very confused by the different buffers used – once with Mg once without, once with scavenger once without. I am unsure how this matters in terms of validity of controls.

For encapsulated asters alone, the base buffer contained 80 mM PIPES pH 6.8, 75 mM KCL, 2 mM MgCl₂, 1 mM EGTA, 1 mM Trolox and 0.1 mg/ml b-casein. In all experiments involving SspB-AuroraB translocation, stathmin gradients and SynMMS (controls, as well as full system), EGTA was consistently replaced with 0.8 mM MnCl₂ because PPλ is a metal dependent phosphatase. Oxygen scavengers were only used in the TIRF-M experiments where microtubule dynamics was characterized, in order to prevent bleaching. We have now better described these procedures in the methods.

In Fig. 4d I see many microtubules poking out of the membrane even in areas where no activation was performed or even before activation. Emergence and disappearance of protrusions during the same activation cycle (Fig 5 h, 6 e,j) are never explained. Why does in the presumed cycle the activation not result in visible microtubules? For all experiments it should be possible to resolve the polymerized microtubules clearly – which is not done and leaves me wondering why.

The microtubules originating from the encapsulated centrosomes exhibit dynamic instability, and therefore when the average aster size matches that of the GUV, they continuously poke and deform the membrane, even in the absence of external stimulus. The SynMMS in the old Fig.4d had such thin dynamic MT-protrusions due to the centrosome proximity to the membrane before activation. The protrusions that are visible after switching the activation to the other side on the other hand are there because the external signals lead to their stabilization. We have now clearly described this in the text related to the new Fig. 6. In the old Fig. 5h, 6e,i (new Figure 9) the dynamic pre-formed protrusions outside of the

irradiation area(s) also arise from dynamic instability of astral microtubules. In this context, we have now better explained how MTs can generate dynamic protrusions without light-induced SspB-AuroraB translocation, and how a positive feedback at the membrane upon light-induced SspB-AuroraB translocation can stabilize a protrusion. We now also have improved the image processing as well as the visualization in the figures to show that protrusions originate from astral-MTs in all experiments.

Fig 5. Why is there the clustering so visible and not in the previous experiments?

In the new version of the manuscript, the newly added SspB-AuroraB angular kymographs shows the clustering (Fig. 6).

Fig. 5 where are the controls for deflated vesicles?

We have now better structured the paper to first demonstrate that the signaling system can enhance MT-growth and include controls that demonstrate that this is not possible in the absence of stathmin in deflated vesicles (New Fig. 5 and Fig. 6 with associated supplementary fig. 7 and 8). The experiments shown in the old Fig.5 address the specific question of SspB-AuroraB accumulation in preformed protrusions (Fig. 5d-f->Fig. 7a-d) for which the controls are presented in Fig. 7i. The SynMMS in Fig. 5h-l has been moved to Fig. 9j-n to address the question how initial morphology relates to morphological response to local stimuli.

Their whole simulations part is based on the assumption of the interplay between they call “self-induced capture” of the microtubule protrusions, yet never show clear data of this happening or discuss this model and the clustering of AuroraB, whose cause is inconclusively explained and not really further investigated. Clustering of AuroraB is also very heterogeneously appearing in the different prestened pictures or movies.

We have now better described how self-induced capture of MTs in a protrusion (SIC) can be concluded from the observed coalescence of MTs in protrusions during temperature change induced growth of MTs of the encapsulated asters (Fig. 1d and Supplementary Fig. 1f). The newly added dynamical system analysis of the model that captures the process of local self-amplification (SIC) that depletes free substrate corroborates the self-organizing process that leads to formation of a polar morphology.

The experiments of encapsulating SspB-AuroraB/C2-iLID alone indicate that the dimensionality reduction on the membrane potentiates its oligomerization based on weak multivalent affinity. Our improved analysis in the amended manuscript of the SspB-Aurora patterns on the membrane (Fig. 3e,f) shows that the identified regularity of SspB-AuroraB clusters is explained by a cooperative clustering/monomer depletion model (Fig. 3e-i). We could exclude a phosphorylation dependent effect, since the cluster pattern was independent of ATP (Fig. 3f). The identified regularity in the self-organized patterns of SspB-

AuroraB are absent in SspB-PP λ (Fig. 3f), showing that AuroraB oligomerization is reversible and is an intrinsic molecular property of AuroraB, and not of the iLID/SspB optical dimerizer system.

What are the dark bubbles in the videos appearing within the vesicles? are these oil residuals from the production method? what control is there, to show that these do not influence measurements, protein activity and/or interpretations?

The peripheral distribution of fluorescent lipids and the C2-iLID/SspB-AuroraB on these occasional dark bubbles show that these are vesicles and not oil droplets (Supplementary Fig. 7f). These vesicles are likely created during the encapsulation process and we occasionally observed them in SynMMS. Comparison of light-induced responses of SynMMS that contain these vesicles and those that don't shows that the protrusion generating process does not depend on those vesicles.

Reviewer #2 (Remarks to the Author):

The manuscript by Gavriljuk et al. presents a synthetic morphogenic membrane system (SynMMS) reconstituted in vitro from purified components. The system is a cell-sized membrane bound compartment that encapsulates centrally organized microtubules and a light-gated signaling system. The signaling part consists in an out of equilibrium steady state system which involves constant cycling of Stathmin phosphorylation status thanks to the kinase AuroraB and the phosphatase PPlambda. The activity of AuroraB is controlled by the iLID-SSPB light-gated dimerizing system via light-induced partitioning of the kinase to the membrane. The authors propose that the behavior of the system is based on three feedbacks: 1) the stabilization of protrusions by diffusional capture of astral MT (SIC), 2) the enhanced clustering of membrane-bound AuroraB by already clustered AuroraB, and 3) the preferential recruitment of AuroraB to deformed membrane areas. Based on these three ingredients, they explore the dynamics and plasticity of SynMMS which presents three main morphologies (spherical, polar, and star-like). The work achieved is impressive. Given the difficulty to reconstitute out of equilibrium signaling systems, it is remarkable that they succeeded in actuating the organization of a proto-cytoskeleton. Overall the data presented are convincing, but the results are not fully consistent along the whole length of the manuscript which lead to uncertainties, especially regarding the feedbacks 2) and 3). Given clarifications on the points raised below, I would recommend its publication in Nat. Comm.

We thank the referee for their insightful comments that helped us improve our manuscript.

AuroraB clustering. The authors present evidences for AuroraB clustering on the membrane of GUVs. First, I was surprised to see that the very large clusters of

AuroraB seems to correlate strongly with clusters of fluorescent lipids (fig 3c). Can the authors comment on that? Is it due to local defects of the membrane? If so, defects might play a very important role in the establishment and maintenance of protrusions and those defects are not considered at all.

Indeed, as the referee noted, the large SspB-AuroraB patches directly correlate with local lipid irregularities on the membrane. These large patches occurred substantially more often in the absence of ATP, which indicates that these were denatured, inactive SspB-AuroraB aggregates. However, we had specifically excluded these lipid-defect regions (now displayed in the new Fig. 3e) when analyzing the spatial regularities of SspB-AuroraB clusters presented in Fig. 3f. This is now better described in the manuscript. However, these SspB-AuroraB aggregates do not deform the membrane and therefore are not affecting the establishment and maintenance of protrusions.

Second, whereas the dynamics of local light-induced gradients are well reported for PPlambda (sup fig3e), the local gradients of AuroraB are only presented at a single time point (fig 3b and 3i,j). How does the clustering of AuroraB affect the gradient extent and stability over time? From supp. fig3e, one can see that the PPlambda gradient is smoothed out rapidly (~50s) because of lateral diffusion. Does the same apply to AuroraB gradients? Or does the clustering sharpen the gradient over time because of coalescence?

We thank the referee for this question. According to this suggestion, we investigated in a greater detail how reversible AuroraB self-association creates a confined steady-state translocation gradient in response to a local light-signal (Fig. 3b-d). As the referee pointed out, this is highly relevant for the locality of the response in SynMMS. We have generated new data and improved our analysis of the gradient extent and stability to address this issue (new Fig. 3). We have also significantly rewritten the corresponding text.

These points are important, for example to understand the data presented in fig 4d,e. Indeed, the induction of a first protrusion is clear and convincing, but the second one is less clear: given that it takes ~200s for the protrusion to appear, one would expect the gradient to be completely smoothed out by lateral diffusion on this time scale.

The new data and analysis in the new Fig. 3 now shows that this gradient along the membrane reaches a stable steady-state in ~50s, and that it is well confined to the light-activation region and is not smoothed out at ~200s.

(as a side note, the authors should show the AuroraB recruitment heat map for this experiment).

We have now done so in the new Fig. 6m.

Third, the authors claim from fig3e that “light-induced translocation drives self-organized SspB-AuroraB pattern formation based on substrate-depletion, thereby generating a persistent and patterned response”. I don’t see how the response is rendered more persistent since the activity of AuroraB on the membrane depends on the lifetime of the SSPB-iLID dimer and not on the clustering.

We agree with the referee and have removed this comment on persistence.

AuroraB recruitment to deformed membranes. An important part of the work is based on the fact that AuroraB is preferentially recruited to deformed membranes. However, this assumption does not appear consistent throughout the manuscript. In fig4h, the correlation is not obvious at all. Focusing on the early recruitment (5-10s), it looks like there is no correlation at all. The large deformation observed at 300° does not lead to AuroraB recruitment, and the largest patch of AuroraB at 150° correlate with a negatively deformed membrane patch. This seems to be in contradiction to authors conclusions (such as “This shows that SspB-AuroraB is preferentially recruited to MT-induced deformed membrane surfaces”). In Fig5f and 6g, the correlation is clear. Yet, in Fig 5i (first activation) and 6k (time 40-100s) one would expect a recruitment of AuroraB in the highly deformed membrane patches, which is not the case.

We agree with the referee that the correlation was difficult to observe. This was due to two reasons: 1) the small protrusions of the SynMMS^{-stat} example used to demonstrate this property in the old Fig. 4h, and 2) the fact that MTs that interact transiently with the membrane also cause SspB-AuroraB local enhancement. The latter property is now better analyzed and described in the manuscript. In light of this, we have restructured our results and we now demonstrate that SspB-AuroraB is preferentially recruited to membrane deformations using experiments where the light-dose, and thereby SspB-AuroraB translocation to the membrane was step-wise increased in SynMMS with strong initial protrusions (new Fig. 7). The quantification presented in the angular kymographs demonstrates that pre-formed MT-protrusions preferentially accumulate SspB-AuroraB.

In old Fig. 5i (new Fig. 9m), the slight lower left bending of the membrane during first activation is not a protrusion but instead caused by MTs that span the membrane and therefore does not accumulate SspB-AuroraB. This became apparent when we analyzed the corresponding transmission images. To observe SspB-AuroraB recruitment by fluorescence, it is necessary to have the protrusions in focus. However, protrusions can be detected even when out-of-focus in transmission images. In the old Fig.6k, a protrusion was moving out of focus. This can be seen by comparing the fluorescence channel to the transmission channel in the new supplementary video 6. This comparative analysis also allowed us to better observe the dynamics of protrusion generation and stabilization, which is now better described for the old Fig. 6k (new Fig. 9a-e), old Fig. 5i (New Fig. 9j-n, Supplementary video 7) and the new Fig. 6a-f (Supplementary Fig. 8a).

Star-like SynMMS. The fact that star-like morphologies can be due to either strong signaling or strong SIC lead to confusions when reading the last part of the manuscript and this last section (Dynamical features of SynMMS enables self-organized morphogenic responses) should be rewritten to be more didactical. Indeed, reading the first paragraph I thought that star-like morphologies were only observed when self-amplified clustering dominates SIC (fig 5b). But the third paragraph present the opposite: a star-like morphology where SIC dominates. An then a bit later we are back with the other limit case (fig 6e,g). At the end, I am wondering if the classification into a star-like morphologies is the good one or not. From the data, it looks like that the main difference is the protrusion morphology: pointed versus rounded. If the feedback in the signaling system dominates the protrusion is small and round; but on the contrary if SIC dominates the protrusion is long and pointed. These two morphologies seem to be mutually exclusive and I am not convinced by the possible plasticity between them (or at least I didn't see any data supporting this plasticity).

The referee is right in noting that star-like morphologies can occur with strong SIC. This is due to the fact that astral-MT density is an additional parameter that affects the initial morphologies (now experimentally verified, Fig. 5m). However, we now also show that for equivalent encapsulated free tubulin concentrations, the star morphology mostly occurs in SynMMS as compared to the encapsulated aster alone (Fig. 5k), which shows that for equivalent astral-MT density, basal SspB-AuroraB signaling is necessary to *stabilize* a star morphology. This is also corroborated by the more extended theoretical analysis (Fig.8, Supplementary Fig. 9). Therefore, the star SynMMS with the extremely large dispersed centrosome cluster (old Fig. 6a, RCS=7) does not represent the regime in which the balance between signaling and SIC determines initial morphologies and have replaced it with a more informative case of a star-like SynMMS (RCS=2.7) with SPs and measurable low signaling that cannot *stabilize* protrusions but instead redistributes them towards a local activation area (New Fig. 9f-i). Importantly, we have now improved our description of how signaling interacts with the MT-aster by showing that the relevant feature that determines initial morphologies and response properties is in fact the balance of *self-amplified* signaling (cooperative clustering: CC) and *self-induced* capture (SIC).

The referee is also right in noting that there are two different types of protrusions that reflect the balance of self-induced capture (SIC) and strong signaling. We now described these two types of protrusion (Fig. 5l; Spiking Protrusions: SIC dominated; and Membrane Sheet Protrusions: SspB-AuroraB signaling dominated). However, the overall morphology is dictated by the balance of SIC vs CC for a given overall MT-density as determined by the tubulin nucleation on the centrosomes (experimental parameter: relative centrosome size: RCS). We therefore now describe that: 1) in addition to SspB-AuroraB basal signaling, astral-MT-density determines the distribution of protrusions on the GUV surface and therefore contributes to the overall morphology (in relation to Fig. 5), and 2)

strong SspB-AuroraB signaling at the membrane accelerates growth of single or few MTs, which results in their bending, favoring the formation of membrane sheet protrusions (MSPs) over spiking protrusions (SPs, in relation to Fig. 6).

Reviewer #3 (Remarks to the Author):

Bastiaens and colleagues take a bottom up approach to build a synthetic «cell» capable of performing a minimal but already complex morphogenesis. This system is comprised of a GUV, a microtubule aster, an optogenetic synthetic signaling system that can be spatially controlled by local light illumination to regulate microtubules. This synthetic signaling system is inspired from a Rho GTPase network known to regulate microtubules. The authors perform a series of carefully designed experiments in which they incrementally test different components (mt aster, mt aster + synthetic signaling system, imposing different spatial cues by light on different non-equilibrium states that the microtubule lattice can generate). I think this work is a technical tour de force in bottom up cell free reconstitution systems. The experiments are also motivated by a strong modeling component (of which I cannot judge the quality due to absence of a strong modelling background).

The authors find that this synthetic system can adopt a number of different non-equilibrium cell morphogenesis states. An important lesson is that these morphogenetic states depend on the geometric state of the microtubule lattice within the GUV. These cell-intrinsic geometric states serve as an additional input that modulates the external spatially regulated signaling inputs imposed by optogenetic control. The study shows how the interplay of the intrinsic and external input co-operate to lead to the different non-equilibrium morphogenesis states. Understanding this interplay provides causal evidence about how these different inputs regulate cell morphogenesis. Obtaining such causalities would be impossible with top-down approaches performed in real cells because it would be obscured by the large amount of feedback systems present in a cell. The clear value of this bottom up reconstitution approach is therefore to provide intuition about the emergent properties that a simple system can already adopt. I can see that this system can be equipped in the future with additional components (an actin cytoskeleton, an energy producing system, ...) which should be able to explore additional emergent properties of morphogenetic systems.

I really enjoyed reading this excellent work. I am ready to accept this work as soon as some key points have been addressed. The text is very dense, and sometimes I wished that the authors would go a bit more in depth in some experimental details rather than confining them in the methods section (I imagine this comes from previous paper submissions).

We thank the referee for their insightful comments that helped us improve our manuscript.

I do not have any major comments. Rather a lot of small things that have to be clarified:

first paragraph: what is the rationale for the low and high tubulin concentrations ? Does it come from the tubulin field, or does it result from empirical screening of a larger grid of tubulin concentrations.

The rationale behind the two different tubulin concentrations is that average MT length is a direct function of tubulin concentration shown in Supplementary Fig. 1c. With this information, we chose concentrations that generated asters sizes that were smaller than the GUV (15-25 μ M tubulin), or aster sizes larger than the GUV that can deform the membrane (35-40 μ M tubulin). We have now better described this in the manuscript.

Sspb-aurora-pattern: does the aggregation of this protein reflect any known in vivo property? Or does it result because the recombinantly expressed protein might have some tendency to aggregate, which might be facilitated by recruitment to the PM, and local enhancement of its concentration. The authors bank on this property as an important input in the synthetic morphogenesis system. I would appreciate if the authors clearly state if this reflects a biological property of this protein, or is a simple consequence of poor folding of this protein. In the 2nd case, the authors should mention that this an “informative” artefact that is instructive about the prototype synthetic morphogenetic system.

It has been previously shown that AuroraB clustering is important for its function (Hindriksen et al. (2017) Front Cell Dev. Biol. 5, 112). However, it is not clear if additional protein factors are necessary to achieve self-association or if this is purely based on dimensionality reduction (by sequestration on, for example, centromeric proteins). The experiments of encapsulating SspB-AuroraB indicate that the dimensionality reduction on the membrane potentiates oligomerization based on a weak multivalent affinity, thereby generating small clusters. However, in addition to small clusters, we also observed some large SspB-AuroraB patches on the membrane. These patches occurred substantially more often in the absence of ATP, indicating that these were denatured, inactive SspB-AuroraB aggregates. However, when analyzing the spatial regularities of SspB-AuroraB clusters presented in Fig. 3f, we specifically excluded these large aggregates by using a fluorescent lipid marker (now displayed in the new Fig.3e). From the identified regularity in the small SspB-AuroraB clusters we can conclude that the self-organized process of SspB-AuroraB cluster patterning (Fig. 3f) is based on *reversible* AuroraB self-association as an intrinsic molecular property, and not due to irreversible aggregation of misfolded protein.

Page 4, line 47: The authors mention: “ these gradients mainly emanated from patterned structures,”. They show Fig.3i. Please clarify how you see this ? I did not understand it clearly.

In old Fig. 3i, a 3D view (nearest-point-projection) of the confocal stack of FRET-ratiometric images shows bright spots, indicating that stathmin phosphorylation on the membrane is highest in discrete locations that most likely originate from high-activity SspB-AuroraB clusters (now Supplementary Fig. 5j). We have adapted the text to better describe this.

In figure 4, is the iLID optogenetic system orthogonal to the Alexa488-sspb-AuroraB visualization ? It seems to me that the absorption spectra are overlapping. Thus global 488 nm illumination might activate the iLID system ? This indeed might depend on the light intensity that is required for membrane recruitment of the AuroraB moiety.

We have now clarified in the text that 488 nm irradiation both activates the iLID and excites the fluorescence of Alexa488-SspB-AuroraB that is used for the observation of SspB-AuroraB translocation, to avoid spectral crosstalk with Alexa647-tubulin. However, our acquisition speed was fast enough to observe the translocation kinetics (including an estimation of R_0), which is now quantified for each presented experiment.

For figures 5 and 6, how many times have the experiments been performed ? I missed a mention to the number of repeats. The authors can control precisely the concentrations of the components of their synthetic system, and I imagine they observe a characteristic population distribution of synMMS in different morphological states across different experiments. What is the phenotypical heterogeneity of this system given that its parameters have been fixed ? I am not asking for a lot of repeats of these complex experiments, rather for a strong comment the phenotypic heterogeneity of such systems (I am aware that noise in such experiments can depend on the quality of different batched of its purified components).

We have now improved the classification of the initial morphologies (for the MT-aster system alone and the full system, SynMMS, new Fig. 5k), showing the variability, given estimated final tubulin concentrations. However, in spite of good encapsulation reproducibility for tubulin, SspB-AuroraB and stathmin, two major factors limited the controlled generation of initial morphologies: variable encapsulation of C2-iLID, as well as variable nucleation of MTs on centrosome clusters. The former determines both the basal as well as maximal level of signaling and thereby cooperative SspB-AuroraB clustering, whereas the latter determines the MT-density and therefore affects self-induced capture. However, the resulting phenotypic variability of SynMMS was used in an informative way because experimentally identified causalities allowed numerical investigation of the relation between initial states and response properties that were verified experimentally.

I would have appreciated that the authors would have come back in the

discussion to the prototype Rho GTPase signaling network they have tried to mimic. They do not do come back to this at all, even if it is a strong selling point in the abstract. What can we learn about this signaling network using the approach presented in this paper ? This would put the results in a perspective and make the work accessible to a wider audience. The authors have more space to discuss this now!

As the referee suggested we have now extensively elaborated on this in the discussion and summarized the conclusions in a new scheme presented in Fig. 10.

Reviewer #1 (Remarks to the Author):

I still find the manuscript very confusing and inconclusive.

Reviewer #2 (Remarks to the Author):

I thank the authors for their work on the revisions. The manuscript is much better now, more clear and more convincing. The authors answered all of my requests.

My only suggestion is regarding the colormap for the new figures: why did the author choose a green to black colormap which is very aggressive to the eye? I largely prefer the original one.

Mathieu Coppey

Reviewer #3 (Remarks to the Author):

I was already very supportive of the manuscript in its first submission. The authors have addressed all my comments. I think they have made several controls, and most importantly they have made a big effort to explain these series of complex experiments in a more didactic fashion. The work presented is extremely innovative, and I am sure that it will raise a lot of interest. I strongly support its publication in NCOMM.

Reviewer #4 (Remarks to the Author):

Please see the PDF file in the Attachment.

The study by Gavriljuk et al. investigates a *de novo* cell-like system that is intended to mimic key elements of the signal-guided cell morphogenesis: (a) dynamic microtubules (MTs) organized by the centrioles and confined within GUVs; (b) stathmin as a MT growth regulator; and (c) a kinase-phosphatase subsystem of which the kinase component is capable of light-induced delocalization to the GUV periphery, thereby producing a radial gradient of stathmin phosphorylation and, hence, a radial gradient of free tubulin. The authors present a step-by-step procedure for constructing such a 'protocell', and all steps as well as their sequence appear logical and well described. It is also particularly pleasing that many of the experiments were motivated by the fundamental theoretical knowledge of nonlinear dynamics and pattern formation in self-organizing reaction-diffusion (RD) systems. In my view, this further strengthens the immense experimental effort and distinguishes the presented work from other similar studies.

From the first round of peer review, it can be observed that the experimental part has been substantially improved. Without having a strong experimental background, I could not fully follow the main message throughout the manuscript. Unfortunately, I was not entirely convinced by the theory/simulation part in its current form as (a) a much more consistent description of the theoretical models and (b) more in-depth, physical justifications would be required. However, a critically revised version of the manuscript that addresses the major and minor concerns below definitely warrants a publication in *Nature Communications*.

MAJOR POINTS

(1) The presented paradigmatic model of morphogenesis is a classical Turing network of substrate-depletion systems in which the activator consumes the substrate for its own activation, leading to the depletion of the substrate by fueling the activator. This is an elegant choice because the model provides a minimal framework that covers a wide spectrum of Turing instability phenomena (i.e., diffusion-driven instability). However, I find that, as presented for the authors' particular case, the model lacks a lot of details and clarifications. Some examples (there are many more):

— not all variables and parameters are introduced and explained properly. For example, what exactly are the "u" and "v" variables in terms of physical quantities? Angular densities? Concentrations at the membrane surface? Numbers of clusters or captured MTs?

— the spatial coordinate "x" is not explained properly. From Fig. 8 and Supp. Figure 9, one might conclude that it is a sort of angular coordinate around the GUV. Then why not consider two angles, azimuthal and polar, as one would expect for a spherical topology? However, from Eqs. 1,2,3,4, it appears that "x" is a simple distance along a 1D "membrane". This discrepancy should be resolved. It is particularly important to define the modeled system geometry consistently throughout the manuscript because the pattern formation itself strongly depends on the geometry. Spherical topology (namely, the spherical part of the Laplace operator) induces interesting effects that would not be observed in a plain Euclidian space.

— the same arguments are valid for boundary and initial conditions. These are not mentioned in the Methods section whatsoever, but required to define the mathematical problem rigorously.

— coupling between the two subsystems is not defined explicitly. In particular, I was not able to find the analytical form of the function "g" in Eqs. 4. Does it coincide with the functional form of the reaction terms in Eqs. 5 derived using linear perturbation theory (strictly speaking, this is not guaranteed)?

(2) The authors make the assumption that the total number of MTs in the system remains constant (directly below Eqs. 1). This is, in fact, not true for dynamic MTs. What makes the authors confident that excluding the stochastic nature of MT growth/shrinkage from the equations is a valid assumption? The experimental part of the manuscript suggests that each SynMMS has a fixed amount of GTP (no external supply) that should decay as the MTs exhibit dynamic instability.

(3) Along similar lines, what is the role of stochasticity in this system? The processes of self-induced capture (SIC) and cooperative clustering (CC) of AuroraB at the membrane in real SynMMSs are not deterministic, such that a statistical mechanics description might be more appropriate. In this respect, I find it especially surprising that the presented RD model qualitatively

yields the key morphologies (star and polar) depending on initial conditions. However, the solutions of the RD system found by the authors would only be valid if they survive a thermal bath. Please comment.

(4) The authors note that the pattern formation "is preserved if only the membrane components of the SspB-AuroraB subsystem are considered", that is, only if the variable "w" (monomers in the lumen) is dropped in Eqs. 2. How is this step justified? By doing so, the authors postulate that the rate at which AuroraB is recruited at the GUV membrane is instantaneous. Given the size of the GUVs and their surface-to-volume ratios, this is counterintuitive. Please comment/improve in the text.

(5) The parameter choice for the bifurcation analysis (total concentration of AuroraB) is confusing. The explanations in the main text regarding "symmetry breaking" suggest that the strength of the subsystem's feedback loop "gamma" is a more appropriate choice. Without this feedback loop, the system would not exhibit cooperative clustering.

Also, what is the respective bifurcation parameter for the subsystem describing SIC of MTs?

Further, did the authors perform a similar bifurcation analysis for the joint system in Eqs. 4? One of the central finding/messages of the manuscript is that SynMMSs reside "in different morphological states depending on which intra-system feedbacks are dominant." Intuitively, I am somehow missing a bifurcation diagram using the ratio "gamma_1/gamma_2" as a parameter because it directly reflects the dominance of SIC over CC and vice versa. Ideally, this should also include a diagram relating the coupling between the subsystems to the intra-system feedback strengths. So far, the authors have investigated the system's dynamics only for selected sets of parameters.

(6) Overall, I believe it would really improve the theoretical part of the manuscript if the authors could (roughly) relate experimentally observed morphological states of the SynMMS (e.g., in Fig. 5) to the steady state solutions of the RD system in Eqs. 4, given that its parameters have been fixed. This would further increase the validity of the theoretical model used.

MINOR POINTS

(1) The authors should also cite the original studies of linear perturbation theory and cellular automata.

(2) The citation for the Xppaut software is missing.

(3) The custom-made Python code should be made available as supplementary material. This would improve the reproducibility.

(4) In "Quantification of SspB-AuroraB membrane translocation": typos in the formula?

(5) In "Quantifying regularity of SspB-AuroraB clusters on the membrane": What type of stereographic projection was used for the entropy calculations? How was the threshold parameter "epsilon" calibrated?

(6) In "Reaction-diffusion simulations of gradients": why did the authors assume a 1:1 stoichiometry between tubulin and stathmin? According to crystal structures and kinetic measurements, one stathmin molecule binds two tubulins.

MAJOR POINTS

(1) The presented paradigmatic model of morphogenesis is a classical Turing network of substrate depletion systems in which the activator consumes the substrate for its own activation, leading to the depletion of the substrate by fueling the activator. This is an elegant choice because the model provides a minimal framework that covers a wide spectrum of Turing instability phenomena (i.e., diffusion-driven instability). However, I find that, as presented for the authors' particular case, the model lacks a lot of details and clarifications.

We thank the referee for the constructive comments, which helped us improve the manuscript. As the referee noted, the paradigmatic model in which the activator depletes the substrate through self-amplification is a minimal framework that allows describing the dynamic features of the signaling as well as cytoskeletal modules and by coupling both modules represent a theoretical model of SynMMS. The clustering of SspB-AuroraB on the membrane is a classical reaction-diffusion (RD) system where the interconversion between clusters and monomers can be described by chemical mass-action kinetics with mass conservation, and must therefore be described through RD formalism. This enables the description of the patterning dynamics of this sub-system. In order to unify the collective dynamics of the coupled MT-membrane /AuroraB system into a single theoretical framework, we described the MT-membrane system with the same RD formalism. This abstraction was possible due to the experimentally identified rules of interconversion between free MTs and those captured by cooperative formation of SIC protrusions. To corroborate that the RD abstraction of the MT-membrane subsystem captures its central self-organizing features, we additionally provide an agent-based model implemented with Monte Carlo simulations that explicitly considers the stochastic MT dynamics. Although the RD abstraction in a simplified geometry does not provide detail about the stochastic MT dynamics, it captures those dynamical features of the MT-membrane/SspB-AuroraB system that allow us to relate biochemical parameters such as concentration and feedback strengths to the two observed initial states (polar and star), and predict how these initial states respond to light cues.

The details of this are provided as a response to specific points below, and also in the revised version of the manuscript/figures.

Some examples (there are many more):

— not all variables and parameters are introduced and explained properly. For example, what exactly are the "u" and "v" variables in terms of physical quantities? Angular densities? Concentrations at the membrane surface? Numbers of clusters or captured MTs?

u_1 / u_2 correspond to the angular densities of SIC-bundled MTs / clustered SspB-AuroraB on the membrane respectively, whereas v_1 / v_2 represents the 1D angular densities of single MTs / SspB-AuroraB monomers on the membrane. The description of the models in Methods has been revised for consistency, as well as the description in the figures and text.

- the spatial coordinate "x" is not explained properly. From Fig. 8 and Supp. Figure 9, one might conclude that it is a sort of angular coordinate around the GUV. Then why not consider two angles, azimuthal and polar, as one would expect for a spherical topology? However, from Eqs. 1,2,3,4, it appears that "x" is a simple distance along a 1D "membrane". This discrepancy should be resolved. It is particularly important to define the modeled system geometry consistently throughout the manuscript because the pattern formation itself strongly depends on the geometry. Spherical topology (namely, the spherical part of the Laplace operator) induces interesting effects that would not be observed in a plain Euclidian space.

In the 1D simulations, periodic boundary conditions were used and represent a closed circle

of unit radius ($R=1$), where the linear coordinate “ x ” is the angle in degrees with respect to the center of that circle, which we term “circular geometry”. To avoid confusion, we have reformulated the respective equations with a more precisely named variable “ θ ”. We have also added a small scheme to Fig. 1 to motivate how we arrive at an angular kymograph in g) from the scheme in e). Additionally, we provided a description of the abstraction leading to the formulation of the RD model for the MT-membrane sub-system.

Although, as the referee suggested, the spherical part of the Laplace operator can introduce much more complex behavior (i.e. formation of labyrinth or stripe patterns), in the experiments we only observed formation of polar and star-like morphology of the coupled AuroraB/MT-membrane system (Fig. 5k). We have therefore adopted the simplified description of the coupled system in a circular geometry as it captures both observed morphologies of star and polar, as well as enables basic predictions - how these basic patterns will change depending on global or local stimuli (Fig. 8). The 1D description also allows demonstrating how the basic collective behavior of the MT-membrane sub-system can generate MT-bundle patterns that resemble the experimentally observed polar morphologies (Fig.1). An exemplary 2D simulation representing SspB-AuroraB pattern formation on a membrane patch (Euclidean grid with no-flux boundary conditions) is additionally provided in Fig. 3h, although computationally expensive, as it could be more easily visually related to the corresponding experimental observation.

— the same arguments are valid for boundary and initial conditions. These are not mentioned in the Methods section whatsoever, but required to define the mathematical problem rigorously.

As mentioned in the answer to the previous comment, we used circular geometry in the RD simulations with periodic boundary conditions, whose description was likely not prominently visible in the previous version of the Methods, since they were included in a longer paragraph with the other the technical details of the simulations. According to the referee’s comment, we now have a separate sentence where the boundary and the initial conditions (uniformly distributed at random around the homogenous steady state) are described. For Fig. 3h, no flux boundary conditions are used, and initial conditions - uniformly distributed at random around the homogenous steady state.

- coupling between the two subsystems is not defined explicitly. In particular, I was not able to find the analytical form of the function “ g ” in Eqs. 4. Does it coincide with the functional form of the reaction terms in Eqs. 5 derived using linear perturbation theory (strictly speaking, this is not guaranteed)?

We apologize for the confusion; in the previous version of the manuscript we had only written the analytical form of “ g ” in relation to the ODE model, in order to avoid redundancy. The ODE system (Eqs.5) is an approximation of the corresponding PDE system (Eqs.4), and is used to calculate the bifurcation diagrams and thereby identify the parameter ranges resulting in patterning. Based on the linear perturbation analysis (LPA), the reaction terms in the ODE model must stem from the reaction terms of the PDE system (Holmes et al., Biophys. J. 108, 2015), and therefore the analytical form of the function “ g ” is equivalent.

To briefly summarize the ODE derivation from the PDE system, in the LPA approach, for $D_{u_i} \ll D_{v_i}$, and considering the limit $D_{u_i} \rightarrow 0, D_{v_i} \rightarrow \infty$, the stability of the homogenous steady-state of the PDE system can be probed with respect to a ‘local’ perturbation in the form of a narrow peak of u_i with a negligible total mass, at some location in the domain. In this limit, the localized peak of $u_i(u_{L,i})$ does not influence the surrounding background u_i levels ($u_{G,i}$), whereas due to the ‘infinitely fast’ diffusion, v_i can be described by a single global variable ($v_{G,i}$) that is approximately uniformly distributed in space. Thus, an ODE

system can be written for the evolution of $u_{L,i}$ and $u_{G,i}$. In the amended version of the manuscript we provide the functional forms in each step in the Methods section.

In contrast to the classical linear stability analysis, the linear perturbation analysis thereby allows to identify the type of bifurcation that characterizes the dynamical transition. Using these identified parameters, the reaction-diffusion simulations are then performed using the PDE systems.

(2) The authors make the assumption that the total number of MTs in the system remains constant (directly below Eqs. 1). This is, in fact, not true for dynamic MTs. What makes the authors confident that excluding the stochastic nature of MT growth/shrinkage from the equations is a valid assumption? The experimental part of the manuscript suggests that each SynMMS has a fixed amount of GTP (no external supply) that should decay as the MTs exhibit dynamic instability.

The assumption that the number of MTs in the system remains constant relies on the observations that the number of microtubules nucleated per centrosome reaches a plateau above 20 μM tubulin (Mitchison and Kirschner, Nature 1984). The amount of MTs in SynMMS is therefore constant and depends on the amount of clustered centrosomes that generate the centrally organizing center from which the MTs emanate. Furthermore, due to dynamic instability, MTs within this fixed population can be captured and can escape from SIC protrusions and this interconversion can therefore be described by a substrate depletion model using an RD formalism. This has been briefly described in the text.

The RD formalism abstracts the experimentally identified rules of interaction in the MT-membrane sub-system, where the cooperative formation of the SIC-bundled MTs depletes the free MTs (substrate-depletion model). To describe the physical aspects of growth and regulation of MTs, a statistical mechanics approach, i.e. using a master equation can be employed (Dogterom and Leibler, 1993; Weichsel and Geichssler, 2016). However, describing the membrane-mediated transition to collective MT organization into polar morphology as in the MT-membrane sub-system (Fig. 1), the self-induced capture must be additionally implemented, i.e. through a Monte Carlo simulation.

In order to validate that the dynamical description using the RD formalism captures the equivalent features as when explicitly modeling the stochastic MT nature, in the amended version of the manuscript we implemented an agent-based model that describes individual MT-dynamics (Dogterom and Leibler, PRL 70, 1993) together with the mechanism of SIC-induced membrane deformations and the effect this has on dynamic instability parameters, using Monte-Carlo simulations. In brief, a defined number of MTs are nucleated and evolve from fixed initial positions, mimicking a growth from a MT-organizing center. The model includes explicitly MT dynamic instability parameters of growth, shrinkage, catastrophe and rescue (Cassimeris et al., PLoS One 0197538, 2018). In addition, the model includes the effect of membrane deformation on these dynamic instability parameters, as well as a cooperative mechanism depending on membrane deformation that results in self-induced capture (complete details provided in the revised Methods).

The MC simulations show that the MTs organize into a polar or star-like distribution of MT bundles for lower or higher total amount of MTs respectively. Snapshots of the evolution of the system are shown below, as well as in Supp. Fig. 1g in the revised text. These patterns correspond to the ones identified from the RD-model, which also depended in a similar way on the total MT-amount (Fig. 1g and Supp. Fig. 1h,i).

Fig. 1 (as in Supplementary Fig. 1g in the manuscript). Top: Snapshots of the evolution towards a stable polar MT-bundle distribution obtained from a Monte Carlo simulation of the MT-membrane sub-system. Total MT amount = 20. Bottom: as above, only depicting evolution towards a star-like MT bundle distributions for total MT amount = 80. Dashed line depicts the position of the membrane.

SynMMS was also designed to retain an excess of GTP for the whole duration of the measurement (maximally ~ 1 h, including the preparation of the GUVs and the measurement). This was based on the reported rate of GTP hydrolysis (Carlier et al., *Biochemistry* 20, 1981; O'Brien et al., *Biochemistry* 26, 1987) at 40 μ M tubulin, where we estimated that the tubulin/MT system hydrolyses $\sim 10\%$ of the excessive GTP reservoir during the maximal time of an entire experiment of 1h.

(3) Along similar lines, what is the role of stochasticity in this system? The processes of self-induced capture (SIC) and cooperative clustering (CC) of AuroraB at the membrane in real SynSMMs are not deterministic, such that a statistical mechanics description might be more appropriate. In this respect, I find it especially surprising that the presented RD model qualitatively yields the key morphologies (star and polar) depending on initial conditions. However, the solutions of the RD system found by the authors would only be valid if they survive a thermal bath. Please comment.

In response to point 2 above, we used a Monte Carlo simulation to explicitly incorporate the stochastic MT nature and demonstrated that the RD formalism, although a simplification, equivalently well describes the self-organization into polar and star-like patterns depending on the total MT number. Further below in this comment, we also demonstrate that the obtained single and multi-peak solutions of the sub-system from the RD simulation (the MT-membrane and AuroraB are formally equivalent in this regard) have different wave numbers with maximum growth rate that are well separated, and thereby represent stable patterns.

Thus, we use the minimal RD framework, as it allowed qualitative description of the dynamics of the coupled system by means of bifurcation analysis using the LPA approach,

(Grieneisen V., PhD thesis, 2009) and uniquely allowed to identify the dynamical transition (pitchfork bifurcation), and to predict and sample the parameter space to identify the different patterns and the signal-induced transitions.

The different stable patterns are obtained for different values of the bifurcation parameter, as described in the text. In relation to the comment of the referee on initial states, we additionally checked how reproducible these patterns are for a fixed parameter set when changing the initial conditions. As shown below, for the coupled system (also now included as Supp. Fig. 9c,d), changing the initial conditions does not generally affect the global pattern, although the position of the peaks or the time for pattern stabilization can vary for different realizations. In the $SIC > CC$ case, due to the closeness of the patterns in parameter space, manifestation of polar morphology with single or two MT bundles was obtained in different realizations. We have also amended the revised manuscript to better explain the basis of the model.

Fig. 2 (as in Supplementary Fig. 9 c,d in the manuscript) Pattern realization under different initial conditions for the full system. Five different realizations of left) polar pattern obtained for $\gamma_1=1, \gamma_2=0.5, k_3=1, k_4=0.5$; right) star-like pattern obtained for $\gamma_1=1, \gamma_2=5, k_3=1, k_4=0.5$.

Comment on :“...the solutions of the RD system found by the authors would only be valid if they survive a thermal bath.” In our opinion, this question can be interpreted in two different directions, and we would like to comment on both.

If the comment refers to considering a thermodynamically extended description of the RD system (Falasco et al., PRL 121, 2018) where the temperature will be considered as a variable, then such a description would imply presence of a temperature variation/gradient that could influence the Turing space (Serna et al., PCCP 19, 2017) or the set of unstable modes (van Gorder, Proc. Royal Soc. A, 2020). In the experimental settings of the SynMMS however, isothermal conditions can be considered since strong temperature variations do not exist, and therefore in our opinion, such a description does not apply to the SynMMS.

If however the comment refers to the stability of observed patterned solutions of the RD model, we note that the chosen parameters are always set after the symmetry-breaking bifurcation. In response to the referee’s comment however, we additionally calculated the admitted unstable modes using the linear stability analysis for the parameter sets for which polar or star solution is observed in the RD simulations. As exemplified for the MT-

membrane sub-system below, the wave number with maximum growth rate was lower for the polar case and well separated as from the one of the star-like solution. Together with the bifurcation analysis provided in the manuscript, this demonstrates that the identified patterned solutions are stable.

Fig. 3. Admitted unstable modes of the MT-membrane sub-system (Eqs. 1 in Methods) estimated using LSA. $\gamma_1=1$, $k_0=0.067$, $\omega_1=0.5$ and $c_1=1.3$ for polar and $c_1=2.6$ for star-like pattern.

(4) The authors note that the pattern formation "is preserved if only the membrane components of the SspB-AuroraB subsystem are considered", that is, only if the variable "w" (monomers in the lumen) is dropped in Eqs. 2. How is this step justified? By doing so, the authors postulate that the rate at which AuroraB is recruited at the GU membrane is instantaneous. Given the size of the GUVs and their surface-to-volume ratios, this is counterintuitive. Please comment/improve in the text.

In the three component SspB-AuroraB system, monomers in lumen exchange with monomers on the membrane that interconvert to clusters, where mass conservation holds for the three interconverting species. This 3-component model yields equivalent SspB-AuroraB cluster patterns (Supp. Fig. 4f,g and Methods) as a 2 component model (with monomers on the membrane that interconvert to clusters) because under continuous light illumination, a fixed amount of SspB-AuroraB is translocated that is limited by the amount of available iLID on the membrane. Thus, the mass conserved 2-component model of AuroraB monomers and clusters on the membrane can qualitatively describe the experimentally observed pattern formation process on the GU membrane. In the amended version of the manuscript, we therefore keep throughout the 2-component model and have amended the text to explain that a fixed amount of SspB-AuroraB is translocated that is limited by the amount of available iLID on the membrane.

(5) The parameter choice for the bifurcation analysis (total concentration of AuroraB) is confusing. The explanations in the main text regarding "symmetry breaking" suggest that the strength of the subsystem's feedback loop "gamma" is a more appropriate choice. Without this feedback loop, the system would not exhibit cooperative clustering.

Following the answer to the previous question and the 2-component model for SspB-AuroraB, we use the total SspB-AuroraB amount on the membrane as a bifurcation parameter, as it is the only parameter that is experimentally controllable by light. We have however performed a 2-parameter (γ_2 vs. total AuroraB on the membrane) bifurcation

analysis (Fig. 4 in the response which we provide for the referee's perusal), showing that there is a wide range of parameter values for which symmetry breaking is observed:

Fig. 4. 2-parameter bifurcation diagram depicting that symmetry breaking can be observed in a broad γ_2 vs. total SspB-AuroraB amount on the membrane parameter range.

In the amended version of the manuscript we have provided an equivalent 2-parameter bifurcation diagram for the full system (Supplementary Fig. 9c).

Also, what is the respective bifurcation parameter for the subsystem describing SIC of MTs?

The bifurcation parameter is the amount of MTs in the system, which is the sum of free MTs and the SIC-bundled MTs.

Further, did the authors perform a similar bifurcation analysis for the joint system in Eqs. 4?

The bifurcation analysis of the joint system was shown in Supp. Fig. 9a. In the amended version of the manuscript, we additionally included the equivalent bifurcation diagram but when considering density of bundled MTs as a variable (Supp. Fig. 9b), and as noted above, a two-parameter bifurcation diagram to demonstrate that the symmetry-broken state can be observed in a broad parameter range (Supp. Fig. 9c).

One of the central finding/messages of the manuscript is that SynMMSs reside "in different morphological states depending on which intra-system feedbacks are dominant." Intuitively, I am somehow missing a bifurcation diagram using the ratio " γ_1/γ_2 " as a parameter because it directly reflects the dominance of SIC over CC and vice versa.

The bifurcation analysis approach (linear perturbation analysis, LPA) does not explicitly give an overview of which spatial patterns are stabilized, as noted in the response to comment 3 above. To identify the possible spatial pattern realizations, we have performed an exhaustive search using numerical simulations when starting from different parameters, given the correspondence between the LPA and the PDE simulations.

To maintain the focus of the manuscript around the experimental results of SynMMS, we have presented only a limited overview of the numerical simulations performed. In the

following figure, we demonstrate that changing the γ_1/γ_2 ratio results in a transition from a polar (ratio > 1) to a star-like (ratio < 1) pattern in a consistent way.

Fig. 5. Transition from polar to star morphology for changing γ_1/γ_2 ratio. A) $\gamma_1/\gamma_2=10$, B) $\gamma_1/\gamma_2=5$, C) $\gamma_1/\gamma_2=1$, D) $\gamma_1/\gamma_2=0.2$, E) $\gamma_1/\gamma_2=0.1$. The exact γ_1, γ_2 values are also given in the figure.

Ideally, this should also include a diagram relating the coupling between the subsystems to the intra-system feedback strengths. So far, the authors have investigated the system's dynamics only for selected sets of parameters.

As shortly described in the manuscript (Supp. Fig. 9c in the previous version of the manuscript, now Supp. Fig. 9f), we have investigated the influence of the inter-systems link strengths on the patterning characteristics for γ_1 or γ_2 dominance. For example, when $\gamma_1 > \gamma_2$, polar patterns are generated. We therefore asked whether in this case, increasing the coupling strength from the MT-membrane to the AuroraB system could “compensate” for the weak γ_2 and induce a star-like pattern. Increasing the coupling link strength even 2 orders of magnitude did not result in significant change, such that polar pattern with single or two MT bundles was always formed:

Fig.6 Polar pattern is formed even when the coupling strength from the MT-membrane to the SspB-AuroraB sub-system is increased for 2 orders of magnitude. A) $k_4/k_3=0.5$, B) $k_4/k_3=1$, C) $k_4/k_3=5$, D) $k_4/k_3=10$.

(6) Overall, I believe it would really improve the theoretical part of the manuscript if the authors could (roughly) relate experimentally observed morphological states of the SynMMS (e.g., in Fig. 5) to the steady state solutions of the RD system in Eqs. 4, given that its parameters have been fixed. This would further increase the validity of the theoretical model used.

We thank the referee for this suggestion. In the amended version of the manuscript, we have included references between the theoretical and experimental findings for the description of Fig. 5 and 9.

MINOR POINTS

(1) The authors should also cite the original studies of linear perturbation theory and cellular automata.

The linear perturbation theory has been initially described in Ref. 70, which is referenced in the manuscript. The respective references for the CA have been included

(2) The citation for the Xppaut software is missing.

We have included the respective reference.

(3) The custom-made Python code should be made available as supplementary material. This would improve the reproducibility.

Upon acceptance of the manuscript, we will include a citable GitHub reference where all of the codes will be deposited.

(4) In "Quantification of SspB-AuroraB membrane translocation": typos in the formula?

The typo has been amended.

(5) In "Quantifying regularity of SspB-AuroraB clusters on the membrane": What type of stereographic projection was used for the entropy calculations? How was the threshold parameter "epsilon" calibrated?

The 2D maps of the GUV surface were projected from confocal stacks using the Map3-2D software (Sendra et al., 2015). This projection roughly preserves pixel dimensions and maps the circular circumference of each slice in the z-Stack to a horizontal line. To further minimize distortions, we have restricted analyses (as indicated) to an equatorial crop of the projection.

The choice of ϵ reflects those from the available spatial recurrence analysis in the literature, for data that is normalized to mean 0 and standard deviation 1 (Marwan et al., 2006; Marwan et al., 2007). We would like to note that the same ϵ is kept fixed for the estimations from the different experimental conditions (SspB-AuroraB / SspB-PP λ) and numerical simulations, as well as the estimation of the entropy in the control cases, when randomizing the spatial patterns.

(6) In "Reaction-diffusion simulations of gradients": why did the authors assume a 1:1 stoichiometry between tubulin and stathmin? According to crystal structures and kinetic measurements, one stathmin molecule binds two tubulins.

We have now performed additional simulations involving a low and high affinity tubulin binding site (9 species) on stathmin and show that the tubulin gradient shape is hardly changed (Supplementary Fig. 6k). The comment of the referee showed us that there was a mix-up in the description of the change in the K_D 's for the low and high affinity binding sites in the supplemental text that we had however correctly used in the simulations. We have now corrected the change in k_D values stated in the methods and clarified the description of the rationale of using a simplified model of the regulation of tubulin-stathmin-dimerization by phosphorylation: Our kinetic measurements of tubulin binding show only a moderate change in the tubulin occupation of the low-affinity binding site upon phosphorylation of stathmin ($k_D \sim 2\mu\text{M}$ unphosphorylated and $13\mu\text{M}$ phosphorylated, with 64% and 34% occupancy, respectively, if there was 1:1 binding of $10\mu\text{M}$ stathmin and $10\mu\text{M}$ tubulin in solution). Therefore, the membrane-proximal release of tubulin from stathmin is mostly driven by the change in affinity of the high affinity binding site upon phosphorylation that switches dimerization from 95% ($k_D \sim 30\text{nM}$ unphosphorylated) to 1.6% ($k_D \sim 610\mu\text{M}$

phosphorylated). The occupancy of the low affinity binding site only affects the amount of free tubulin ($\sim 4\mu\text{M}$ more as compared to a 1:1 stoichiometry) and not the shape of the resulting tubulin gradient.

Reviewer #4 (Remarks to the Author):

I thank the authors for addressing all the concerns. The model description has been substantially improved, whereas additional modifications in the main text have helped to further improve the connection between experiment and simulation. The implemented model to allow for stochastic microtubule dynamics and cooperative bundling has been particularly convincing. Thus, I would like to strongly support the publication of the improved manuscript in Nature Communications.

My original comment with respect to "surviving a thermal bath" referred to cases in which the steady-state solution of a RD system is disturbed by fluctuations, causing the parameter subspace (on which it exists) to shrink, vanish or shift in an erratic manner. However, the Monte Carlo simulations provided by the authors largely resolve my question.

Best, Maxim Igaev.